# Towards Explicit Discrete Holography:
# Aperiodic Spin Chains from Hyperbolic Tilings

Pablo Basteiro, Giuseppe Di Giulio[*], Johanna Erdmenger, Jonathan Karl,
René Meyer and Zhuo-Yu Xian

Institute for Theoretical Physics and Astrophysics and
Würzburg-Dresden Cluster of Excellence ct.qmat, Julius-Maximilians-Universität
Würzburg, Am Hubland, 97074 Würzburg, Germany
* giuseppe.giulio@physik.uni-wuerzburg.de

## Abstract

We propose a new example of discrete holography that provides a new step towards establishing the AdS/CFT duality for discrete spaces. A class of boundary Hamiltonians is obtained in a natural way from regular tilings of the hyperbolic Poincaré disk, via an inflation rule that allows to construct the tiling using concentric layers of tiles. The models in this class are aperiodic spin chains, whose sequences of couplings are obtained from the bulk inflation rule. We explicitly choose the aperiodic XXZ spin chain with spin 1/2 degrees of freedom as an example. The properties of this model are studied by using strong disorder renormalization group techniques, which provide a tensor network construction for the ground state of this spin chain. This can be regarded as discrete bulk reconstruction. Moreover we compute the entanglement entropy in this setup in two different ways: a discretization of the Ryu-Takayanagi formula and a generalization of the standard computation for the boundary aperiodic Hamiltonian. For both approaches, a logarithmic growth of the entanglement entropy in the subsystem size is identified. The coefficients, i.e. the effective central charges, depend on the bulk discretization parameters in both cases, albeit in a different way.

# 1 Introduction

The holographic principle [1,2] is a fundamental paradigm in theoretical physics that states a deep relation between gravitational theories in $(d+1)$ dimensions and quantum field theories (QFTs) in $d$ dimensions at their boundary. The most tractable and well understood realization of the holographic principle is the AdS/CFT correspondence [3–5]. It states a duality between bulk gravity theories in a negatively curved Anti-de Sitter (AdS) spacetime and conformal field theories (CFTs) defined on the asymptotic boundary of this spacetime. In recent years, concepts from quantum information theory have been introduced into the AdS/CFT correspondence, such as entanglement entropy [6–8] and quantum complexity [9–13]. Driven by this fundamental relation between quantum information and holography, the complete reconstruction of bulk geometries from information-theoretic data of the boundary theory has been proposed [14].

Due to its string theory origin, the AdS/CFT correspondence is defined in terms of continuous variables, e.g. quantum fields being smooth functions over continuous spacetimes. On the other hand in what we collectively denote in this work as *discrete holography*, discrete variables are considered and in particular spacetime is taken to be discrete. Interest in a discrete holographic duality has gained momentum recently. Various approaches have been proposed in this direction and have shed light onto the properties that such a discrete duality ought to have. On the one hand, progress in the simulation of hyperbolic space through experimentally accessible topolectric circuits [15–20], as well as the mathematical characterization of the underlying discretization of hyperbolic space [21, 22], open a promising door to realizing holographic predictions in the laboratory. On the other hand, mathematical investigations of string theory based on discrete number fields such as the **p**-adics $\mathbb{Q}_{\mathbf{p}}$ give rise to **p**-adic AdS/CFT [23–25], which allows to gain insight into continuum properties of holography through *adelic formulas* [26]. A further formal approach to finding discrete holographic dualities is *modular discretization* [27–29], where coset constructions for $\text{AdS}_{1+1}$ and $\text{CFT}_1$ are exploited to construct discrete Hilbert spaces for both theories. Additionally, tensor network (TN) constructions provide important tools for realizing holographic dualities [30–43] in general and for the construction of a discretized bulk in particular. These reproduce some features of holography, such as the Ryu-Takayanagi (RT) formula for holographic entanglement entropy [31,34,40,43]. However, an open question of tensor network approaches based on regular hyperbolic tilings is the exact nature of the boundary theory. Recent progress in this direction was obtained in [41] by considering a disordered Ising chain on the boundary of matchgate tensor networks. This construction relies on a renormalization in radial direction towards the boundary in order to find the proper boundary theory, whose ground state is approximated by the tensor network contraction. However, despite the recent progress, a complete discrete holographic duality has not yet been realized at the dynamical level, in the sense that an equality of partition functions based on field-operator maps has not yet been found.

In this work, we propose a new step towards establishing a discrete holographic duality by investigating a regular discretization of the bulk and aperiodic spin chains on its boundary. In this construction the bulk discretization gives rise to a dynamical boundary theory in a natural way. Aperiodic spin chains have attracted a lot of attention after the discovery of quasicrystals [44] and are very well-known in condensed matter theory [45–50]. Specifically, we start from regular hyperbolic tilings, which are canonical discretizations of hyperbolic space and have previously been considered in many studies of discrete hyperbolic geometry [15,19,21,31,39–41,51–56]. These tilings are characterized by their Schläfli symbol $\{p, q\}$, denoting a tiling where $q$ regular $p$-gons meet at each vertex. The infinite set of vertices of these hyperbolic tilings define a discrete geometry that approximates that

of continuum hyperbolic space. In particular, this discrete geometry can be constructed using concentric layers of tiles and has a boundary whose fractal, aperiodic structure can be investigated by means of *vertex inflation rules*. These classify the vertices using two letters associated to the number of neighboring edges in the same layer. For all pairs $\{p, q\}$ with $q \neq 3$, it has been shown that two such letters are sufficient to characterize the entire tiling [52]. In this way, the tiling and its boundary can be generated by recursive iteration of an appropriate substitution (inflation) rule on an initial set of vertices (letters). In this work, we focus only on the properties of "empty" AdS and thus consider solely the hyperbolic tilings, without any matter fields. We explain how to define and compute the length of discrete geodesics in terms of the characteristic lengths of polygons in the tiling.

The construction of the tilings through inflation rules allows us to define an explicit boundary theory in a natural way that incorporates the aperiodic structure of the bulk. Specifically, we define an aperiodic XXZ chain on the boundary, whose modulation is determined by the choice of hyperbolic tiling in the bulk. This means that the couplings along the XXZ chain are not homogeneous and follow an aperiodic sequence determined by the inflation rule of the bulk tiling. The spin variables of the XXZ chain are chosen to be spin $1/2$, because in this case the model becomes computationally tractable and the features of such aperiodic chains are well-understood in the literature [57, 58]. Even in the presence of aperiodic disorder, this model is critical in a certain regime of the anisotropy parameter $\Delta_0$ that appears in the Hamiltonian of the XXZ spin chain. This is an attractive property from the point of view of holography, since boundary theories in AdS/CFT describe physical systems at criticality.

We are aware that choosing spin $1/2$ degrees of freedom implies that we are not in a large $N$ regime required to suppress quantum gravity effects. As our results show, the structure of discrete holography will be very different than in the continuum case, and it is yet to be determined how the large $N$ limit will enter. We leave this for future work. Here we point out that our construction has the new feature that the bulk discretization itself determines the couplings of the boundary theory, a property that by definition is not realizable in a continuum setting. It remains an open question how to obtain a precise mapping between bulk and boundary in discrete holography. However, we expect our results to be useful in completing this program.

In order to study the critical properties of the model described above, we employ strong-disorder renormalization group (SDRG) techniques. This tool assumes that we are working at low energies and that one coupling constant $J$ is much larger than the other. We argue that the aperiodicities induced by the discrete tilings in the bulk act as relevant perturbations in the case of the XXX chain ($\Delta_0 = 1$) on the boundary, in the sense that the system flows to a new, strong-disorder induced fixed point with respect to the homogeneous model. Finally, based on this SDRG approach, we construct a tensor network that exactly reproduces the ground state of aperiodic XXX chains at the boundary of hyperbolic tilings. The natural holographic structure of TNs allows us to reconstruct the bulk by embedding this TN onto the Poincaré disk. We find that the structure of this TN is different from that of the tiling, due to the fact that the TN is constructed through SDRG which selects a specific direction on the Poincaré disk. We explain how the global symmetries of the boundary Hamiltonian emerge within our TN construction. Moreover, we fully characterize the symmetries of the TN graph, finding that they do not match those of the corresponding tilings. This is in contrast to previous works, where TNs are constructed on hyperbolic tilings in such a way that each tensor has the same symmetry as a tile [31, 34, 40].

The setup we consider provides an explicit description of dynamical degrees of freedom on the boundary theory, but does not address any dynamical fluctuations in the discretized

bulk. In this sense, we do not provide a complete duality. Nevertheless, the natural way in which the boundary theory can be constructed from the discretized bulk suggests that we can provide a first step in this direction by considering the behavior of known holographic quantities in our discrete setup. As a prime example, throughout our analysis, we consider entanglement entropy as a benchmark of the different setups we investigate, since it is a well-understood quantity in the context of both quantum spin chains and tensor networks. Importantly, it has a holographic description due to Ryu and Takayanagi [6, 7], which is a reference point of our analysis. The Ryu-Takayanagi (RT) formula states that the bipartite entanglement entropy of a region $A$ of the boundary CFT is proportional to the the area $|\gamma_A|$ of the minimal codimension-one hypersurface $\gamma_A$ in the bulk that is homologous to $A$,

$$S_A = \frac{|\gamma_A|}{4G_N} \, , \qquad (1)$$

with $G_N$ being Newton's constant. It was shown in [6, 7] that evaluating (1) in $\mathrm{AdS}_{2+1}$ reproduces the known behavior for entanglement entropy in two-dimensional CFTs [59–63]. The RT formula (1) for holographic entanglement entropy relies on the large $N$ limit of the AdS/CFT correspondence, which is equivalent to a low-energy limit and suppresses quantum gravity effects. This limit is also equivalent to assuming a holographic CFT with a large central charge $c$, which is related to bulk quantities through the Brown-Henneaux formula $c = \frac{3R}{2G_N}$, with $R$ being the curvature radius of AdS [64]. The central charge $c$ can be seen as a measure of the number of local degrees of freedom of the theory. Away from the large $N$ or large $c$ limit, the RT formula is conjectured to admit higher corrections originating from entanglement entropy in the bulk [65].

In spite of this, the simple yet fundamental character of the RT formula makes it a useful quantity to kick-start approaches to discrete holography. A central aspect of this work is to compare different approaches to entanglement entropy in a discrete setup, both by assuming the validity of the RT formula in a discretized bulk geometry and by calculating the entanglement entropy directly for the aperiodic spin-1/2 chain. We note that further proposals for characterizing the RT formula in discrete settings, in particular through tensor networks, can be found in [31, 34, 40].

The main results of this work are summarized as follows. We provide a natural, simple and well-defined method for defining an explicit theory with dynamical degrees of freedom on the boundary of hyperbolic tilings. Our construction relies on a minimal number of ingredients, namely the inflation rule associated to each $\{p, q\}$ tiling. The aperiodic structure of this boundary allows us to naturally define a boundary theory by aperiodically modulating the couplings in an XXZ chain with spin-1/2 variables according to the asymptotic sequence generated by each $\{p, q\}$ inflation rule. We also construct a tensor network that exactly reproduces the ground state of this model in the regime where the anisotropy parameter is $\Delta_0 = 1$ (XXX chain), and which implements an SDRG flow. We thus obtain a discrete geometric structure in the bulk based on the Hamiltonian of the boundary theory. This can be interpreted as bulk reconstruction in the spirit of AdS/CFT.

Let us stress that our TN construction is different from that proposed in [41] in various aspects: First, our method for defining a boundary theory is independent of a tensor network construction in the bulk, instead relying solely on the inflation rules for $\{p, q\}$ tilings. A TN construction is nonetheless possible *a posteriori* in our setup and provides an exact description of the ground state of the XXX chain on the boundary rather than an approximation. Second, the couplings of the boundary model in [41] do not follow the simple aperiodic sequence of the tiling's boundary, but are rather determined by a construction denoted as *multi-scale quasi-crystal ansatz* [40, 41]. This relies on an RG flow of the couplings from the center of the tilings (IR regime in the holographic RG sense) to the boundary

(UV regime). On the other hand, our TN describes an RG flow of the couplings from the UV to the IR, which is more reminiscent of the standard notions of holographic RG [66–73].

We also obtain results for the entanglement entropy, which we compute via two different methods. In the bulk, we provide a straightforward geometric argument that approximates the length of discrete geodesics on the tiling, taking into account the fractal structure of the boundary. This allows us to derive a discrete form of the RT formula (1), given in (20), exhibiting a logarithmic growth of entanglement entropy with the subsystem size. For the boundary theory, we extract the effect of the aperiodicity on the entanglement entropy of a subsystem of the chain via SDRG. This includes a generalization of previous results for entanglement entropy in aperiodic chains to a larger class of modulations [74, 75]. We obtain a piecewise linear behavior of the entanglement entropy (57), with logarithmic enveloping functions (58). The proof we provide for these results is applicable to a given class of $\{p, q\}$ inflation rules.

For both approaches provided in our work, a logarithmic growth of entanglement entropy with the subsystem size thus appears. The coefficients of these logarithms can then be interpreted as *effective central charges*. Our results exhibit a remarkable dependence of the effective central charges on the parameters $p$ and $q$ of the tiling in the bulk. While the functional dependence on $p$ and $q$ is different in both cases, this implies that the geometry of the bulk influences the entanglement structure of the boundary theory, similarly to usual continuum holographic dualities. In one case this dependence directly comes from the discretization of the RT formula, while in the other case it arises from the fact that the boundary Hamiltonian depends on $p$ and $q$ by construction.

In order to venture beyond this first naive comparison and to find agreement of the $p, q$ dependence of the effective central charges in the bulk and boundary computations, it will be necessary to explore different dynamical degrees of freedom than the spin $1/2$ XXZ chain at the boundary, in particular to understand the role of the large $N$ limit. Both the discretization of the bulk action and the specific choice of boundary degrees of freedom will have to be investigated. Let us stress that while our construction focuses on an explicit Hamiltonian for the boundary theory, the AdS geometry is discretized but so far left without dynamics. Thus, it is promising to pursue generalizations of our setup that include gravitational fluctuations in the discretized bulk. This could be achieved for example via introduction of bulk scalar fields, or by allowing for graph fluctuations of the tilings itself. We provide a more detailed discussion of these future perspectives in Sec. 7.

This work is structured as follows. In Section 2 we introduce Anti-de Sitter spacetime in 2+1 dimensions and explain how to discretize a constant time slice of it through regular hyperbolic tilings. A derivation of a discrete version of the RT formula is contained in this section. Section 3 introduces our proposed dual theory on the boundary of the tilings, namely an aperiodically disordered XXZ quantum spin chain. Strong-disorder renormalization group techniques are reviewed in this section and used to characterize the relevance of the aperiodic disorder with respect to the homogeneous case. We follow up with Section 4 where we provide the derivation of the entanglement entropy for the XXX chain, for which aperiodicity shifts the system to a new, disorder-induced fixed point. Section 5 provides the detailed construction of a TN that reproduces the ground state of the aperiodic XXX model, while also providing a reconstruction of a discretized bulk. Symmetries and properties of the resulting TN are discussed at length in this section. The aforementioned sections contain a small summary of their content and results at the end. Section 6 is devoted to a comparison of the results within this paper with those of previous works [40, 52] studying hyperbolic tilings in a holographic setting. Finally, we conclude in Section 7 with

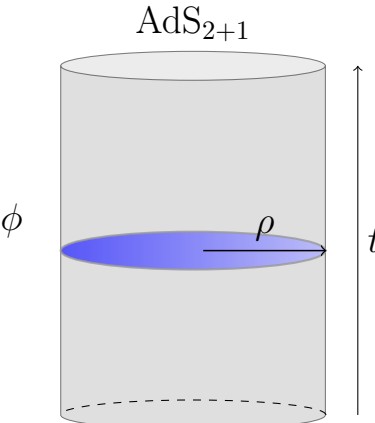

Figure 1: AdS$_{2+1}$ spacetime in the global coordinates (2). The conformal boundary coincides with the surface of the cylinder and is located at $\rho \to 1$. A constant time-slice (blue) is isomorphic to the Poincaré disk $\mathbb{D}^2$. The induced metric on this manifold is given by (3), which has an equivalent formulation in Poincaré coordinates (5).

a summary of the main results of this paper, as well as with promising perspectives for future work. Some technical details and additional results are reported in Appendices A-C.

## 2 Regular hyperbolic tilings

We begin by explaining in detail how regular hyperbolic tilings can be embedded in AdS$_{2+1}$, providing a natural discretization for the spatial part of the manifold. We elaborate on the properties and features of these tilings and show how they can be systematically constructed via inflation rules.

### 2.1 From AdS$_{2+1}$ to hyperbolic tilings

We consider AdS$_{2+1}$ in global coordinates $\{\rho, t, \phi\} \in \{[0,1), \mathbb{R}, [0, 2\pi)\}$ with invariant line element

$$ds^2 = R^2 \frac{-(1+\rho^2)^2 d\tau^2 + 4\, d\rho^2 + 4\rho^2\, d\phi^2}{(1-\rho^2)^2} \, . \tag{2}$$

Here, $R$ is the AdS radius and defines the constant negative curvature $K$ of the manifold via $K = -\frac{1}{R^2}$. The conformal boundary lies at $\rho \to 1$. In the context of the AdS/CFT correspondence, the CFT whose ground state is holographically dual to the AdS vacuum is defined at this boundary. More specifically in this case, the dual CFT is defined on a circle $S^1$ with circumference $\mathcal{L}$. The coordinates in (2) make the cylinder topology $\mathbb{R} \times S^2$ of AdS$_{2+1}$ manifest, as shown in Fig. 1.

Cauchy surfaces of (2) at constant time $t = const$ are isomorphic to the Poincaré disk model $\mathbb{D}^2 = \{w \in \mathbb{C} | |w| < 1\}$ of hyperbolic space in polar-like coordinates $w = \rho e^{i\phi}$. This can be seen as the Euclidean version of two-dimensional AdS as well, i.e. EAdS$_2$. The hyperbolic metric induced by (2) on this manifold is

$$ds^2 = (2R)^2 \frac{d\rho^2 + \rho^2\, d\phi^2}{(1-\rho^2)^2} \, . \tag{3}$$

Geodesics w.r.t. the Poincaré metric (3) are given by circle segments that are perpendicular to the unit disk and diametric lines. The distance between two points $w_1 = \rho_1 e^{i\phi_1}, w_2 =$

$\rho_2 e^{i\phi_2} \in \mathbb{D}^2$ is given by

$$d(z_1, z_2) = R \operatorname{arccosh} \left( 1 + \frac{2(\rho_1^2 + \rho_2^2 - 2\rho_2\rho_1 \cos(\phi_1 - \phi_2))}{(1 - \rho_1^2)(1 - \rho_2^2)} \right) . \tag{4}$$

Equivalently, we can describe the hyperbolic disk $\mathbb{D}^2$ via so-called *Poincaré patch* coordinates,

$$ds^2 = R^2 \frac{dz^2 + dx^2}{z^2} , \tag{5}$$

with $z \in (0, \infty)$, $x \in \mathbb{R}$. The conformal boundary now lies at $z \to 0$. Since we consider Euclidean AdS space, the Poincaré patch covers the entirety of the spacetime, as opposed to the Lorentzian case, where it would only cover a patch. These coordinates allow for easier computations and will be used for the derivation of coordinate-independent quantities in the following sections. However, we keep the global coordinates description of (3) for a better visualization.

A canonical way of discretizing the Poincaré disk $\mathbb{D}^2$ is through *regular hyperbolic tilings* [76, 77]. These are gapless fillings of hyperbolic space with regular polygons. General regular tilings of two-dimensional spaces are characterized by their *Schläfli symbol* $\{p, q\}$, denoting a tiling where $q$ regular $p$-gons meet at each vertex. The Schläfli symbol further contains information about the curvature of the space that is being tessellated. Spherical tilings obey $(p - 2)(q - 2) < 4$, while Euclidean ones obey $(p - 2)(q - 2) = 4$, e.g. square or hexagonal tilings. Tilings of hyperbolic space are obtained whenever $(p - 2)(q - 2) > 4$, thus providing an infinite number of different solutions. Some examples of hyperbolic $\{p, q\}$ tilings are shown in Fig. 2.

Given that hyperbolic space introduces a natural length scale via its radius of curvature, the size of the tiles is fixed with respect to this length. Polygon edges are thus geodesic segments of fixed length and internal angles are given by $2\pi/q$. Moreover, the distance of the center of a polygon to any of its vertices, the so-called *circumradius* $r_0$, is also fixed to be

$$r_0(p, q) = \sqrt{\frac{\cos\left(\frac{\pi}{p} + \frac{\pi}{q}\right)}{\cos\left(\frac{\pi}{p} - \frac{\pi}{q}\right)}} . \tag{6}$$

Using hyperbolic trigonometry [77], we can find the other two characteristic lengths of a polygon, namely the minimal distance $\rho_0$ from the center to an edge, and the length $s_0$ of an edge,

$$\rho_0(p, q) = \operatorname{arctanh}\left(\cos\left(\frac{\pi}{p}\right) \tanh(r_0(p, q))\right) ,$$
$$s_0(p, q) = 2 \operatorname{arcsinh}\left(\sin\left(\frac{\pi}{p}\right) \sinh(r_0(p, q))\right) . \tag{7}$$

These lengths are expressed in terms of coordinates on the Poincaré unit disk $\mathbb{D}^2$, which has a fixed length scale given by its (unit) radius. Their corresponding hyperbolic length in units of the AdS radius $R$ is obtained from the distance function $d(\cdot, \cdot)$ (4) associated to the Poincaré metric (3). We thus define the following hyperbolic lengths,

$$d(r_0(p, q), 0) \equiv r(p, q, R) \equiv r ,$$
$$d(\rho_0(p, q), 0) \equiv \rho(p, q, R) \equiv \rho ,$$
$$d(s_0(p, q), 0) \equiv s(p, q, R) \equiv s , \tag{8}$$

all of which are visualized in Fig. 2.

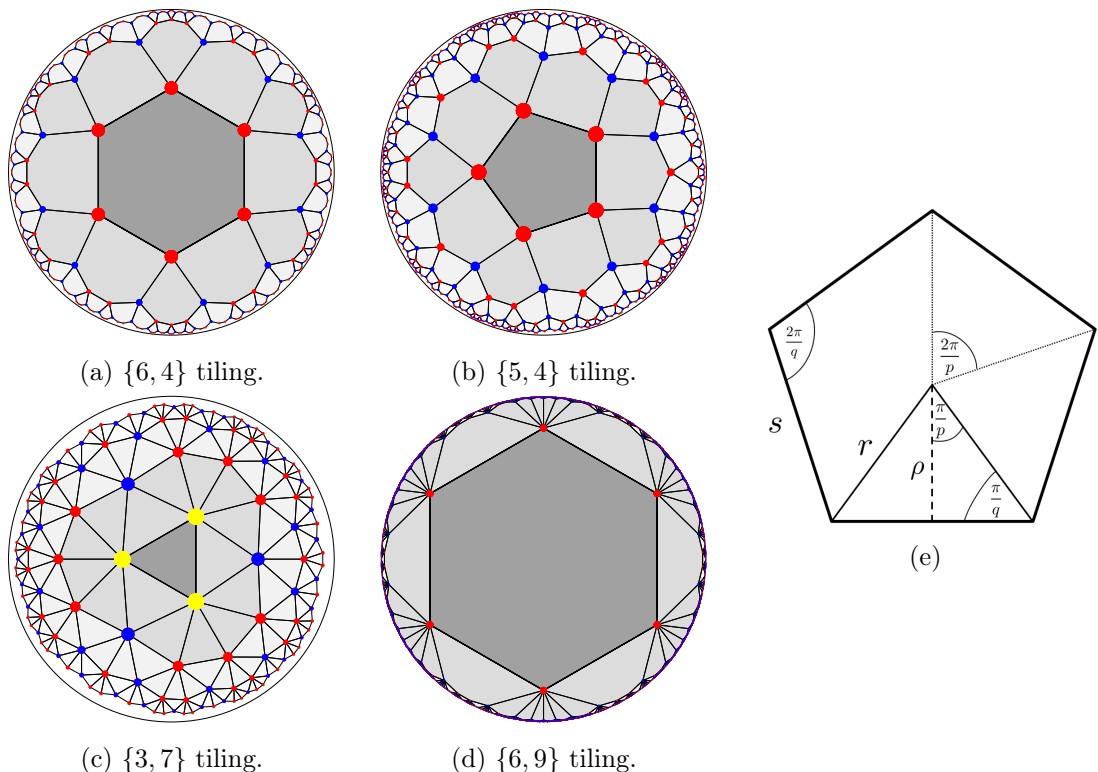

Figure 2: a)-d): Examples of $\{p, q\}$ regular hyperbolic tilings for all parity combinations of the Schläfli parameters. The red, blue and yellow dots denote vertices of type a, b and ⊛, respectively, according to the inflation rules introduced in Sec. 2.2. The different shades of gray depict successive layers of this inflation procedure. e) Characteristic lengths and angles of a single polygon in $\{5, q\}$ tiling, as defined in (8). Due to the intrinsic length scale given by the curvature radius, these lengths are fixed as a function of $p$ and $q$ and cannot be varied.

## 2.2 Hyperbolic tilings through inflation rules

Geometrically, all hyperbolic tilings can be constructed from a central triangle by mirroring along geodesic edges [76]. This is useful, e.g., for practical implementations that require the exact positions of the vertices. We are interested in a different construction method, which generates the bulk layer by layer and highlights the aperiodic, fractal-like structure arising on the boundary.

**Aperiodic sequences**

We consider first the properties of general aperiodic sequences before restricting to those associated to $\{p, q\}$ tilings. An aperiodic sequence of letters is generated by repeated application of a substitution or *inflation rule* to a starting set of letters denoted as *seed word* [78]. For binary sequences, meaning that only two different letters $a$ and $b$ appear, the general inflation rule reads

$$\sigma : \begin{cases} a \mapsto w_a(a, b) \,, \\ b \mapsto w_b(a, b) \,, \end{cases} \tag{9}$$

where $w_a(a, b)$ and $w_b(a, b)$ are words made up of $a$ and $b$. An infinite aperiodic sequence is generated by the iterated application of the rule (9) an infinite number of times. Inflation

rules that can be mapped to each other via word conjugation, i.e. $w_a(a,b) \mapsto w'_a(a,b) = u w_a u^{-1}$ and $w_b(a,b) \mapsto w'_b(a,b) = u w_b u^{-1}$ with a finite word $u = u(a,b)$, are said to be equivalent and lead to the same infinite aperiodic sequence [78]. Note that two inflation rules such that one is obtained by applying the other an integer number of times are also equivalent in this sense. Given two inflation (or later, deflation) rules $\sigma$ and $\sigma'$ we denote their equivalence by $\sigma \sim \sigma'$. The properties of the infinite sequence generated by the inflation rule (9) are encoded in the *substitution matrix* defined as

$$M_\sigma = \begin{pmatrix} \#_a(w_a) & \#_a(w_b) \\ \#_b(w_a) & \#_b(w_b) \end{pmatrix}, \tag{10}$$

where $\#_i(w_j)$ is the number of letters of $i$-th type into the word $w_j$, with $i, j = a, b$. Since inflation matrices are real and non-negative by construction, the Perron-Frobenius theorem guarantees the uniqueness of their largest eigenvalue $\lambda_+$. This gives the asymptotic scaling factor of the sequence length after a large number of iterations. The corresponding statistically normalized right eigenvector (Perron-Frobenius eigenvector) $\mathbf{v}_+ = (p_a, p_b)^t$ determines the frequencies of the letters $a$ and $b$ in the asymptotic sequence. The left eigenvector $\mathbf{u}_+ = (l_a, l_b)^t$ associated to $\lambda_+$ is normalized in such a way that $\mathbf{u}_+ \cdot \mathbf{v}_+ = 1$. This normalization naturally introduces two length scales, $l_a$ and $l_b$, associated to the letters $a$ and $b$ respectively. It is worth stressing that these interpretations for the eigenvalues and eigenvectors of the inflation matrix presuppose a large number of inflation steps and are only valid in this asymptotic limit. Only in this scenario is a large finite sequence a good representative of the aperiodic structure.

**Inflation rules for $\{p, q\}$ tilings**

The properties of aperiodic sequences described above are valid for general inflation rules. To make contact to the hyperbolic tilings that discretize a constant time-slice of $\mathrm{AdS}_{2+1}$, we restrict ourselves to those inflation rules that generate such tilings. The construction of a $\{p, q\}$ hyperbolic tiling from an inflation rule is as follows. Our starting point is a polygon centered around the origin of the Poincaré disk. This will be the $0^{th}$-layer of the tiling and the vertices represent the seed word. Subsequent layers of tiles are introduced concentrically around the central polygon. We adapt the notation introduced in [52] and define two types of vertices: $a$ and $b$. Given a fixed layer of the tiling, there will be vertices that are connected to the previous layer and those who are not. The latter will have a lower effective number of neighbors and we denote it by the letter $a$. Those vertices connected to the previous layer have a larger effective coordination number (within this fixed layer) and will be denoted by the letter $b$. This exact number of effective neighbors depends on the specific $\{p, q\}$ tiling, but the association with $a$ and $b$ letters can always be done. The case $p = 3$ requires the introduction of an auxiliary vertex, denoted by the symbol $\circledast$, which only appears in the central tile before it disappears from the tiling. With this notation, the seed word providing the starting point for inflation is given by the sequence $aa \ldots a$ of length $p$ for $p > 3$ and the sequence $\circledast \circledast \circledast$ for $p = 3$. The $n^{\text{th}}$ layer of the tiling is then constructed by applying a tiling-dependent inflation rule $\sigma_{\{p,q\}}$ to the $(n-1)^{\text{th}}$ layer, as shown in Fig. 2 for two initial inflation steps. Different types of vertices are color-coded. The general inflation rules for $p = 3$ and $q \geqslant 7$ are given by [40, 52]

$$\sigma_{\{3,q\}} = \begin{cases} \circledast \mapsto a^{q-4}b, \\ a \mapsto a^{q-5}b, \\ b \mapsto a^{q-6}b. \end{cases} \tag{11}$$

Since the auxiliary vertex ⊛ disappears from the sequence after the first inflation step, (11) can be effectively treated as a binary inflation rule. For the more general $p > 3$ case, the inflation rule is

$$
\sigma_{\{p,q\}} = \begin{cases} a \mapsto a^{p-4}b(a^{p-3}b)^{q-3}\,, \\ b \mapsto a^{p-4}b(a^{p-3}b)^{q-4}\,. \end{cases} \tag{12}
$$

The $q = 3$ case is somewhat pathological, in the sense that it requires three letters as well as the introduction of a deletion rule that removes a particular letter each step. This case is considered separately in Appendix B where we provide an alternative approach. For the general case, the substitution matrix (10) reads

$$
M_{\{p,q\}} = \begin{cases} \begin{pmatrix} (p-3)(q-3) + p - 4 & (p-3)(q-4) + p - 4 \\ q - 2 & q - 3 \end{pmatrix}, & p > 3, \\ \begin{pmatrix} q - 5 & q - 6 \\ 1 & 1 \end{pmatrix}, & p = 3\,, \end{cases} \tag{13}
$$

and its largest eigenvalue is

$$
\lambda_+(p,q) = \begin{cases} \frac{1}{2}\left(pq - 2p - 2q + 2 + \sqrt{(pq - 2p - 2q + 2)^2 - 4}\right), & p > 3\,, \\ \frac{1}{2}\left(q - 4 + \sqrt{q^2 - 8q + 12}\right), & p = 3\,. \end{cases} \tag{14}
$$

Notice that $\lambda_+(p,q) > 1$ for all $p$ and $q$. The corresponding properly normalized right and left eigenvectors are

$$
\mathbf{v}_+ = \begin{pmatrix} \frac{\omega(p,q)}{2(p-2)} - \frac{q}{2} + 2 \\ -\frac{\omega(p,q)}{2(p-2)} + \frac{q}{2} - 1 \end{pmatrix}, \tag{15}
$$

$$
\mathbf{u}_+ = \left(\frac{2}{-p(q-2) + \omega(p,q) + 2q}, \frac{-p(q-2) + \omega(p,q) + 4q - 8}{(q-2)\left(-p(q-2) + \omega(p,q) + 2q\right)}\right),
$$

where we introduced the short-hand notation $\omega(p,q) \equiv \sqrt{(p-2)(q-2)(p(q-2) - 2q)}$ for clarity. It is a straightforward computation to check that the normalization condition $\mathbf{u}_+ \cdot \mathbf{v}_+ = 1$ holds.

Successive application of these inflation rules generates the hyperbolic tiling one layer at a time. Infinite inflation steps tessellate the whole Poincaré disk, while truncation of the tiling after some finite number $n$ of inflation steps introduces a radial cutoff. The letter sequence after $n$ inflation steps characterizes uniquely the vertices on the boundary of the tiling. It is worth stressing that, while the inflation of the seed word produces a truly aperiodic sequence, the boundary of the hyperbolic tiling still enjoys a $\mathbb{Z}_p$ rotational symmetry with respect to the central tile. After a large number $n \to \infty$ of inflation steps, we can restrict ourselves to a $p$-th of the tiling boundary and still view large sub-sequences of it as good representatives of the aperiodic structure.

## 2.3 Entanglement entropy in discretized AdS$_3$: a toy model

We proceed to explain a geometric derivation of entanglement entropy on hyperbolic tilings. We consider the vacuum state of AdS$_{2+1}$ with a UV cutoff radius $\tilde{\rho}$, dual to the ground state of a holographic two-dimensional CFT, with central charge $c$, defined on a circle with

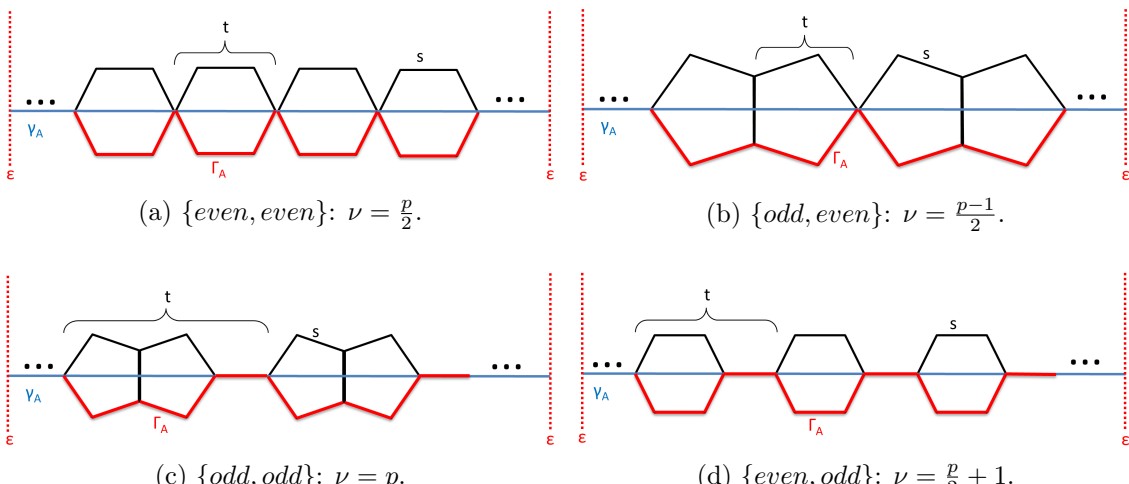

(a) $\{even, even\}$: $\nu = \frac{p}{2}$.  (b) $\{odd, even\}$: $\nu = \frac{p-1}{2}$.

(c) $\{odd, odd\}$: $\nu = p$.  (d) $\{even, odd\}$: $\nu = \frac{p}{2} + 1$.

Figure 3: Hyperbolic patterns encompassing repeated segments of length $t(p, q)$ of a continuous geodesic $\gamma_A$ (blue line) for different parities of $\{p, q\}$. We take $\gamma_A$ to be a diametric geodesic of $\mathbb{D}^2$ here for clarity. Suitable choice of radial cutoff $\epsilon$ guarantees that the endpoints of $\gamma_A$ coincide with vertices of the tiling. A single pattern contributes a number $\nu$ of edges to the discrete path $\Gamma_A$ (red line) that approximates $\gamma_A$.

circumference $\mathcal{L}$. We assume that the RT formula (1) for the entanglement entropy of a boundary region $A$ of length $\ell$ holds after the hyperbolic tiling is introduced. In AdS$_{2+1}$ the minimal area hypersurface $\gamma_A$ involved in (1) is a geodesic anchored at the endpoints of the boundary subsystem $A$. On the tiling, the length of the continuous geodesic $\gamma_A$ can be approximated by that of a discrete path $\Gamma_A$ consisting purely of polygon edges and that is anchored on the same boundary region $A$. The length $|\Gamma_A|$ then equals its total number of edges $\mathcal{N}$ times the geodesic edge length $s$. The discretized version of the RT formula then reads,

$$S_A = \frac{|\Gamma_A|}{4G_N} = \frac{\mathcal{N}s}{4G_N} \,. \tag{16}$$

In the following, we explain how to derive explicit expressions for $\mathcal{N}$ for different parities of the Schläfli parameters $p$ and $q$.

The starting point of the construction is the proper orientation of the tiling. For any continuous geodesic $\gamma_A$ of $\mathbb{D}^2$, we orient the tiling in the most symmetric way, which is characterized by the continuous geodesic cutting subsequent polygons in a predictable manner. This can always be achieved by first transforming the geodesic to a diametric geodesic via a isometric transformation and then fixing the tiling orientation. In particular, this introduces the notion of a *pattern*, which is a local geometric substructure of the tiling that repeats exactly along the continuous geodesic. We visualize the patterns for different $\{p, q\}$ tilings in Fig. 3. The anchoring points of $\Gamma_A$ coincide with those of $\gamma_A$. Indeed, proper definition of an entangling region on the boundary theory requires the geodesic to end on a vertex, as will be explained in Sec. 3.

After fixing the tiling's orientation, the continuous geodesic $\gamma_A$ will consist of a number $n$ of patterns, each of which encompasses the geodesic length $t(p, q)$, i.e. $|\gamma_A|_c = n \, t(p, q)$. This length can be expressed in terms of the characteristic lengths of a polygon via (8). The discrete path $\Gamma_A$ consists purely of the edges of the polygons through which $\gamma_A$ runs, cf. Fig. 3. Given a pattern, we assign to it a number $\nu(p, q)$ counting the number of edges it adds to the discrete approximation of the geodesic. Thus, we have $\mathcal{N} = n\nu(p, q)$ and

(16) turns into

$$S_A = \frac{n\,s(p,q)\,\nu(p,q)}{4G_N}\,, \tag{17}$$

where $s(p,q)$ is defined in (8). In principle, the number $n$ cannot be retrieved from information about the tiling. However, it can be removed from (17) by inserting the continuous length of the geodesic $\gamma_A$. It is known [6] that integration of the metric (5) along general solutions to the geodesic equation up to a radial cutoff $z = \epsilon \ll 1$ yields a closed expression for the length of the continuous geodesic

$$|\gamma_A| = 2R\ln\left(\frac{\ell}{\epsilon}\right) + \mathcal{O}(\epsilon^2)\,, \tag{18}$$

where $\ell$ is the length of the entangling region $A$ defined by the set of boundary points $x \in (-\frac{\ell}{2}, \frac{\ell}{2})$ in the Poincaré coordinates of (5). Inserting (18) into (17) yields

$$S_A = \frac{2R\,s(p,q)\,\nu(p,q)}{4G_N t(p,q)}\ln\left(\frac{\ell}{\epsilon}\right)\,. \tag{19}$$

Finally, let us emphasize that the above analysis does not yet take into account the subtle relation between lengths on the bulk and lengths on the boundary, which arises from the fractal structure of the tiling's boundary. This introduces a scaling exponent, the *fractal dimension* $d = \ln(\lambda_+)/t$, relating the number of sites $L$ in the entangling region to its length $\ell$ via $\ln(L) = d\ln(\ell/\epsilon)$. A derivation of the fractal dimension for $\{p,q\}$ tilings is provided in Appendix A. Taking this into account, we find an expression for the tiling-dependent entanglement entropy

$$S_A = \frac{2R}{4G_N}\frac{s(p,q)\,\nu(p,q)}{t(p,q)}\frac{t(p,q)}{\ln\lambda_+(p,q)}\ln(L) \equiv \frac{c_{\text{eff,bulk}}(p,q)}{3}\ln(L)\,, \tag{20}$$

with the tiling-dependent effective central charge

$$c_{\text{eff,bulk}}(p,q) = \frac{c\,s(p,q)\,\nu(p,q)}{\ln\lambda_+(p,q)}\,, \tag{21}$$

characterizing the logarithmic growth of the entanglement entropy with the subsystem size. We have made use of the Brown-Henneaux formula $c = \frac{3R}{2G_N}$ to introduce the central charge of the CFT in consideration. The exact value of $c$ is left undefined.

A few comments on (21) are in order. First, we emphasize that the effective central charge is defined as the coefficient of the logarithmic growth of the entropy, but it does not carry the standard interpretation of measuring the number of degrees of freedom in the theory. Nevertheless, even in this simplified setup, the prefactor does depend on $\{p,q\}$, which are the only parameters characterizing the discretization of the bulk. We are thus able to see a non-trivial dependence of the entanglement entropy on the details of the discretization considered. Second, a similar analysis has been presented in [52], where maximal effective central charges for perfect tensor networks on hyperbolic tilings have been derived. Let us emphasize that the construction in [52] makes explicit use of a TN structure with perfect tensors of fixed bond dimension $\chi$. In particular, this implies a dependence of the Brown-Henneaux formula on the Schläfli parameters, which we do not assume in our construction. In contrast, our analysis provides a much simpler, geometrical derivation of the consequences of the discretization on the logarithmic scaling of the boundary system size. A more detailed comparison of the resulting effective central charges for different tilings derived in our setup with those of [52] is provided in Sec. 6.

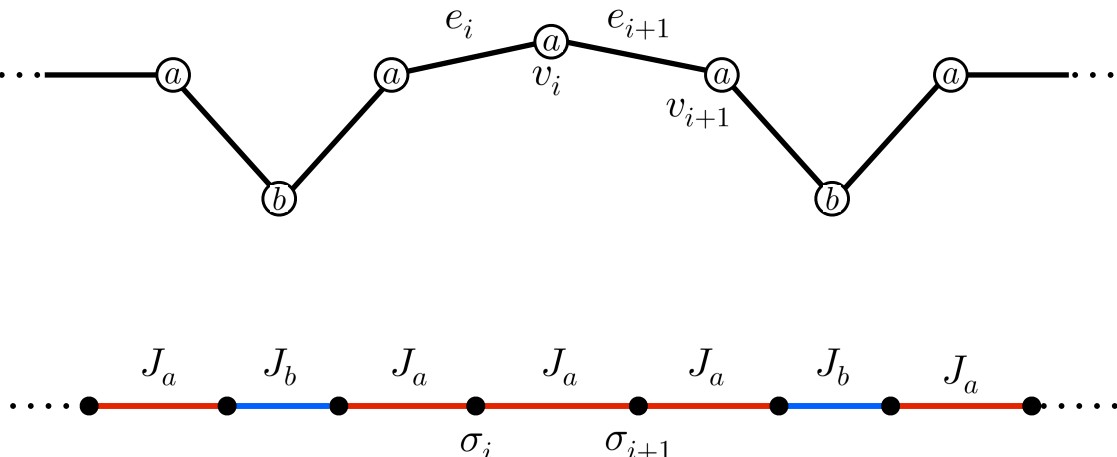

Figure 4: Construction of an aperiodic spin chain (bottom) based on the sequence of letters that the inflation procedure for a $\{p, q\}$ tiling of the Poincaré disk (cf. Sec. 2.2) induces on the boundary (top). The letter sequence on the tiling's boundary is associated to the coupling distribution in the corresponding spin chain. Different colors depict bonds with different couplings $J_k$ with $k = a, b$.

## 3    Aperiodic quantum spin chains

Based on the analysis of Sec. 2 and in view of establishing a bulk-boundary correspondence on regular hyperbolic tilings, we consider aperiodic quantum spin chains to be a promising candidate for a boundary theory. In order to study the critical properties of these models, we discuss the strong disorder renormalization group approach for aperiodic spin chains [57, 58]. We subsequently apply it to our example of interest in this class, namely XXZ chains with aperiodicities induced by the inflation rule $\sigma_{\{p,q\}}$. Finally, we discuss the consequences of aperiodicity in this model and characterize its relevance with respect to the homogeneous model.

### 3.1    Aperiodic spin chains on the boundary of hyperbolic tilings

Consider a regular $\{p, q\}$ hyperbolic tiling on the Poincaré disk obtained after a large number of inflation steps, as described in Sec. 2.2, and the sets $\{e_i\}_{i=1}^N$ and $\{v_i\}_{i=1}^N$, respectively containing the edges and the vertices on the tiling's boundary. Each $v_i$ is endowed with a letter $x_i \in \{a, b\}$ such that the sequence $\{x_i\}_{i=1}^N$ is determined by the inflation rule $\sigma_{\{p,q\}}$, as depicted in Fig. 4. Given that the full tiling enjoys a $\mathbb{Z}_p$ rotational symmetry, we restrict to one $p$-th of the boundary, where the sequence of letters is a proper aperiodic sequence generated by $\sigma_{\{p,q\}}$, in the sense that there are no sub-sequences repeated with a fixed periodicity. We assume that the number of inflation steps through which the boundary has been generated is large such that this sequence can be well approximated by an infinite aperiodic sequence. This leads us to consider infinitely many edges $\{e_i\}_{i\in\mathbb{Z}}$ and vertices $\{v_i\}_{i\in\mathbb{Z}}$, the latter ones in one-to-one correspondence with letters following an infinite aperiodic sequence $\{x_i\}_{i\in\mathbb{Z}}$. As explained in Sec. 2.2, the properties these sequences are encoded in the substitution matrix (10), its largest eigenvalue and the corresponding right and left eigenvectors.

At any inflation step, the boundary of the tiling is characterized by a discrete set of

vertices and edges. Thus, if a correspondence between a discrete theory defined on the tiling and a model on its boundary exists, we expect that the latter can be described by a quantum chain, e.g. a spin chain. We also expect that the degrees of freedom along the spin chain follow a pattern determined by the same sequence $\{x_i\}_{i\in\mathbb{Z}}$ which characterizes the boundary of the tiling. The procedure we follow for constructing such a suitable spin chain is pictorially represented in Fig. 4. We associate spin degrees of freedom, generically denoted by $\sigma_i$, to each edge $e_i$ and a bond to each vertex $v_i$ of the tiling's boundary. Moreover, given a vertex $v_i$, we assign to the corresponding bond the coupling $J_a$ if $x_i = a$ and the coupling $J_b$ if $x_i = b$. Models constructed in this way are examples of aperiodic quantum chains, a well known class of systems in the literature of condensed matter theory [45–50]. The precise nature of the couplings depends on the specific spin model one considers. In this work, we restrict ourselves to nearest-neighbor chains and to the case where the couplings $J_a$ and $J_b$ are hopping parameters. In particular, the case $J_a = J_b \equiv J$ recovers a homogeneous spin chain. For concreteness, one can imagine that this underlying homogeneous model is in a gapless regime, which in the continuum limit is described by a CFT with central charge $c$. An important question that can be raised is whether the presence of aperidiocity modifies the critical properties of the homogeneous model. An aperiodic modulation is called *relevant* if the critical behavior changes, being governed by a new aperiodicity-induced fixed point. If instead the critical properties are unchanged after the introduction of the aperiodicity, the modulation is called *irrelevant*. Finally, we denote a modulation as *marginal* when the criticality of the system (or, more concretely, its critical exponents) develops a continuous dependence on the values of $J_a$ and $J_b$.

In our construction the nature of the spin variables defined on each site of the chain, as well as of the explicit form of the Hamiltonian, is not fixed *a priori*, but it can be chosen according to the features we require for the boundary theory. These choices do not influence in any way the aperiodic modulation we impose on the boundary chain, which is the only feature determined by the pair $\{p,q\}$ associated to the bulk tessellation. For instance, as we are going to specify in the next subsection, one can require the boundary model to be gapless, imposing constraints on the type of spin degrees of freedom and on the Hamiltonian describing their behavior. The most common choice is considering $SU(2)$ spins, where the spin variables are operators satisfying the Lie algebra associated to the group $SU(2)$. Moreover, within this choice, one can further choose the irreducible representation of $SU(2)$, which is associated to a spin quantum number that can assume either integer or half integer numbers. As we will discuss later, for some specific choices of Hamiltonian, the nature of the spin quantum number can strongly influence the physical properties of the model, independently of the presence of aperiodicity [79–81]. In this work we will focus on the spin-1/2 case, for which the properties of the aperiodic spin chains are well understood, cf. the discussion in Sec. 3.2.
Furthermore, we find it worth mentioning that generalizations to $SU(N)$-spin aperiodic chains as boundary theories are feasible in principle. In that case, the spin variables at each site are represented by the $N^2 - 1$ generators of the $SU(N)$ group [82–87]. Although treating this class of systems can be very difficult, considering them as possible boundary theories opens very interesting scenarios which we will briefly mention in Sec. 7.

## 3.2 Aperiodic XXZ spin chain and strong disorder renormalization group

Thus far, we have discussed the properties for generic spin chains with couplings modulated by two-letter inflation rules (9). In this subsection, we choose a specific aperiodic chain in order to study how the critical properties of the underlying homogeneous model are modified by introducing a modulation generated by $\sigma_{\{p,q\}}$.

We introduce the aperiodic XXZ spin chain, governed by the following Hamiltonian

$$H = \sum_{i \in \mathbb{Z}} J_i \left[ \sigma_i^{(x)} \sigma_{i+1}^{(x)} + \sigma_i^{(y)} \sigma_{i+1}^{(y)} + \Delta_0 \sigma_i^{(z)} \sigma_{i+1}^{(z)}, \right] \tag{22}$$

where $\sigma_i^{(\alpha)}$ with $\alpha = x, y, z$ are the Pauli matrices localized on the $i$-th site of the chain, $J_i = J_a$ if the bond between the $i$-th and the $i + 1$-th site is of the type $a$ and $J_i = J_b$ otherwise. We define the aperiodic XXZ chain (22) with a modulation of the hopping parameters $J_i$ generated by the inflation rule $\sigma_{\{p,q\}}$ as the boundary theory of the $\{p, q\}$ tiling of a Poincaré disk. For the remainder of this work, we focus on the properties of the ground state of this model and their dependence on the Schläfli parameters.

Since the Hamiltonian can be rescaled by a constant factor without changing the properties of the model, from (22) it is straightforward to see that the only physical parameters are $r \equiv J_a/J_b$ and $\Delta_0$. The parameter $\Delta_0$ is sometimes called the *anisotropy parameter* and we restrict ourselves to the regime $0 \leqslant \Delta_0 \leqslant 1$ in the following. This is because, in this range of values for $\Delta_0$, the underlying homogeneous model (homogeneous XXZ chain) is gapless and is described by a compactified free boson CFT ($c = 1$) in the continuum limit, with the compactification radius determined by $\Delta_0$ [88]. One of our motivations for considering the XXZ chain is that it can be mapped into a theory of interacting fermions *via* the Jordan-Wigner (JW) transformation, which relates spins to spinless fermions. In particular, when $\Delta_0 = 0$, (22) reduces to the Hamiltonian of the aperiodic XX model, which is a free model in the sense that it is mapped by the JW transformation into a chain of free fermions. An exact approach for determining the relevance of the aperiodic modulations in XX chains has been developed in [49]. Instead, when $\Delta_0 = 1$, we have the so-called aperiodic XXX spin chain.

We require the aperiodic spin chain that we define on the tiling's boundary to be critical. This is motivated by the standard continuum AdS/CFT, where a conformal field theory, which provides a good description for gapless systems, is defined on the boundary of the AdS spacetime. Thus, it is in our interest to verify if the criticality of the homogeneous XXZ model is maintained in the presence of aperiodicity. The effect of aperiodic modulations on the critical properties of the XXZ chain has been first studied in [57, 58], where the strong disorder renormalization group (SDRG) developed in [89, 90] for systems with random disorder has been adapted to the case of aperiodic modulations. In these works it was argued that, in presence of binary aperiodicities with equal fractions of letters $a$ (or $b$) at even and odd sites, the XXZ spin chain with $0 \leqslant \Delta_0 \leqslant 1$ is still in a gapless regime. By this we mean that if we were to consider only the $a$ letters in a sequence, these would be uniformly distributed among even and odd numbered sites of the full chain. A criterion to determine whether a given aperiodic sequence does fulfill this property is provided in [49]. Following that argument, we have checked that the $\{p, q\}$ modulations (11) and (12) considered here satisfy these requirements and therefore the theory we have chosen to define on the boundary of a $\{p, q\}$ tiling is critical in the given parameter regime. Interestingly, note that when considering integer-spin representations of $SU(2)$ instead of spin-$1/2$ degrees of freedom along an aperiodic chain, criticality is no longer guaranteed. This happens, for instance, in spin-1 aperiodic XXX chains, which are found to have a finite gap in the spectrum, differently from its spin-$1/2$ counterpart [81]. Thus, we are motivated to focus on half-integer spins aperiodic chains and, more specifically, spins-$1/2$ chains, where the criticality condition is well understood [49, 57, 58].

We now proceed to briefly review the SDRG method for the aperiodic XXZ chain [57, 58]. Consider a subsystem of the chain made up with $n + 2$ spins connected by the sequences of nearest neighbour couplings $J_1, J_0, J_0, ..., J_0, J_r$ and $\Delta_1, \Delta_0, \Delta_0, ..., \Delta_0, \Delta_r$. We assume that $J_0 \gg J_{l,r}$ and $\Delta_0 \gg \Delta_{l,r}$ and therefore the bonds with coupling $J_0$ are

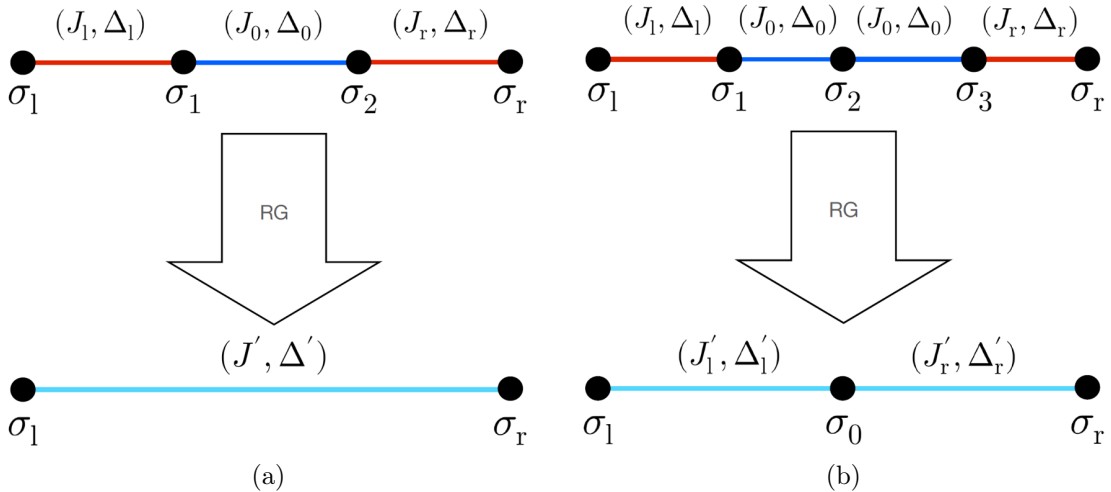

Figure 5: Strong disorder renormalization step involving a block of 2 (a) and 3 (b) spins connected by strong bonds. Strong bonds are depicted in blue, while all other weaker bonds are shown in red. Whether the bonds in the renormalized chain (depicted in cyan) are strong or not can only be determined a posteriori, by comparing the renormalized couplings after the RG transformation has been applied to the whole chain.

called strong bonds. The local Hamiltonian for the $n$ internal spins reads

$$H_n = J_0 \sum_{i=1}^{n} \left[ \sigma_i^{(x)}\sigma_{i+1}^{(x)} + \sigma_i^{(y)}\sigma_{i+1}^{(y)} + \Delta_0 \sigma_i^{(z)}\sigma_{i+1}^{(z)} \right] , \tag{23}$$

while the coupling of the $n$ spins with the the two external ones, denoted by $\sigma_{\mathrm{l,r}}^{(\alpha)}$ with $\alpha = x, y, z$, is

$$\delta H = J_\mathrm{l} \left[ \sigma_\mathrm{l}^{(x)}\sigma_1^{(x)} + \sigma_\mathrm{l}^{(y)}\sigma_1^{(y)} + \Delta_\mathrm{l}\sigma_\mathrm{l}^{(z)}\sigma_1^{(z)} \right] + J_\mathrm{r} \left[ \sigma_n^{(x)}\sigma_\mathrm{r}^{(x)} + \sigma_n^{(y)}\sigma_\mathrm{r}^{(y)} + \Delta_\mathrm{r}\sigma_n^{(z)}\sigma_\mathrm{r}^{(z)} \right] . \tag{24}$$

The idea behind the SDRG is that at low temperature (below the smallest gap of $H_n$), $\delta H$ can be regarded as a small perturbation of $H_n$. Under this assumption, the $n$ internal spins can be decimated out from the system, since they couple into their ground state which gives a negligible contribution to the thermodynamic properties. The decimation of the internal spins induces effective couplings between the two external ones, whose magnitudes can be estimated through the second order perturbation theory. In order to provide the explicit expressions, let us distinguish between the cases where $n$ is even and $n$ is odd. When $n$ is even, the ground state of $H_n$ is the singlet

$$|T\rangle = \sum_{\{m_i = \pm\}} |m_1\rangle \ldots |m_n\rangle \, T_{m_1 \ldots m_n} , \tag{25}$$

where we have defined the basis of the local Hilbert space such that $\sigma^{(z)}|\pm\rangle = \pm|\pm\rangle$. The decimation process of the internal spins induces an effective coupling between $\sigma_\mathrm{l}$ and $\sigma_\mathrm{r}$ described by the Hamiltonian

$$H'_{\mathrm{even}} = J' \left[ \sigma_\mathrm{l}^{(x)}\sigma_\mathrm{r}^{(x)} + \sigma_\mathrm{l}^{(y)}\sigma_\mathrm{r}^{(y)} + \Delta'\sigma_\mathrm{l}^{(z)}\sigma_\mathrm{r}^{(z)} , \right] \tag{26}$$

where the parameters $J'$ and $\Delta'$ are given by [57, 91]

$$J' = \gamma_n(\Delta_0)\frac{J_\mathrm{l}J_\mathrm{r}}{J_0} , \qquad \Delta' = \delta_n(\Delta_0)\Delta_\mathrm{l}\Delta_\mathrm{r} , \tag{27}$$

with $\gamma_n$ and $\delta_n$ functions of $\Delta_0$ and $n$ such that $|\delta_n(\Delta_0)| < 1$ when $0 \leqslant \Delta_0 < 1$ and $\delta_n(1) = 1$. The decimation of a block of $n = 2$ spins is shown pictorially in Fig. 5 a.

On the other hand, when $n$ is odd, the ground states of $H_n$ is a generic linear combination of two degenerate states in a doublet

$$|T^{m_0}\rangle = \sum_{\{m_{i\neq 0}=\pm\}} |m_1\rangle \dots |m_n\rangle \, T^{m_0}_{m_1\dots m_n}\,, \tag{28}$$

where the sum does not run over $m_0 = \pm$, regarded here as a free index labeling each of the two ground states. Notice that the coefficients $T^{m_0}_{m_1\dots m_n}$ have not been specified yet. In order to fix them, in the spirit of SDRG [57, 91], we consider the total spin of the block within the subspace spanned by the degenerate ground states $|T^{\pm}\rangle$. More precisely, we impose that the total spin operator $M_n^{(\alpha)} \equiv \sum_{i=1}^n \sigma_i^{(\alpha)}$, for $\alpha = x, y, z$, of an $n$-spin block is equal, up to normalization factors, to a single effective spin $\sigma_0^{(\alpha)}$, namely

$$\left\langle T^{m_0} \left| M_n^{(\alpha)} \right| T^{m'_0} \right\rangle = \eta_n^{(\alpha)}(\Delta_0) \, \langle m_0| \sigma_0^{(\alpha)} |m'_0\rangle\,, \quad \alpha = x, y, z. \tag{29}$$

where $\eta_n^{(z)}(\Delta_0) = 1$ for generic anisotropies $\Delta_0$. In the special case of $\Delta_0 = 1$, we have $\eta^{(\alpha)}(1) = 1$. Eq. (29) indicates that, along the SDRG, we can replace the block in its ground state by an effective spin $\sigma_0^{(\alpha)}$ coupled to $\sigma_{l,r}^{(\alpha)}$ through the Hamiltonian

$$H'_{\mathrm{odd}} = J'_l \left[ \sigma_l^{(x)}\sigma_0^{(x)} + \sigma_l^{(y)}\sigma_0^{(y)} + \Delta'_l \sigma_l^{(z)}\sigma_0^{(z)} \right] + J'_r \left[ \sigma_0^{(x)}\sigma_r^{(x)} + \sigma_0^{(y)}\sigma_r^{(y)} + \Delta'_r \sigma_0^{(z)}\sigma_r^{(z)} \right]\,, \tag{30}$$

where [57, 91]

$$J'_{l,r} = \gamma_n(\Delta_0)J_{l,r}\,, \qquad \Delta'_{l,r} = \delta_n(\Delta_0)\Delta_{l,r}\,. \tag{31}$$

The decimation of a block of $n = 3$ spins is shown pictorially in Fig. 5 b. Notice that analytical expressions for $\gamma_n$ and $\delta_n$ as functions of $\Delta_0$ are available only for small values $n$, while for large blocks they can be obtained numerically. For later convenience we report the results for $n = 2$, which read [89, 90]

$$\gamma_2(\Delta_0) = \frac{1}{1 + \Delta_0}\,, \qquad \delta_2(\Delta_0) = \frac{1 + \Delta_0}{2}\,. \tag{32}$$

For the following discussion, the explicit expressions of $\gamma_n$ and $\delta_n$ are not necessary; it is enough to know that $\delta_n \leqslant 1$ when $0 \leqslant \Delta_0 \leqslant 1$ and $\gamma_n(1) < 1$ [91]. We have checked these inequalities numerically for several values of $n$, as discussed in Appendix B. The results are reported in Fig. 16, where strong evidence of a decay in the values of $\gamma_n(1)$ for growing $n$ is provided.

Given the initial aperiodic sequence of couplings along the whole chain, the decimation process described above is applied simultaneously to all blocks of consecutive spins coupled by the strong bonds. This is then iterated and leads to a renormalization of the spatial distribution of the bonds along the chain. We denote a single iteration of this process as an RG step. Given the self-similarity of the aperiodic sequences, the bond distribution in the chain reaches a periodic attractor after a certain number of RG steps. If the attractor arises every $k$ RG steps, we denote it as a $k$-cycle. In the rest of the manuscript, the transformation that realizes a $k$-cycle is called *sequence-preserving transformation*. In other words, the sequence-preserving SDRG transformation is obtained by the composition of $k$ RG steps and will be denoted $\Xi$. Notice that, even if initially we do not have aperiodicities in the anisotropy parameter (see (22)), the modulation of hopping parameters induces an effective modulation on $\Delta_0$ which is renormalized to two different couplings, $\Delta_a$ and $\Delta_b$.

In order to determine the critical properties of the chain, we first employ (27) and (31) to work out the recursion relations for the effective couplings induced by $k$ RG steps. Then we apply $M$ times the recursion relation to the initial couplings, obtaining

$$r^{(M)} = F_M(r, \Delta_0), \qquad \Delta_a^{(M)} = G_M^{(a)}(r, \Delta_0), \qquad \Delta_b^{(M)} = G_M^{(b)}(r, \Delta_0), \qquad (33)$$

and finally we take $M \to \infty$ and we find the effective coupling at the so-called strong disorder fixed point

$$r^* = \lim_{M \to \infty} F_M(r, \Delta_0), \qquad \Delta_a^* = \lim_{M \to \infty} G_M^{(a)}(r, \Delta_0), \qquad \Delta_b^* = \lim_{M \to \infty} G_M^{(b)}(r, \Delta_0). \qquad (34)$$

The explicit expressions of the functions $F_M$, $G_M^{(a)}$ and $G_M^{(b)}$ depend on the specific aperiodic modulation that we consider.

Let us qualitatively discuss some important scenarios occurring regardless of the specific form of aperiodicity for the couplings. On the one hand, given that $\delta_n(\Delta_0) < 1$ when $0 \leqslant \Delta_0 < 1$, (27) and (31) imply that the anisotropy parameter flows to a XX fixed point with $\Delta_a^* = \Delta_b^* = 0$. Thus, the system enjoys the same critical behavior as the aperiodic XX chain. On the other hand, when $\Delta_0 = 1$, the anisotropy parameter does not flow and stays constantly equal to one. The behavior of the coupling ratio $r$ under SDRG is determined by looking at its flow equation, which explicitly depends on the inflation rule. Suppose that, iterating the RG steps, the coupling ratio becomes smaller and smaller reaching ultimately the fixed point $r^* = 0$. If this happens, we conclude that the aperiodicity drives the system towards a strong inhomogeneity and therefore we expect the modulation to be relevant. In contrast, when the coupling ratio $r^*$ at the fixed point is non zero and depends on the initial value $r$, the aperiodic modulation which has induced the flow is marginal. We find it worth stressing that, whenever the RG flow leads to a fixed point with $r^* = 0$ for the coupling ratio, we can argue that the SDRG method presented here becomes asymptotically exact. Indeed, recall that the relations (27) and (31) for the flows of couplings and anisotropies have been obtained using the second-order perturbation theory, with $J_0$ much larger than all the other couplings.

Based on this discussion of the general behavior of the XXZ spin chain under aperiodic modulations, let us now consider the special case of interest to us, namely aperiodic modulations induced by the inflation rules of $\{p, q\}$ tilings. The critical behavior of these models can be pictorially represented in parameter space $(r, \Delta_a, \Delta_b)$, as shown in Fig. 6. Notice that different Schläfli symbols $\{p, q\}$ give rise to different phase diagrams. When $r = 1$, the chain is homogeneous and the fixed points at various values of $\Delta_0$ (yellow dots in Fig. 6) are described by $c = 1$ CFTs. In the presence of aperiodicity ($r < 1$), for any $0 < \Delta_0 < 1$, the XXZ chain flows to an XX chain under SDRG. This is represented for an exemplary point by the light blue arrow in the bottom part of Fig. 6. We stress that, since the SDRG method is asymptotically exact at strong modulations, this picture is reliable only when $r \ll 1$. At the fixed points with $\Delta_a^* = \Delta_b^* = 0$, the critical behavior can be determined by exploiting the exact methods developed in [49] and one can show that the modulations generated by $\sigma_{\{p,q\}}$ are marginal for any pair $\{p, q\}$. In App. B this result has been verified in the range of validity of the SDRG. In the phase diagram, the marginality of the modulation on the XX chain corresponds to the line of fixed points represented in Fig. 6 as green dots along the vertical line $\Delta_a = \Delta_b = 0$.

The aperiodic XXX chain requires a separate analysis. Exploiting the techniques reviewed in this section, we show in App. B that all the aperiodicities generated by $\{p, q\}$ inflation rules lead the system to a fixed point different from the homogeneous one. Thus, all these modulations are relevant. This fact is represented in Fig. 6 by the blue dot flowing

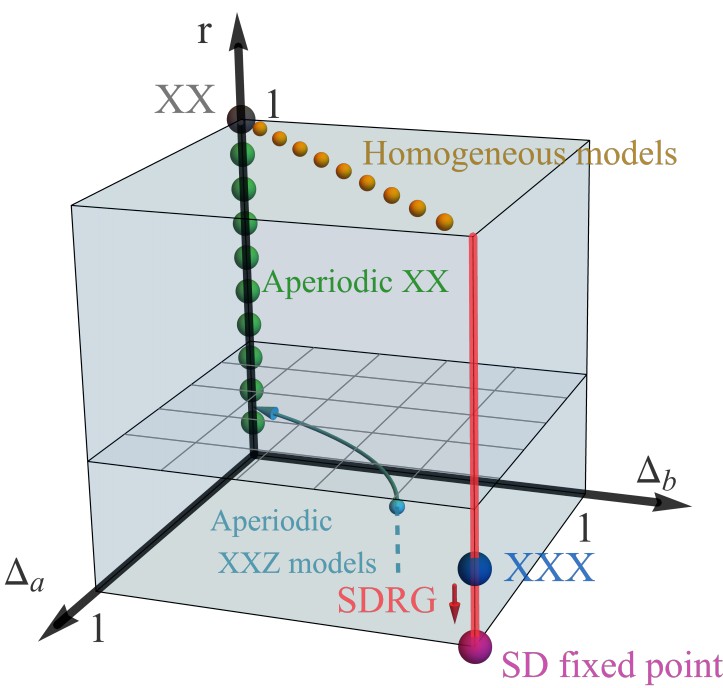

Figure 6: Phase diagram of the aperiodic XXZ chain in the critical regime. The SDRG approach is reliable in the bottom part of the parameter space only (indicated schematically by the grid plane), while the exact methods in [49] allow to establish the marginality of the modulations in aperiodic XX chains for any $0 \leqslant r < 1$. The resulting fixed points are $r$-dependent and are shown in green. The axes $\Delta_a$ and $\Delta_b$ are necessary because of the effective aperiodicity induced by the SDRG on the anisotropy $\Delta_0$, which is not modulated in the original chain (see (22)).

towards the strong disorder fixed point (purple dot) along the red vertical edge. In the following subsection we justify this statement through a detailed analysis of the modulation induced by $\sigma_{\{6,q\}}$ with $q \geqslant 4$. The generalization to any pair $\{p, q\}$ is rather involved using the standard methods presented here. In Sec. 5 we provide an equivalent graphical approach, based on a tensor network construction, that allows to prove the relevance of $\{p, q\}$ modulations in aperiodic XXX spin chain for all $p$ and $q$ in a much simpler way.

Finally, we find it useful to comment upon the relation of this aperiodic setup with the case of random quantum spin chains. The latter are often of interest for condensed matter systems since they provide a good description of inhomogeneities and defects in solid states. The SDRG procedure introduced above has originally been implemented for these random spin chains. However, the fixed points arising from the presence of random modulations are not the same as those originating from aperiodic modulations. In particular, they are characterized by different critical exponents and thus do not belong to the same universality class. For a more detailed comparison in this context, we refer to [49, 57, 91, 92].

## 3.3 Prime example: $\{6, q\}$ modulations

Throughout this work, we will use the modulation generated by the inflation rule $\sigma_{\{6,q\}}$ as a reference example. The reasons are twofold: first, in general, even values of $p$ are

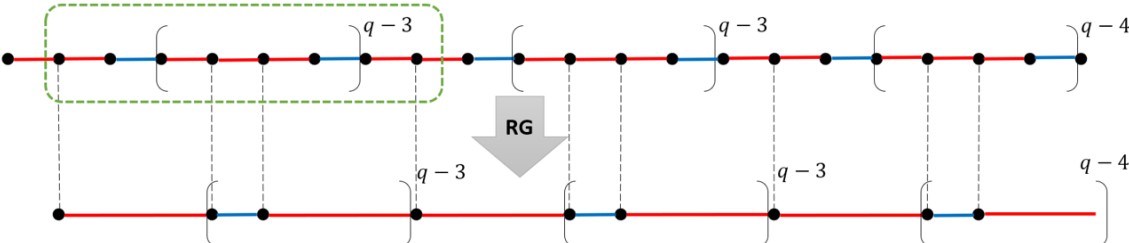

Figure 7: Top line: Typical asymptotic sequence arising from the $\{6, q\}$ inflation rule (35). Strong bonds $J_b$ are depicted in blue, weak bonds $J_a$ in red. The gray brackets imply the repetition of the letter sequence enclosed by them in analogy to usual exponentiation, i.e. $(abb)^2 \equiv abbabb$. A decimation procedure associated to the RG step (36) is performed, whereby some repeated sub-structures like the one inside the green box can be renormalized as a whole to a new sequence. Bottom line: Resulting sequence after one RG step which is different from the original sequence.

simpler to study and allow us to analyze general families of the form $\{p = 2k, q\}$ with $k \in \mathbb{N}\backslash\{1\}$, which are of interest for obtaining expressions valid for all $q$. This simplicity is explained in more detail in Sec. 5.1. Second, the $p = 6$ case has a tractable yet rich real-space renormalization structure that allows for clear visualization of the nuances that might arise in the RG procedure, such as $k$-cycle bond distribution attractors.
From (12), we can write the inflation rules for $\{6, q\}$ tilings as

$$\sigma_{\{6,q\}} = \begin{cases} a \mapsto aab(aaab)^{q-3}\,, \\ b \mapsto aab(aaab)^{q-4}\,. \end{cases} \tag{35}$$

Let us comment on some general features of the asymptotic sequence generated by (35). A typical string of letters extracted from the asymptotic sequence arising on the boundary of the $\{6, q\}$ tiling is shown on the top line of Fig. 7. As in the previous section, we denote strong bonds with blue lines and weak bonds with red ones. It is clear from the inflation rules (35) that the sequence will not have any isolated $a$ letters. Moreover, there will also never be consecutive strong bonds characterized by $bb$ (or longer) sub-sequences, which give rise to doublet ground states in the RG procedure, as explained in Sec. 3.2. Also, the $aab(aaab)^{q-4}$ sub-sequence never appears consecutively, e.g. $aab(aaab)^{q-4}aab(aaab)^{q-4}aab$ is not present in the asymptotic sequence. Finally, notice that the powers of the $(aaab)$ blocks are always either $(q-3)$ or $(q-4)$, separated by $aa$ strings. In particular, concatenations to a $(2q-7)$ block do not appear.
The decimation of spin-blocks is performed as explained in the Sec. 3.2. The effective spins of the renormalized chain will be defined on the two spins of the middle weak bond of all $aaa$ sub-sequences, as well as the middle spin of all the $aa$ sub-sequences. Since the $(aaab)^{q-3}$ and $(aaab)^{q-4}$ structures are just repetitions of these spin-block structures, their decimation can be performed independently, yielding new sub-sequences $(ba)^{q-3}$ and $(ba)^{q-4}$, respectively (cf. green dashed box in Fig. 7). Thus, the first RG transformation is independent of $q$ and reads

$$\mathrm{RG}_1^{\{6,q\}} = \begin{cases} aba \mapsto a \\ a \mapsto b \end{cases} \iff M_1^{\{6,q\}} = \begin{pmatrix} 2 & 1 \\ 1 & 0 \end{pmatrix}\,, \tag{36}$$

where $M_1^{\{6,q\}}$ is the substitution matrix (cf. (10)) associated to the inflation rule which implements the inverse of $\mathrm{RG}_1^{\{6,q\}}$. Applying the decimation formulas (27) to the local

spin-blocks of the chain, we can perform the $RG_1$ (36). The renormalized couplings and anisotropies are given by

$$J'_a = \gamma_2(\Delta_0)\frac{J_a^2}{J_b}, \quad J'_b = J_a,$$

$$\Delta'_a = \delta_2(\Delta_0)\Delta_0^2, \quad \Delta'_b = \Delta_0, \tag{37}$$

with the functions $\gamma_2(\Delta_0)$ and $\delta_2(\Delta_0)$ reported in (32). The resulting chain after one RG step (36) is shown in the bottom line of Fig. 7. However, the resulting letter sequence is manifestly not the original one. In other words, the single RG step is not equivalent, in terms of letter sequences, to a deflation step $\sigma_{\{6,q\}}^{-1}$ of (35). Instead, as we will show in the following, it turns out that the RG bond distribution attractor for $\{6, q\}$ modulation is a 2-cycle for any $q \geqslant 4$. In order to verify this, consider a longer version of the sequence resulting from the first RG step in (36), shown in Fig. 8. Consider further a second, $q$-dependent RG transformation given by

$$RG_2^{\{6,q\}} = \begin{cases} a(ba)^{q-3} \mapsto a \\ a(ba)^{q-4} \mapsto b \end{cases} \iff M_2^{\{6,q\}} = \begin{pmatrix} q-2 & q-3 \\ q-3 & q-4 \end{pmatrix}. \tag{38}$$

Exploiting again (27), this leads to the renormalized couplings and anisotropies

$$J''_a = \left(\gamma_2(\Delta'_b)\right)^{q-3}\frac{J_a'^{q-2}}{J_b'^{q-3}}, \quad J''_b = \left(\gamma_2(\Delta'_b)\right)^{q-4}\frac{J_a'^{q-3}}{J_b'^{q-4}},$$

$$\Delta''_a = \left(\delta_2(\Delta'_b)\right)^{q-3}\Delta_a'^{q-2}, \quad \Delta''_b = \left(\delta_2(\Delta'_b)\right)^{q-4}\Delta_a'^{q-3}. \tag{39}$$

The resulting sequence, shown in the bottom line of Fig. 8, is now the original one. Thus, we can relate the renormalized parameters in (39) to the original ones $(J_a, J_b, \Delta_0)$ by inserting the expressions in (37). We obtain,

$$J''_a = \left(\gamma_2(\Delta_0)\right)^{2q-5}\frac{J_a^{q-1}}{J_b^{q-2}}, \quad J''_b = \left(\gamma_2(\Delta_0)\right)^{2q-7}\frac{J_a^{q-2}}{J_b^{q-3}},$$

$$\Delta''_a = \left(\delta_2(\Delta_0)\right)^{2q-5}\Delta_0^{2q-4}, \quad \Delta''_b = \left(\delta_2(\Delta_0)\right)^{2q-7}\Delta_0^{2q-6}. \tag{40}$$

We can compute the RG flow of the coupling ratio $r = J_a/J_b$ from (40)

$$r'' = \left(\gamma_2(\Delta_0)\right)^2 r. \tag{41}$$

Let us stress that this is the result after two SDRG transformations, or equivalently after a single sequence-preserving transformation $\Xi^{\{6,q\}} = RG_2^{\{6,q\}} \circ RG_1^{\{6,q\}}$. A complete flow of the couplings can be then computed by repeatedly applying this analysis and producing further generations of couplings which follow the original sequence but whose values get renormalized, cf. (33) and (34). Regardless of the explicit value of the couplings at each RG step, the fixed points can be determined from properties of $\gamma$ and $\delta$. For an initial anisotropy $0 \leqslant \Delta_0 < 1$, we have $\delta_2(\Delta_0) < 1$ and thus the anisotropies flow towards a XX chain fixed point with $\Delta_a^* = \Delta_b^* = 0$, as discussed below (34). Correspondingly, the coupling ratio flows to a non-vanishing $r^*$, which depends on the initial value of $r$. This is consistent with the marginality of $\{p, q\}$ modulations in XX chains. For $\Delta_0 = 1$, i.e. the XXX chain, the anisotropies are not renormalized and equal unity throughout the whole RG procedure. Since $\gamma_2(\Delta_0 = 1) = \frac{1}{2}$, we find that the bare coupling flows towards a strong disorder fixed point, characterized by $r' \to r^* = 0$. Thus, $\{6, q\}$ aperiodic modulations are

Figure 8: Top line: Rescaled sequence appearing in the bottom line of Fig. 7 after the first RG step. Applying a second RG step (38), $(q-3)$ blocks and their preceding bonds get renormalized to a weak coupling $J_a''$, while the $(q-4)$ blocks are renormalized to a strong coupling $J_b''$, both given in (39). Bottom line: After two RG steps, we recover the original sequence, indicating that the bond attractor for $\{6,q\}$ modulations is a 2-cycle.

relevant when applied to XXX chains.

In this section we have introduced a straightforward construction for defining a theory on the boundary of a given $\{p,q\}$ tiling of the Poincaré disk. We have associated a spin 1/2 to each edge and a bond connecting nearest-neighbor spins to each vertex. The letters $a$ and $b$ of the asymptotic sequence on the boundary have been related to two different couplings $J_a$ and $J_b$ along the spin chain. This way, we have constructed an aperiodic spin chain on the boundary of the tiling whose modulation is governed by the letter sequence generated by $\sigma_{\{p,q\}}$. We have focused on infinite aperiodic XXZ chains with modulations of the hopping parameters and homogeneous anisotropy parameter $\Delta_0$ (see the Hamiltonian (22)). We have reviewed the SDRG techniques as a method for determining the critical properties of aperiodic systems. Exploiting this method, we have argued that for any pair $\{p,q\}$ and for $0 \leqslant \Delta_0 < 1$, the aperiodic modulation is marginal and the system is characterized by a line of fixed points depending on the coupling ratio. In contrast, for $\Delta_0 = 1$ all the $\{p,q\}$ modulations are relevant and the system flows towards a strong-disorder fixed point independent of the couplings (see Fig. 6). This will be the case of interest for us in the next sections.

## 4 Entanglement entropy in aperiodic XXX chains

As pointed out in Sec. 3.2, the infinite aperiodic XXX chain with modulations generated by $\sigma_{\{p,q\}}$ has a critical behavior governed by an aperiodicty-induced fixed point, which depend on the Schläfli parameters $\{p,q\}$. In this section we address the question of how the entanglement properties of this aperiodic model depends on the pair $\{p,q\}$ that determine the modulation of the couplings. Notice that, because of the relevance of the modulation, the entanglement in the aperiodic XXX chain is expected to depend on $p$ and $q$ only. Instead, in cases where the modulation generated by $\sigma_{\{p,q\}}$ is marginal, as in the aperiodic XX chain, we expect the entanglement to depend also on the coupling ratio [75].

### 4.1 Entanglement entropy in aperiodic singlet phases

In this section we compute the entanglement entropy of a block of consecutive spins in an infinite XXX chain, whose Hamiltonian is given by (22) with $\Delta_0 = 1$, in presence of a particular class of aperiodic modulations. Notice that, because of the spatial inhomogeneity of these models, we have to consider the entanglement entropy averaged over different starting positions of the subsystem. With a slight abuse of notation we will refer to this

average simply as entanglement entropy. In what follows we consider aperiodic sequences of couplings for which the SDRG procedure described in Sec. 3.2 produces exclusively 2-spin singlets. Given that only singlets are created along the RG flow, after a large number of RG steps, the ground state of the chain consists of singlets of spins separated by arbitrary large distance. The system is then said to be in an *aperiodic singlet phase* [93]. Let us stress that, within the geometric setup introduced in the previous sections, this state is defined on one $p$-th of the boundary of a $\{p, q\}$ tiling, while the whole boundary chain is achieved by exploiting the $\mathbb{Z}_p$ symmetry.

Given the infinite chain in its ground state, consider now a subsystem $A$ consisting of $L$ consecutive spins. If the system is in the aperiodic singlet phase, the entanglement entropy of $A$ can be computed analytically, since in that case one simply has to count the singlet bonds connecting the subsystem with the rest of the chain [93]. Each singlet will give a contribution of $\ln 2$ to the entanglement entropy. In [74] this idea was applied to aperiodic XXZ chains with modulations given by the so-called *singlet producing self-similar sequences*, namely sequences generated by inflation rules corresponding to the inverse of the SDRG steps. As explained in Sec 3.2, these sequences all have a 1-cycle as bond distribution attractor by construction. In the following, we generalize the computation of [74] to those aperiodicities that along SDRG flow are characterized by a 2-cycle as bond distribution attractor (see Sec. 3.2). The sequence-preserving transformations of this class of flows can be written as the combination of only two RG steps, namely $\Xi = \mathrm{RG}_2 \circ \mathrm{RG}_1$. The example discussed in detail in Sec. 3.3 belongs to this class.

Since the bond distribution attractor is a 2-cycle, the original sequence is renormalized into itself through the application two distinct deflation rules, which define a sequence-preserving transformation. Let us call $M_1$ and $M_2$ the substitution matrices of the two corresponding inflation rules: in this notation, the sequence-preserving transformation corresponds to $M_2^{-1}M_1^{-1} = (M_1M_2)^{-1}$. When $M_1 = M_2$ we recover exactly the setup of [74]. We assume that $M_1$ and $M_2$ are both symmetric matrices; this holds in all the cases of our interest. Under this assumption, $M_2M_1$ and $M_1M_2$ have the same eigenvalues. We call $\lambda_+^{(12)}$ the largest of them, which, in general, is not equal to the product of the largest eigenvalues of $M_1$ and $M_2$ given that the two matrices do not commute.

As explicitly shown in the example reported in Sec. 3.3, when the bond-distribution attractor is a 2-cycle, the coupling distribution along the chain alternates between even and odd generations (obtained by even and odd numbers of RG steps respectively). The $k$-th even generation (or $2k$-th overall generation), with $k \in \mathbb{N}$, is achieved by applying on the $k - 1$-th even one the matrix $(M_1M_2)^{-1}$ and therefore the system undergoes a rescaling of $\left(\lambda_+^{(12)}\right)^{-1}$. On the other hand, the $k$-th odd generation (or $2k - 1$-th overall generation) is achieved by applying on the $k - 1$-th odd one the matrix $(M_2M_1)^{-1}$ and, also in this case, the system undergoes a rescaling of $\left(\lambda_+^{(12)}\right)^{-1}$. An initial condition relating the first even and odd generations is required. This involves a single application of $M_1$, but the corresponding scaling cannot be directly inferred from it since $M_1$ does not generate the even generations asymptotically (because $[M_1, M_2] \neq 0$). We denote the scaling factor corresponding to this first transformation $\tilde{\lambda}$ and we will determine its exact value later.

Let us treat even and odd generations separately. Notice that the singlets in the $k$-th generation correspond to the strong bonds in the $k - 1$-th generation and therefore their characteristic length $\Lambda_k$ reads

$$\Lambda_{2k-1} \;=\; l_b^{(e)} \left(\lambda_+^{(12)}\right)^{k-1}, \tag{42}$$

$$\Lambda_{2k} \;=\; l_b^{(o)} \left(\lambda_+^{(12)}\right)^{k-1} \tilde{\lambda}, \tag{43}$$

where $l_b^{(o)}$ and $l_b^{(e)}$ are the $b$-th components of the left eigenvectors associated to $\lambda_+^{(12)}$ of $M_2 M_1$ and $M_1 M_2$ respectively. Moreover, we have made use of the fact that the first generation of singlets corresponds to the distribution of strong bonds in the original chain, thus implying $\Lambda_1 = l_b^{(e)}$. In contrast, the concentration of singlets in the $k$-th generation reads

$$\rho_{2k-1} = p_b^{(e)} \left(\lambda_+^{(12)}\right)^{-k+1}, \tag{44}$$

$$\rho_{2k} = p_b^{(o)} \left(\lambda_+^{(12)}\right)^{-k+1} \tilde{\lambda}^{-1}, \tag{45}$$

where $p_b^{(o)}$ and $p_b^{(e)}$ are the $b$-th component of the right eigenvectors associated to $\lambda_+^{(12)}$ of $M_2 M_1$ and $M_1 M_2$ respectively.

In order to obtain $\tilde{\lambda}$, recall that after a large number of RG transformations the system is supposed to be in an aperiodic singlet phase. This means that the sum over the singlet concentrations of all the possible generations must give $\frac{1}{2}$, namely

$$\sum_{k=1}^{\infty} \left(p_b^{(o)} \tilde{\lambda}^{-1} + p_b^{(e)}\right) \left(\lambda_+^{(12)}\right)^{-k+1} = \left(p_b^{(o)} \tilde{\lambda}^{-1} + p_b^{(e)}\right) \frac{\lambda_+^{(12)}}{\lambda_+^{(12)} - 1} = \frac{1}{2}. \tag{46}$$

Inverting this relation, we obtain

$$\tilde{\lambda} = \frac{2 p_b^{(o)} \lambda_+^{(12)}}{\lambda_+^{(12)} \left(1 - 2 p_b^{(e)}\right) - 1}. \tag{47}$$

Moreover, assuming that $M_1$ and $M_2$ are symmetric, it is straightforward to verify that

$$p_b^{(o)} l_b^{(o)} = p_b^{(e)} l_b^{(e)}. \tag{48}$$

The fraction of the chain occupied by the singlets of the $k$-th generation is given by $W_k = \rho_k \Lambda_k$ and therefore, using (42)-(45), we have

$$W_{2k-1} = p_b^{(e)} l_b^{(e)}, \qquad W_{2k} = p_b^{(o)} l_b^{(o)}, \tag{49}$$

for all $k \in \mathbb{N}$. In turn, due to (48), this means that $W_k$ does not depend on $k$. Following the procedure of [74], the generations of singlets with length $\Lambda_k < L$ do contribute to the entanglement entropy as

$$S_{A,<} = 2 \ln 2 \, p_b^{(e)} l_b^{(e)} \left(n^{(e)}(L) + n^{(o)}(L)\right) \equiv 2 \ln 2 \, p_b^{(e)} l_b^{(e)} n(L), \tag{50}$$

where $n^{(e)}(L)$ and $n^{(o)}(L)$ are respectively the number of even and odd generations of singlets such that $\Lambda_k < L$, whose expressions read

$$n^{(e)}(L) = \left\lfloor \frac{\ln\left(L/l_b^{(o)}\right) - \ln\tilde{\lambda}}{\ln\lambda_+^{(12)}} \right\rfloor, \qquad n^{(o)}(L) = \left\lfloor \frac{\ln\left(L/l_b^{(e)}\right)}{\ln\lambda_+^{(12)}} \right\rfloor + 1, \tag{51}$$

with $\lfloor \cdot \rfloor$ indicating the floor function. On the other hand, the generations of singlets with $\Lambda_k > L$ give the following contribution to the entanglement entropy

$$S_{A,>} = 2L \ln 2 \sum_{k=n(L)+1}^{\infty} \rho_k = 2L \ln 2 \left(\frac{1}{2} - \sum_{k=1}^{n(L)} \rho_k\right). \tag{52}$$

A distinction between even and odd values of $n(L)$ is required. When $n(L)$ is odd, we have

$$
S_{A,>} = 2L \ln 2 \left[ \frac{1}{2} - \sum_{k=1}^{[n(L)+1]/2} p_b^{(e)} \left( \lambda_+^{(12)} \right)^{-k+1} - \sum_{k=1}^{[n(L)-1]/2} p_b^{(o)} \tilde{\lambda}^{-1} \left( \lambda_+^{(12)} \right)^{-k+1} \right] \quad (53)
$$

$$
= 2L \ln 2 \frac{p_b^{(e)} + p_b^{(o)} \tilde{\lambda}^{-1} \lambda_+^{(12)}}{\lambda_+^{(12)} - 1} \left[ \lambda_+^{(12)} \right]^{-(n(L)-1)/2} \equiv S_{A,>}^{(o)}, \quad (54)
$$

while for $n(L)$ even we have

$$
S_{A,>} = 2L \ln 2 \left[ \frac{1}{2} - \sum_{k=1}^{n(L)/2} p_b^{(e)} \left( \lambda_+^{(12)} \right)^{-k+1} - \sum_{k=1}^{n(L)/2-1} p_b^{(o)} \tilde{\lambda}^{-1} \left( \lambda_+^{(12)} \right)^{-k+1} \right], \quad (55)
$$

$$
= 2L \ln 2 \frac{p_b^{(e)} + p_b^{(o)} \tilde{\lambda}^{-1}}{\lambda_+^{(12)} - 1} \left[ \lambda_+^{(12)} \right]^{-n(L)/2+1} \equiv S_{A,>}^{(e)}, \quad (56)
$$

where we have exploited the last equality in (46) in both cases. Finally, given that $S_A = S_{A,<} + S_{A,>}$, we have

$$
S_A = 2 \ln 2 \, p_b^{(e)} l_b^{(e)} n(L) + \begin{cases} S_{A,>}^{(e)} & \text{when } n(L) \text{ is even}, \\ S_{A,>}^{(o)} & \text{when } n(L) \text{ is odd}, \end{cases} \quad (57)
$$

where $S_{A,>}^{(o)}$ and $S_{A,>}^{(e)}$ are defined in (54) and (56) respectively. The expression for the entanglement entropy in (57) represents a main result of this section and generalizes the results found in [74] in the sense that it holds for a larger class of inflation rules. When the sequence-preserving transformation is given by a single deflation step, namely the bond-distribution attractor is a 1-cycle rather than a 2-cycle, we recover the scenario considered in [74]. In particular, if we take $M_1 = M_2 = M_\sigma$, where $M_\sigma$ is the substitution matrix of the inflation rule generating the original sequence, it is straightforward to verify that the entanglement entropy (57) reduces to the result of [74], which does not depend on the parity of $n(L)$.

A more detailed investigation of (57) shows that the dependence of $S_A$ on $L$ occurs through $n(L)$, which is a floor function. This leads to a piecewise linear behavior of the entanglement entropy, typical of aperiodic systems [74, 75]. This behavior is different from the one observed in one-dimensional homogeneous critical systems, where the entanglement entropy of an interval grows logarithmically in the subsystem size [59, 60]. Nevertheless, it is possible to make contact between these two classes of systems through the following observation. The piecewise curve (57) has breaking points corresponding to $L = \Lambda_k$, where $\Lambda_k$ is defined in (42)-(43). The sets of points $\{\Lambda_{2k-1}\}$ and $\{\Lambda_{2k}\}$ uniquely determine two distinct logarithmic envelopes of (57) with equal coefficients, but different additive constant. They read

$$
S_{\text{env},1} = \frac{c_{\text{eff}}}{3} \ln L + \kappa_1, \qquad S_{\text{env},2} = \frac{c_{\text{eff}}}{3} \ln L + \kappa_2, \quad (58)
$$

where

$$
c_{\text{eff}} = \frac{12 p_b^{(e)} l_b^{(e)} \ln 2}{\ln \lambda_+^{(12)}}. \quad (59)
$$

Notice that the two sets of breaking points become a unique series when $M_1 = M_2$ and therefore the two envelopes coalesce, allowing to recover the behavior found in [74]. Despite

the different additive constant, the two envelope functions (58) have equal coefficients in the logarithmic terms. Therefore, inspired by [74, 75] and by the results for homogeneous critical systems, we are led to interpret this prefactor as an effective central charge $c_{\text{eff}}$. In Fig. 9 we show the entanglement entropy (57) with the envelopes (58) for our prime example of a 2-cycle bond distribution attractor $\{6, 5\}$. In the next subsection we consider the aperiodicities that satisfy the requirements discussed above and are generated by $\sigma_{\{p,q\}}$ for some $\{p, q\}$ and discuss how the corresponding $c_{\text{eff}}$ depends on $p$ and $q$. Thus, we remand a detailed explanation of Fig. 9 to that section.

Let us conclude this subsection with the following remark on the Rényi entropies of the system. Within the approximation we are considering, our system is supposed to be in an aperiodic singlet phase and therefore its full density matrix $\rho$ factorizes into singlet (pure state) density matrices $\rho_i^{(2s)}$. Thus, any possible reduced density matrix is the product of pure state density matrices and the reduced density matrices $\rho_j^{(1s)}$ of individual spins in the singlets cut by the entangling points, namely

$$
\rho_A = \left( \bigotimes_{i \in \text{singlets}} \rho_i^{(2s)} \right) \otimes \left( \bigotimes_{j \in \text{cut singlets}} \rho_j^{(1s)} \right) . \tag{60}
$$

It is well known that the entanglement entropy of a single spin in a singlet is equal to $\ln 2$. Moreover, all other Rényi entropies $S_A^{(\alpha)} = \frac{1}{1-\alpha} \ln \left( \text{Tr}(\rho_A^\alpha) \right)$ with $\alpha > 1$ are also $\ln 2$. This means that generalising our computation to encompass the Rényi entropies is trivial and would not change the result for the entanglement entropy (as long as we stay within our assumptions). This result is very different from the behavior of the Rényi entropies as function of the Rényi index in homogeneous critical lattice models [62], where one finds a logarithmic growth in the subsystem size with a coefficient explicitly dependent on $\alpha$, rather than a piecewise linear behavior totally independent of the Rényi index.

## 4.2 Effective central charge in $\{p, q\}$ aperiodic spin chains

In this section, we apply the results of Sec. 4.1 to those aperiodic modulations generated by $\sigma_{\{p,q\}}$ that satisfy the assumptions of singlet producing self-similar sequences. In particular, our generalization of the results in [74] given in Sec. 4.1 allows us to use (57) and (59) to investigate the entanglement entropy of aperiodic chains with a 2-cycle bond distribution attractor under SDRG. As detailed in Sec. 3.2, $\sigma_{\{6,q\}}$ with $q \geqslant 4$ belongs to the class of inflation rules mentioned above. For this case, the matrices $M_1$ and $M_2$ describing the sequence-preserving transformation are given by (36) and (38), respectively. It is straightforward to compute the corresponding $p_b^{(e)}$, $l_b^{(e)}$, $p_b^{(o)}$, $l_b^{(o)}$, $\tilde{\lambda}$ and $\lambda_+^{(12)}$, as described in Sec. 4.1. Using these quantities, one can compute the entanglement entropy (57), whose specific expression depends on $q$. Plugging $p_b^{(e)}$, $l_b^{(e)}$ and $\lambda_+^{(12)}$ into (59), we get the coefficient of the logarithmic envelopes of the piecewise linear entanglement entropy, i.e. the effective central charge, given by

$$
c_{\text{eff}}(6, q) = \frac{6 - 2q + \sqrt{q^2 - 5q + 6}}{\ln \left( 2q - 5 + 2\sqrt{q^2 - 5q + 6} \right)} \frac{6 \ln 2}{6 - 2q} , \tag{61}
$$

which exhibits a non trivial dependence on $q$.

In Fig. 9 we present the entanglement entropy for the exemplary case of $q = 5$ (black piecewise curve). The breaking points occur at $L = \Lambda_{2k-1}$ (blue vertical dashed lines) and $L = \Lambda_{2k}$ (purple vertical dashed lines) given in (42) and (43), respectively. Moreover, Fig. 9 shows the two envelopes (58) with (61) evaluated for $q = 5$ as coefficient of the logarithm

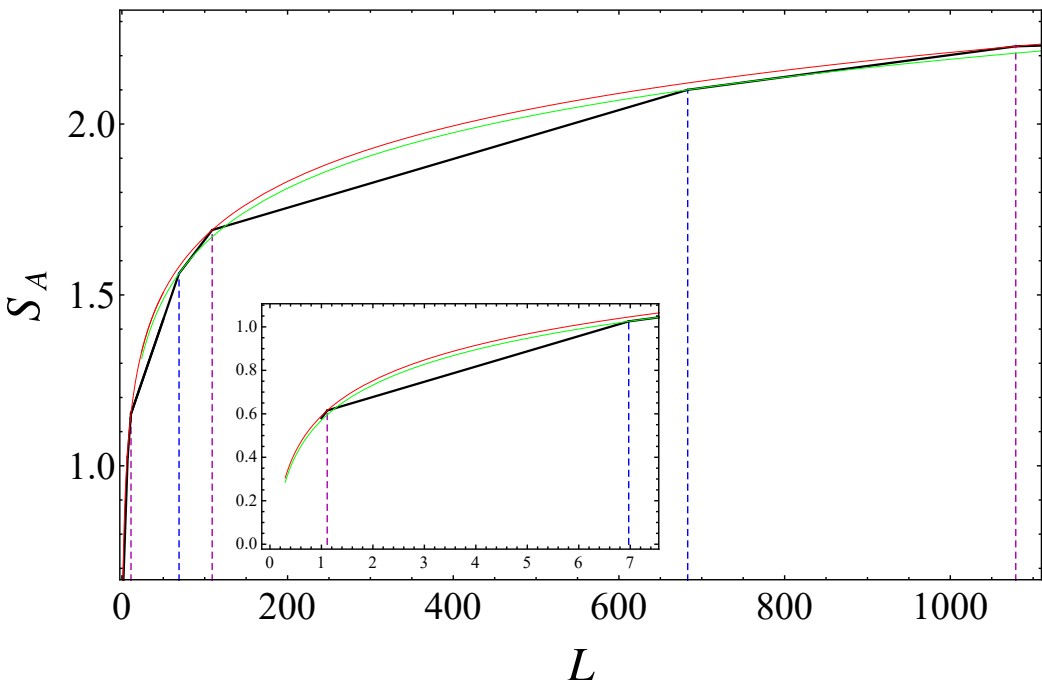

Figure 9: Entanglement entropy of a block of $L$ consecutive sites in an aperiodic XXX chain with modulation induced by the inflation rule $\sigma_{\{6,5\}}$. The black solid curve is obtained from (57), while the coloured ones correspond to the logarithmic envelopes in (58) with $\kappa_1 \simeq 0.576$ (green) and $\kappa_2 \simeq 0.591$ (red). Both curves have a coefficient of the logarithm given by (61) with $q = 5$. The vertical dashed lines correspond to the breaking points of the black curve occurring at $L = \Lambda_{2k-1}$ (blue lines) and $L = \Lambda_{2k}$ (purple lines) given in (42) and (43), respectively.

(coloured curves). The additive constants of the these two curves have been fitted and read $\kappa_1 \simeq 0.576$ (green) and $\kappa_2 \simeq 0.591$ (red). The green envelope touches the piecewise linear entropy function in the breaking points at $L = \Lambda_{2k-1}$, while the red one in the breaking points at $L = \Lambda_{2k}$.

Another case of interest is $q = 4$, for which $M_1^{\{6,4\}} = M_2^{\{6,4\}}$ and the bond distribution attractor is a 1-cycle. More precisely, the asymptotic sequence generated by $\sigma_{\{6,4\}}$ is known as *silver mean sequence*. Applying this modulation to the XXX chain leads to a piecewise linear entanglement entropy with a unique logarithmic envelope and to an effective central charge $c_{\text{eff}}(6,4) \simeq 0.6910$, consistently with the findings of [74]. Thus, we not only recover a known result, but also provide a new interpretation as effective central charge of a possible boundary theory for the regular $\{6,4\}$ tessellation of the Poincaré disk.

Other examples of aperiodic modulations generated by $\sigma_{\{p,q\}}$ that satisfy the hypothesis in Sec. 4.1 for being singlet producing self-similar sequences are those corresponding to $\{3,8\}$ and $\{5,4\}$ tilings. Interestingly, we observe that $M_1^{\{3,8\}} = M_2^{\{5,4\}}$ and $M_2^{\{3,8\}} = M_1^{\{5,4\}}$. According to the discussion in Sec. 4.1, this means that the distribution of bonds in even generations in the aperiodic $\{3,8\}$ chain is equal to the distribution in the odd ones in the aperiodic $\{5,4\}$ chain and viceversa. This fact leads to entanglement entropies with different shapes in the two cases, i.e. different additive constants, but to the same effective central charge

$$c_{\text{eff}}(3,8) = c_{\text{eff}}(5,4) = \frac{2(3-\sqrt{3})\ln 2}{\text{arcosh}\, 7} \simeq 0.6674 \, . \tag{62}$$

In other words, the effective central charge depends only on $M_1$ and $M_2$, regardless of their order of occurrence in the SDRG procedure.

Finally, notice that $\sigma_{\{3,7\}} \sim \sigma_{\{5,5\}}$ in the sense explained below (9), meaning that their asymptotic sequences are equal. In fact, this resulting asymptotic sequence is known in the literature as the *Fibonacci sequence*. Indeed, considering the substitution matrices (10) associated to $\sigma_{\{3,7\}}$ and $\sigma_{\{5,5\}}$ and computing the corresponding eigenvectors $\mathbf{v}_+$ and $\mathbf{u}_+$, one finds for both pairs of Schläfli parameters

$$\mathbf{v}_+^{\text{t}} = \left( \frac{\sqrt{5}-1}{2}, \frac{3-\sqrt{5}}{2} \right), \qquad \mathbf{u}_+^{\text{t}} = \left( \frac{5+3\sqrt{5}}{10}, \frac{5+\sqrt{5}}{10} \right), \tag{63}$$

which identify the Fibonacci sequence as asymptotic sequence. The aperiodic XXX chain with this modulation is sometimes called Fibonacci XXX chain. The entanglement entropy of a block of consecutive spins in Fibonacci XXX chains has been computed in [74,75] and has an expression consistent with (57) once we apply our construction to the Fibonacci modulation. Given that the bond distribution attractor of the Fibonacci XXX chain under the SDRG is a 1-cycle, the piecewise linear entanglement entropy has a unique logarithmic envelope, whose coefficient determines the effective central charge as [74,75]

$$c_{\text{eff}}(3,7) = c_{\text{eff}}(5,5) = 3\frac{5-\sqrt{5}}{5} \frac{\ln 2}{\text{arcsinh}2} \simeq 0.7962. \tag{64}$$

Thus, through our approach we have recovered the known results for the entanglement entropy in an infinite Fibonacci XXX chain, together with the coefficient of its logarithmic envelope. Moreover, we provided the interpretation of the latter as the effective central charge of the theory on the boundary of a $\{3,7\}$ (or $\{5,5\}$) hyperbolic tiling.

In summary, we have computed the entanglement entropy of a block $L$ of consecutive spins in the aperiodic XXX chain with modulations induced by a specific class of tiling inflation rules $\sigma_{\{p,q\}}$. This class contains singlet producing self-similar sequences whose bond attractor under the SDRG procedure is a 2-cycle, thus generalizing the analysis from [74, 75]. We have found that the entanglement entropy exhibits a piecewise linear behavior as function of the subsystem size. This is a peculiar feature of entanglement in aperiodic chains [74, 75]. The piecewise linear entanglement entropy (57) for 2-cycle sequences exhibits two logarithmic envelopes which only differ by an additive constant. The coefficient of the logarithm is interpreted as an effective central charge, and we have derived its explicit dependence on the Schläfli parameters $p$ and $q$ that determine the modulation $\sigma_{\{p,q\}}$.

In Sec. 2.3, we have considered the same bipartition on the boundary of a $\{p, q\}$ tiling and we have computed the entanglement entropy in the discretized bulk. The result (20) grows logarithmically in the subsystem size, differently from (57) which exhibits a piecewise linear behavior. Moreover, in (21) we have defined $c_{\text{eff,bulk}}(p,q)$, relating it to the maximal central charge for perfect TNs introduced in [52]. Thus, in order to have a thorough comparison of all the available results, we can compare $c_{\text{eff}}(p,q)$ computed in this section with the findings of [52]. First we have observed that, for all the pairs $\{p,q\}$ considered above, the latter effective central charge is always larger than the former. Moreover, for $p = 6$ and $q > 4$, we have observed that (61) and the corresponding result in [52] are both decreasing functions of $q$ and vanish as $q \to \infty$. We refer to Sec. 6 for a more detailed discussion.

# 5 Tensor network states of aperiodic XXZ chains

The construction presented in the previous sections started from a hyperbolic tiling of $\mathbb{D}^2$ and its associated inflation rule (Sec. 2). This lead us to define an aperiodically modulated XXZ model based on the letter distribution of the tiling's boundary (Sec. 3). We now present an additional construction, based on tensor networks, that allows us to exactly recover the ground state of the aperiodic XXX chain ($\Delta_0 = 1$).

Tensor networks naturally implement the idea of real space RG [30–41, 94, 95] since they approximate a state on their boundary by a construction extending in one dimension higher. Here, we incorporate the SDRG transformations introduced in Sec. 3.2 in tensor networks and show how these provide a general derivation of RG flows of the XXZ couplings for arbitrary values of $p$ and $q$. We stress that the embedding of the TN onto the Poincaré disk allows it to inherit some but, crucially, not all of the symmetries of the tiling. We discuss in detail why this is the case and how it is related to a choice of coordinates for the inflation procedure.

## 5.1 Tensor network representation of SDRG

We proceed with the construction of the TN that implements the SDRG transformations introduced in Sec. 3 on the aperiodic XXZ spin chain (22). Our construction is an implementation of the ideas introduced in [34].

As explained in Sec. 3.2, within the SDRG approximation, the ground state of the local Hamiltonian $H_n$ (23) of an $n$-spin block can be approximated by the singlet state (25) and a superposition of the doublet states (28). For the construction of the TN, it is practical to introduce the following notation for these states, labeled by the number $n$ of spins in the block that is to be renormalized,

$$|T_n\rangle = \begin{cases} \frac{1}{\sqrt{2}} \sum_{\{m_i = \pm\}} \delta^{m_0}_{m_1} |m_0\rangle |m_1\rangle \,, & n = 1 \,, \\[2mm] \sum_{\{m_i = \pm\}} T_{m_1 \dots m_n} |m_1\rangle \dots |m_n\rangle \,, & n \text{ even} \,, \\[2mm] \frac{1}{\sqrt{2}} \sum_{\{m_i = \pm\}} T^{m_0}_{m_1 \dots m_n} |m_0\rangle |m_1\rangle \cdots |m_n\rangle \,, & n \text{ odd} \neq 1 \,. \end{cases} \tag{65}$$

Notice that the first line of (65) defines an additional state $|T_{n=1}\rangle$ describing the spins that are not renormalized under a given SDRG step, i.e. the unaffected spins. Moreover, the state $|T_n\rangle$ for $n$ even is precisely (25), while for $n$ odd we define a superposition of the states in (28) by further summing over the free index $m_0$. Written in this form, we can interpret the coefficients $\delta^{m_0}_{m_1}$, $T_{m_1 \dots m_n}$, $T^{m_0}_{m_1 \dots m_n}$ as tensors of rank 2, $n$ and $n + 1$, respectively. These tensors, or equivalently, their corresponding states in (65), implement the fundamental steps involved an SDRG transformation (as depicted previously in Fig. 5). For a generic step in the RG flow, we denote the sequence of spins before the application of an SDRG transformation as the UV chain, and the sequence after the transformation as IR chain. In the tensors $\delta^{m_0}_{m_1}$, $T_{m_1 m_2 \dots m_n}$ and $T^{m_0}_{m_1 m_2 \dots m_n}$ (denoted in Fig. 10 by $\bigcirc, \triangle, \square$, respectively) each lower index corresponds to a tensor leg connected to a spin on the UV chain, while each upper index corresponds to a tensor leg connected to a spin on the IR chain. We will use the terminology of tensors and states interchangeably in the following. Recall that each index $m_i$ corresponds to a local Hilbert space $\mathcal{H}$ of dimension two, while also labeling a complete basis of states $\{|m_i = \pm\rangle\}$ in this Hilbert space. The state $|T_n\rangle$ corresponding to a tensor is thus defined in a local product Hilbert space spanned by the tensors legs, i.e. $|T_n\rangle \in \mathcal{H}^{\otimes 2\lfloor (n+1)/2 \rfloor}$. Assigning these tensors according to the SDRG procedure through the full RG flow, we obtain a collection of tensors, each denoting a local

state at some point of the graph which we label by the index $b$. The state corresponding to this collection of tensors is given by their outer product

$$\bigotimes_b |T_n\rangle_b \, , \tag{66}$$

where the index $b$ labels the position of the tensors in the collection. The set of all the positions of the tensors in the collection determines a disconnected graph, where each node corresponds to a tensor. Examples of these positions are visualized by the symbols $\bigcirc, \triangle, \square$ in Fig. 10.

The next step in order to construct the TN is to connect the tensors through internal lines. These internal lines are drawn wherever two tensors share a common leg according to the geometry of the TN graph. Incidentally, these are precisely the positions of spins along the chain at that particular point along the RG flow. For this, it is useful to label the Hilbert spaces corresponding to tensor legs by the vertices of the graph that they connect. For example, if we have two tensors at positions $b$ and $b'$, we label the Hilbert spaces of their shared legs as $\mathcal{H}_{bb'}$ and $\mathcal{H}_{b'b}$, respectively. There are also some legs that are connected to only one tensor, these are called *open* or *dangling* legs. In our setup, these are precisely the legs describing the degrees of freedom where the SDRG starts, which we refer to as the *original UV chain*.

As in the standard SDRG approach introduced in Sec. 3.2, the fundamental decimation steps described above are applied simultaneously to all spin-blocks on the UV chain. In terms of tensors, simultaneous application means taking the outer product of the tensors decimating individual spin-blocks. By iterating the SDRG, we construct the state (66), which describes the collection of all the tensors. The full network is then constructed by contracting upper and lower indices of different tensors if they share a leg. In practice, the contraction of a common leg of tensors at positions $b$ and $b'$ consists of a projection in the local product Hilbert space $\mathcal{H}_{bb'} \otimes \mathcal{H}_{b'b}$ onto a maximally entangled state denoted by $|bb'\rangle$. Since the local Hilbert spaces in our TN are all equal and of dimension two, this state is an EPR state in the local basis

$$\big|bb'\big\rangle = \frac{1}{\sqrt{2}} \left(|+\rangle\,|+\rangle + |-\rangle\,|-\rangle\right). \tag{67}$$

Thus, schematically, the full *tensor network state* $|\Psi\rangle$ is obtained by performing this contraction over all pairs of tensors that share a leg, resulting in

$$|\Psi\rangle = \left(\bigotimes_{\langle bb'\rangle} \langle bb'|\right) \left(\bigotimes_b |T_n\rangle_b\right), \tag{68}$$

where the indices $b, b'$ run over all the tensor positions in the TN graph, and $\langle bb'\rangle$ denotes a shared leg. TN states constructed as in (68) are known as *projected entangled pair states* (PEPS) [96]. The TN state $|\Psi\rangle$ is defined in the product Hilbert space of all dangling legs of the original UV chain and describes the ground state of the aperiodic XXX chain. Notice that this statement holds for XXX chains only, because the $\{p, q\}$ aperiodic modulations of XXZ chains are only relevant for $\Delta_0 = 1$. As explained in Sec. 3.2, this implies that close to the strong-disorder fixed point characterized by $r^* = 0$ the SDRG becomes asymptotically exact. Since the construction described above relies completely on the SDRG procedure, our TN inherits the same validity regime and thus reproduces only the ground states of $\{p, q\}$ aperiodic XXX chains. For the general case of XXZ chains with $0 \leqslant \Delta_0 < 1$, this TN can still be used to approximate the ground state, but only in the regime of parameters which keeps the coupling ratio $r$ much smaller than one along the whole SDRG flow. In the following, we will thus focus only on aperiodic XXX chains.

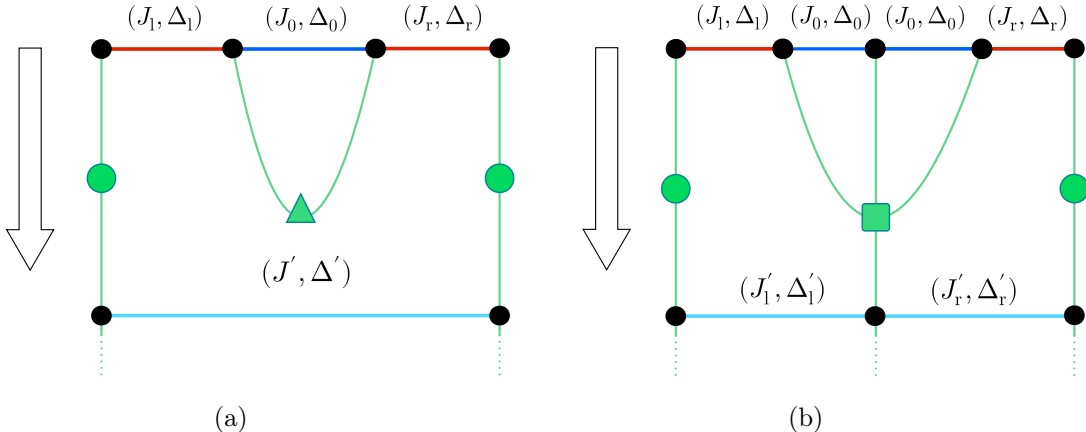

(a)             (b)

Figure 10: Tensor representation of the fundamental SDRG transformations previously introduced in Fig. 5. Tensors associated to the states $|T_n\rangle$ in (65) are depicted as green circles ($n = 1$), triangles ($n$ even) and squares ($n \neq 1$ odd), with the green legs representing their indices. The spins $\sigma_{l,r}$ are contracted with the identity tensor associated to $|T_1\rangle$ and remain unchanged. a) Decimation of a 2-spin singlet by contracting the 2 spins with the rank-2 tensor associated to $|T_2\rangle$. b) Decimation of a 3-spin doublet by contracting the 3 spins with the rank-4 tensor associated to $|T_3\rangle$. For clarity, we will drop the green squares, triangles and circles in the following figures of TNs.

The construction described above does not yet address the other end of the TN, namely the uncontracted indices on the last IR chain of the network. For this, we have to make a distinction between an XXX chain defined on an infinite line and it being on a finite circle. In the former case, the SDRG can be repeated indefinitely and the resulting IR chain will still be infinite. The remaining uncontracted upper indices are not relevant in this situation, since they do not affect the properties of any local measurement at finite scale on the UV chain. However, in the latter case, the finiteness of the circle implies that the SDRG procedure has to end after a finite number of steps, since every decimation step reduces the number of degrees of freedom. In this situation, the remaining upper indices of the final IR chain have to be contracted. The spins associated to these open tensor legs are governed by an IR XXX Hamiltonian whose couplings are modulated by the seed sequence, i.e. the starting sequence for the inflation rule. As explained in Sec. 2.2, in the case of the $\{p, q\}$ inflation rules considered in this work, the seed sequence is $a^p$ (for $p > 3$) and $\circledast^3$ for $p = 3$. Thus, the IR Hamiltonian is a homogeneous XXX model with $p$ sites, whose ground state is the singlet state in (25) for even $p = n$ and one of the doublets in (28) for odd $p = n$. Consequently, we may contract the open tensor indices of the final IR chain with the rank-$p$ tensor corresponding to its ground state to obtain the full TN state. This final tensor of the IR chain is akin to the "top-level tensor" introduced in [97] for the MERA tensor network on a circle.

Based on this TN picture of fundamental decimation steps in the SDRG procedure, let us clarify the precise relation between inflation and SDRG. As mentioned in the previous sections, single RG transformations following the decimation steps shown in Fig. 5 and Fig. 10 are not necessarily sequence-preserving. This prompted the introduction of the cycle number $k$ in Sec. 3 to specify how many RG transformations are required to recover the original sequence. Then, a sequence-preserving SDRG transformation, denoted by $\Xi$, corresponds to the product of $k$ individual RG transformations. Equivalently, we can also find the number $m$ of inverse-inflation, i.e. deflation, steps according to the original inflation rule $\sigma_{\{p,q\}}$ that implement one such sequence-preserving transformation. In other

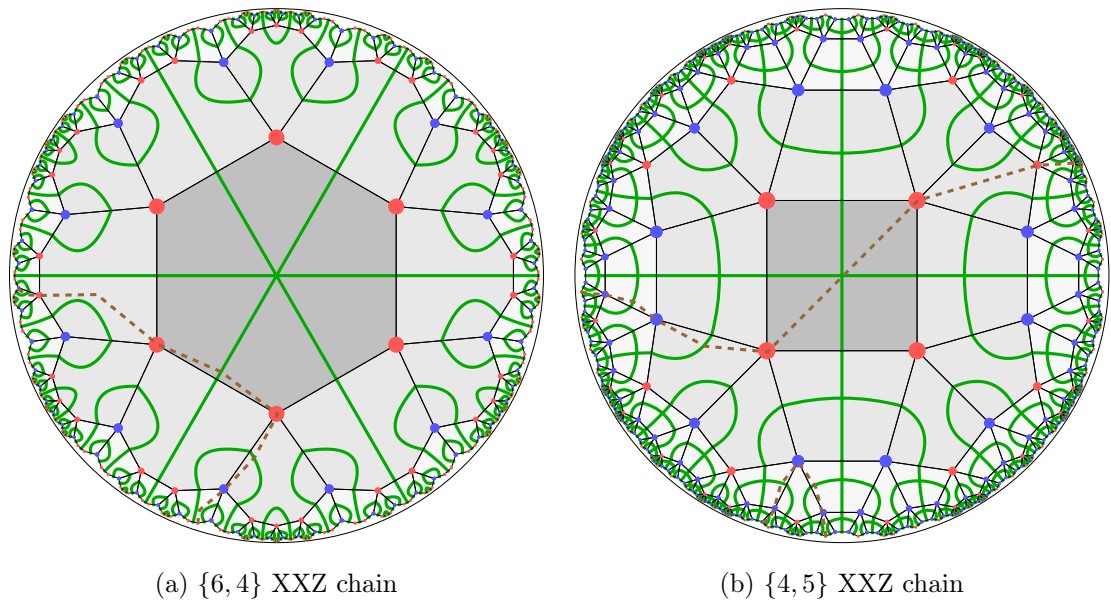

(a) $\{6,4\}$ XXZ chain

(b) $\{4,5\}$ XXZ chain

Figure 11: Tensor networks of the ground states of an aperiodic $\{p,q\}$ XXX chain on a circle. The TNs are depicted by green lines and fit the tiling commensurately. The weak and strong bonds in Fig. 10 at each step of the RG flow are denoted by red and blue dots, respectively. a) TN of the $\{6,4\}$ XXX chain, consisting of identical rank-2 tensors (excluding the central tensor). b) TN of the $\{4,5\}$ XXX chain, consisting of identical rank-4 tensors and rank-2 tensors. Various minimal cuts, homologous to the corresponding subregions on the boundary, are shown with brown dashed polylines.

words, a sequence-preserving SDRG transformation of the $\{p,q\}$-modulated chain is always equivalent to $m$ deflation steps

$$\Xi^{\{p,q\}} \sim \sigma_{\{p,q\}}^{-m}, \tag{69}$$

where we find that $m$ only takes on the values $m = 1, 2, 3$. Thus, we may classify sequence-preserving SDRG transformations and their corresponding TNs in three classes, denoted by I, II and III, respectively. Moreover, this classification can also be expressed directly in terms of the Schläfli parameters for a given $\{p,q\}$ tiling,

- when $p$ is even, the SDRG belongs to class I.

- when $p$ is odd and $q$ is even, the SDRG belongs to class II.

- when $p$ is odd and $q$ is odd, the SDRG belongs to class III.

We stress that the fact that this classification can always be achieved for all $p$ and $q$ is highly non-trivial and we provide a constructive proof of it in Appendix B.

As described above, repeated application of the SDRG transformation in its TN description defines how the tensors are to be connected with each other and thus determines the form of the TN. In particular, since $m$ is a finite integer, we can place the TN on the tiling in a commensurable way, where the layer of tensors generated by a sequence-preserving transformation $\Xi$ is placed on $m$ layers of polygons in the tiling. We illustrate the TN reproducing the ground states of the $\{6,4\}$ and $\{4,5\}$ XXX chain, embedded on the corresponding tilings, in Fig. 11. These two TNs belong to class I, and their sequence-preserving transformations are given by $\Xi^{\{6,4\}} = \sigma_{\{6,4\}}^{-1} = \{abaaaba \mapsto a, \ aba \mapsto b\}$ and

$\Xi^{\{4,5\}} = \sigma_{\{4,5\}}^{-1} = \{babab \mapsto a, \ bab \mapsto a\}$. Note that the structure of the TN does not coincide with that of the hyperbolic tiling. This will be discussed in depth in Sec. 5.4.

In the next subsections, we discuss some applications and exploit the properties of the tensor network introduced above to gain insights into the ground state of the corresponding XXX chain. In particular, we provide an intuitive algorithm to write down the SDRG flows of the couplings of the model and we determine an upper bound on the coefficient of the logarithmic envelope (cf. (58)) for the entanglement entropy of a block of consecutive spins.

## 5.2 Algorithm of RG flow from tensor networks

In this subsection we provide a graphical algorithm for determining the flows of the couplings (33) along SDRG in a $\{p,q\}$ aperiodic XXZ chain. This approach relies on the structure of the TN introduced in Sec. 5.1 and on the underlying $\{p,q\}$ tiling.

Consider a layer of tiles in a $\{p,q\}$ tessellation of the Poincaré disk. In Fig. 12 we have pictorially represented such a layer for the exemplary case of a $\{8,4\}$ tiling. For visual convenience, we have represented this portion of tessellation on a strip and the blue and red vertices of the tiling (see for instance the ones shown in Fig. 11) are depicted as edges of the same colors. Following the terminology introduced in Sec. 5.1, we denote the upper horizontal line in Fig. 12 as UV chain and the lower one as IR chain. The black edges between them are edges of the tiling while the green curves represent the TN. Following Fig. 4, we associate to each edge on the UV chain a pair of couplings $(J_i, \Delta_i)$, with $i \in \{a, b\}$. The aim is to renormalize the couplings along the UV chain into the ones along the IR chain. Consider an edge on the IR chain and let us denote its unknown couplings by $(J', \Delta')$. Notice that, by construction, the endpoints of each edge on the IR chain, i.e. the positions of the renormalized spins, are given by the intersections of the legs of the TN with that chain. Thus, we may define a region of the tiling bounded by the following elements: the IR edge with couplings $(J', \Delta')$ (bottom), the TN legs carrying the renormalized spins (left and right) and the sequence of edges in the UV chain encompassed by these legs (top). An example of this domain is depicted in yellow in Fig. 12. The sequence of couplings of the UV chain delimited by this region define the word that will be renormalized to the new coupling $(J', \Delta')$ under a deflation step. The explicit values of $(J', \Delta')$ can be written down exploiting the following algorithm, which is derived from the rules (27) and (31):

- a singlet of $n$ spins connected by couplings $(J_i, \Delta_i)$ contributes to $J'$ with a factor $\gamma_n(\Delta_i)/J_i$ and to $\Delta'$ with $\delta_n(\Delta_i)$.

- a doublet of $n$ spins connected by couplings $(J_i, \Delta_i)$ contributes to $J'$ with a factor $\gamma_n(\Delta_i)$ and to $\Delta'$ with $\delta_n(\Delta_i)$.

- An isolated bond with couplings $(J_i, \Delta_i)$ contributes to $J'$ with a factor $J_i$ and to $\Delta'$ with a factor $\Delta_i$.

Note, however, that not all internal couplings in the yellow shaded region go into the rules of the algorithm. Only those contained in the gray striped region in Fig. 12 are to be taken into account. While in the case of TNs implementing class I SDRG procedures this distinction is not essential, this becomes of crucial importance for studying the flow of the couplings for classes II and III, cf. (69). These cases require the introduction of auxiliary, internal letters that make the definition of the gray striped region non-trivial.

As an example, it is instructive to apply this algorithm explicitly to the $\{8,4\}$ tiling and its corresponding TN shown in Fig. 12. We stress that the corresponding SDRG is a representative of class I and so the algorithm only contains $J_a$ and $J_b$ couplings. Thus, for the IR chain, we find

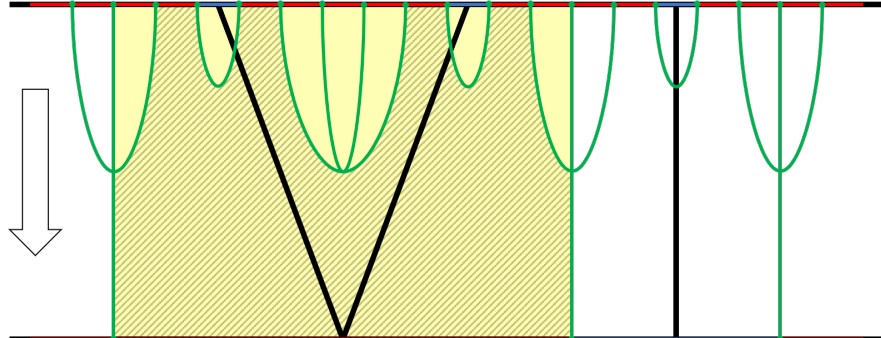

Figure 12: Stretched representation of a layer of tiles (delimited by black segments) in a $\{8, 4\}$ tiling of the Poincaré disk and the corresponding TN (green lines) introduced in Sec. 5.1. The upper and horizontal lines are called UV and IR chains, respectively. Vertices $a, b$ of the tiling, which correspond to weak and strong bonds $J_a$, $J_b$ on the spin chain according to Fig. 4, are represented in this figure as (horizontal) edges. The RG direction is indicated by the arrow. The yellow region encompasses the letter sequence that is renormalized, while the gray striped region highlights those couplings which effectively enter the RG flow algorithm given in (70)-(73).

$$J_a' = \left(\gamma_3(\Delta_a)\right) J_a \left(\frac{\gamma_2(\Delta_b)}{J_b}\right) J_a \left(\frac{\gamma_4(\Delta_a)}{J_a}\right) J_a \left(\frac{\gamma_2(\Delta_b)}{J_b}\right) J_a \left(\gamma_3(\Delta_a)\right), \tag{70}$$

$$\Delta_a' = \left(\delta_3(\Delta_a)\right) \Delta_a \left(\delta_2(\Delta_b)\right) \Delta_a \left(\delta_4(\Delta_a)\right) \Delta_a \left(\delta_2(\Delta_b)\right) \Delta_a \left(\delta_3(\Delta_b)\right), \tag{71}$$

$$J_b' = \left(\gamma_3(\Delta_a)\right) J_a \left(\frac{\gamma_2(\Delta_b)}{J_b}\right) J_a \left(\gamma_3(\Delta_a)\right), \tag{72}$$

$$\Delta_b' = \left(\delta_3(\Delta_a)\right) \Delta_a \left(\delta_2(\Delta_b)\right) \Delta_a \left(\delta_3(\Delta_b)\right). \tag{73}$$

For the convenience of the reader, we have separated the factors in (70)-(73) according to their appearance after each of the rules in the algorithm, such that it is clear where each factor comes from.

As explained in Sec. 3.2, in order to determine the complete flows of the couplings, we start from the chain with the original aperiodic $\{p, q\}$ sequence of couplings $J_i$ and homogeneous anisotropy $\Delta_0$ and repeatedly apply the algorithm a large number of times. On the other hand, given the properties of the coefficients $\gamma_n$ and $\delta_n$ discussed in Sec. 3.2 (cf. Fig. 16), whether a modulation is relevant, marginal or irrelevant can be determined without computing the entire flow, but only the change of the couplings along a single sequence-preserving transformation. In Appendix B, we present the results obtained by applying this algorithm to aperiodic XX and XXX chains defined on the boundary of $\{p, q\}$ tiling of the Poincaré disk. We consider all the pairs $\{p, q\}$, classified by the index $m$ introduced in (69), finding that all the modulations induced by $\sigma_{\{p,q\}}$ on XXX chain are relevant, while on the XX chain they are marginal. This latter result is consistent with the predictions achievable through the exact approach in [49]. Even though the critical properties of the $\{p, q\}$ aperiodic XX and XXX chains can be obtained as explained in Sec. 3.2, the approach discussed in this subsection allows for a more practical and intuitive treatment. This makes a general derivation of the results in Appendix B, for generic $p$ and $q$, more accessible.

## 5.3 Effective central charge from tensor networks

Tensor networks offer an intuitive way to study the entanglement entropy of the ground state of aperiodic XXX chains. In general, given a TN with uniform bond dimension $\chi$, the entanglement entropy of a subsystem $A$ on the boundary is bounded from above by

$$S_A \leqslant |\Lambda_A| \ln \chi \,, \tag{74}$$

where $|\Lambda_A|$ is the minimal number of tensor legs cut by a curve in the bulk. A curve that is homologous to $A$ and realizes the minimal number of cuts is called *minimal cut* and is denoted by $\Lambda_A$. The bound (74) is saturated when the tensors are perfect tensors [34]. Notice that, in order to realize the ground state of our aperiodic XXX as boundary state of the TN, the bond dimension has to be set equal to two. Nonetheless, we keep this parameter general in most of the following discussion.

We assume, for simplicity, that $A$ is a block of $L$ consecutive edges on the boundary of the tiling (or, equivalently, a block of $L + 1$ vertices). The boundary of $A$, denoted by $\partial A$, consists of the two outermost vertices of the block: these are the initial and the final points of $\Lambda_A$. In order to draw the minimal cut, consider the two vertices of $\partial A$ separately. We connect each of them to its own image vertex through the SDRG sequence-preserving transformation, in such a way that the resulting path cuts the minimal number of tensor legs. Then we iterate this process deeper and deeper in the bulk, determining two *RG trajectories* starting from the vertices in $\partial A$. At this point, there are two possibilities: either the two RG trajectories meet in an intermediate layer or they both reach the central tile. In any case, we have constructed the minimal cut $\Lambda_A$. Some examples of minimal cuts have been represented in Fig. 11 through brown dashed lines.

Notice that the RG trajectories starting at different vertices are independent. Thus, in order to give an upper bound for the entanglement entropy, we estimate the average number $\tilde{n}$ of legs cut by a single step along the RG trajectory. In other words, when starting from a generic vertex, the step will, in average, cut through $\tilde{n}$ tensor legs. For this purpose, we review the method introduced in [52] for calculating $\tilde{n}$.

We refine the classification of the vertices of a $\{p, q\}$ tiling by introducing the letters $\{a_i\}_{i \in \mathbb{N}_0}$, $\{b_i\}_{i \in \mathbb{N}_0}$. The $a$-type and the $b$-type vertices have the same properties defined in Sec. 2.2, while the subscripts $i$ identify the number of tensor legs cut by one step along the RG trajectory starting from the corresponding vertex. Given a $\{p, q\}$ tiling and the aperiodic XXX chain defined on its boundary, the allowed subscripts for $a_i$ and $b_i$ are restricted to two subsets of non-negative integer numbers $K_a$ and $K_b$. According to this fine-grained classification of vertices, the inflation rules (11) and (12) generalize to rules $\tilde{\sigma}_{\{p,q\}}$. This generalized inflation rule is constructed by considering the original words $w_a(a, b)$, $w_b(a, b)$ and replacing each $a$ and $b$ with $a_i$ and $b_i$, respectively, according to the criteria explained above. The explicit expression of these new substitution rules can be derived for all $p$ and $q$ in principle, although the constructions for class II and class III SDRG flows are too involved to report here. Later in this subsection, we shall focus on class I flows which nevertheless exhibit all the relevant properties of these new rules.

Once the refined inflation rule $\tilde{\sigma}_{\{p,q\}}$ has been introduced, we may construct the corresponding substitution matrix $\widetilde{M}$, as described in Sec. 2.2. Notice that $\widetilde{M}$ is a $(K_a + K_b) \times (K_a + K_b)$ matrix, which depends on $p$ and $q$. To lighten the notation, we drop this dependence in the following. The matrices $\widetilde{M}$ and $M$ can be decomposed into $\widetilde{M} = \tilde{A}\tilde{B}$ and $M = \tilde{B}\tilde{A}$, where $\tilde{A}$ and $\tilde{B}$ are $(K_a + K_b) \times 2$ and $2 \times (K_a + K_b)$ matrices, respectively. Thus, $\widetilde{M}$ and $M$ share the same largest eigenvalue $\lambda_+$ given in (14). The left and right eigenvectors $\tilde{\mathbf{u}}_+$ and $\tilde{\mathbf{v}}_+$ of $\widetilde{M}$ corresponding to $\lambda_+$ are constructed as $\tilde{\mathbf{u}}_+ = \mathbf{u}_+ \tilde{B}$ and $\tilde{\mathbf{v}}_+ = \tilde{A}\mathbf{v}_+$ respectively, with $\mathbf{u}_+$ and $\mathbf{v}_+$ the left and right eigenvectors of $M$ correspond-

ing to $\lambda_+$, given in (15). To study the RG trajectories, we introduce the deflation matrix $\widetilde{D}$ by exploiting these quantities, defined as

$$\widetilde{D}_{ij} = \frac{\widetilde{M}_{ij}\tilde{v}_j}{\lambda_+ \tilde{v}_i} \, . \tag{75}$$

The entry $\widetilde{D}_{ij}$ represents the probability of reaching a $j$-type vertex through a deflation step starting from a $i$-type vertex. The deflation matrix $\widetilde{D}$ has a left eigenvector $\mathbf{p}$ with eigenvalue 1, whose components are found to be

$$p_i = \tilde{u}_i \tilde{v}_i \, , \tag{76}$$

up to normalization. This is proportional to the probability of reaching an $i$-type vertex through a deflation step. To further encoding the network structure, we define the $(K_a + K_b) \times (K_a + K_b)$ entanglement matrix $\widetilde{E}$ whose component $\widetilde{E}_{ij}$ is equal to the minimal number of legs cut by any curves connecting the vertex $i$ in the UV chain and the vertex $j$ in the IR chain.

To proceed with the computation, we find it convenient to distinguish among the classes of SDRG flows introduced in (69). For $\tilde{\sigma}_{\{p,q\}}$ whose SDRG flows belong to class I, the matrix $\widetilde{D}$ directly gives the upper bound for the entanglement entropy. Indeed, in this case, the average number of legs $\tilde{n}$ is equal to

$$\tilde{n} = \frac{\sum_{ij} \widetilde{D}_{ij}\widetilde{E}_{ij}p_i}{\sum_{ij} \widetilde{D}_{ij}p_i} = \frac{\sum_{ij} \widetilde{E}_{ij}\widetilde{M}_{ij}\tilde{u}_i\tilde{v}_j}{\lambda_+ \sum_i \tilde{u}_i\tilde{v}_i} \, , \tag{77}$$

which is determined by the network structure only. According to (74), the decrease of the entanglement entropy under a sequence-preserving SDRG transformation (which, for this class of flows, corresponds to one deflation step) is bounded as

$$\Delta S_A \leqslant 2\tilde{n}\ln\chi \, . \tag{78}$$

As already mentioned in previous sections, we define the effective central charge as the coefficient of the logarithmic growth in the subsystem size of $S_A$, namely $S_A \simeq \frac{c_{\text{eff}}}{3}\ln L$. Under a sequence-preserving SDRG transformation, the subsystem size $L$ is rescaled to $L/\lambda_+$. Thus, we obtain the following upper bound for the effective central charge in $\{p,q\}$ aperiodic XXX chains

$$c_{\text{eff}} = \frac{3\Delta S_A}{\ln\lambda_+} \leqslant \frac{6\tilde{n}}{\ln\lambda_+}\ln\chi = \frac{6\sum_{ij}\widetilde{E}_{ij}\widetilde{M}_{ij}\tilde{u}_i\tilde{v}_j}{\lambda_+ \ln\lambda_+ \sum_i \tilde{u}_i\tilde{v}_i}\ln\chi \, , \tag{79}$$

where the inequality comes from (74) and therefore it is saturated when the tensors are perfect tensors [31]. A generalization of the bound (79) to classes II and III can be achieved by replacing $\left(\widetilde{M}, \lambda_+\right)$ with $\left(\widetilde{M}^2, \lambda_+^2\right)$ and $\left(\widetilde{M}^3, \lambda_+^3\right)$, respectively.

As explicit example, we focus on flows belonging to class I, where the TN states are constructed by the SDRG of $\{p,q\}$ XXX chain with even $p$ and $q \geqslant 4$. Notice that the case $\{p,3\}$ with $p$ even still belongs to class I, but, as explained in Sec. 2.2, requires a different inflation rule, which is given in Appendix B. For even $p$ and $q \geqslant 4$, the inflation rule $\tilde{\sigma}_{\{p,q\}}$ with the fine-grained classification of vertices reads

$$\tilde{\sigma}^{\mathrm{I}}_{\{2k,q\}} = \begin{cases} \left\{a_0 \mapsto (b_1 a_0)^{q-3}b_1, \ b_1 \mapsto (b_1 a_0)^{q-4}b_1\right\}, & k = 2 \\[2mm] \{a_i \mapsto a_{k-3}\cdots a_1 a_0 b_1 (a_0 a_1 \cdots a_{k-2}\cdots a_1 a_0 b_1)^{q-3}a_0 a_1 \cdots a_{k-3}, \\ \phantom{\{} b_1 \mapsto a_{k-3}\cdots a_1 a_0 b_1 (a_0 a_1 \cdots a_{k-2}\cdots a_1 a_0 b_1)^{q-4}a_0 a_1 \cdots a_{k-3}\}, & k \geqslant 3 \\[1mm] i = 0, 1, ..., k-2, \end{cases} \tag{80}$$

where $k = p/2$. The sequence-preserving SDRG transformation of class I is the inverse of (80), i.e. $\Xi = \tilde{\sigma}_{\{2k,q\}}^{-1}$. The SDRG maps the spins in the word $a_0 b_1 a_0$ to a 2-spin singlet, the spins in the word $a_0 a_1 \cdots a_{k-2} \cdots a_1 a_0$ to a $(2k-4)$-spin singlet and the spins in the word $a_0 a_1 \cdots a_{k-3} a_{k-3} \cdots a_1 a_0$ to a $(2k-5)$-spin doublet. Notice that, when $p = 6$, the SDRG only consists of 2-spin singlet, the corresponding 2-rank tensors are prefect, and the entanglement entropy saturates (74). From (80) it is straightforward to obtain the $k \times k$ substitution matrix, which reads

$$
\widetilde{M} = \begin{pmatrix}
2q-4 & \cdots & 2q-4 & 2q-6 \\
\vdots & \ddots & \vdots & \vdots \\
2q-4 & \cdots & 2q-4 & 2q-6 \\
q-3 & \cdots & q-3 & q-4 \\
q-2 & \cdots & q-2 & q-3
\end{pmatrix} ,
\tag{81}
$$

where the rows and columns are arranged in lexicographic order with respect to the fine-grained letters, i.e. $a_0, a_1, a_2, ..., a_{k-2}, b_1$. The left and right eigenvectors of (81) associated to the largest eigenvalue, given by (14), are

$$
\tilde{\mathbf{u}} = \left(1, \cdots 1, 1, x\right) ,
\tag{82}
$$

$$
\tilde{\mathbf{v}} = \left(2, \cdots 2, x, 1\right)^{\mathrm{t}} ,
\tag{83}
$$

$$
x = 2 - k + \sqrt{(k-1)\left(k - \frac{q}{q-2}\right)} .
\tag{84}
$$

Based on the fine-grained vertices introduced above, and the definition of $\widetilde{E}$, the $k \times k$ entanglement matrix is simply

$$
\widetilde{E} = \begin{pmatrix}
0 & \cdots & 0 \\
1 & \cdots & 1 \\
\vdots & \ddots & \vdots \\
k-2 & \cdots & k-2 \\
1 & \cdots & 1
\end{pmatrix} .
\tag{85}
$$

Thus, from (79) with $\chi = 2$, we obtain an inequality for effective central charge

$$
c_{\text{eff}}^{\text{Class I}} \leqslant \frac{3 \ln 2}{\ln \lambda_+} \frac{(k-1)x + (k-3)(k-2)}{(k-2) + x} .
\tag{86}
$$

Notice that, for class I SDRG flows, the tensors in the TN considered in this section have lower rank than those in [31, 52]. Thus, the entanglement matrix (85) has smaller components than the matrix $\widetilde{E}$ for the TNs of the HaPPY codes in [31] or the Majorana dimers in [52]. In turn, this means that the resulting maximal effective central charge will be smaller in our case.

Remarkably, when $p = 6$, (86) is saturated and the result (61) is recovered, providing a non-trivial cross check for the computations of Sec. 4. By properly adapting (79) as explained before, we can obtain the upper bounds for the effective central charge in some simple examples of SDRG flows belonging to Class II and Class III. In particular, the results given in Sec. 4 for modulation induced by $\sigma_{\{p,q\}}$ with $\{p,q\} = \{3,7\}, \{5,5\}, \{3,8\}, \{5,4\}$, whose TNs consist of rank-2 tensors only, have been recovered through this approach.

In Appendix C we present a detailed computation of the entanglement structure of the TN states (65) based on a numerical analysis. We provide numerical results for various

values of $n$ and check that tensors appearing in our TNs are only perfect if they are rank-2. Moreover, we show that for any of the TN states in (65), the Rényi entropies depend on the Rényi index in a way which is different from the behavior in homogeneous critical lattice models [62]. Finally, exploiting this numerical analysis, we discuss a procedure to improve the upper bound (79).

## 5.4 Symmetries of tensor network states

We now investigate the tensor network construction introduced in the previous subsections in view of two types of symmetries: internal and spatial. The former are transformations that act on the space of variables of the boundary model and do not involve the spacetime coordinates. In our case, these will be the global $SU(2)$ and $U(1)$ symmetries of the XXX and XXZ Hamiltonians, respectively. The latter refer to transformations with respect to the spatial coordinates within the TN in the bulk.

**Internal symmetries**

As reviewed in [73], within holography, the global symmetries of the boundary theory have to match with local symmetries in the bulk. Both the aperiodic XXZ Hamiltonian (22) and the homogeneous XXZ Hamiltonian (23), enjoy global $SU(2)$ symmetry for $\Delta_0 = 1$ and a global $U(1)_z$ symmetry for $\Delta_0 \neq 1$. The subscript in the latter symmetry group denotes that the $U(1)$ rotation is performed around the $z$-direction of the spin variables. Before discussing the symmetries of the entire TNs state $|\Psi\rangle$ given in (68), let us discuss the symmetries of its constituent blocks, namely the tensor states in (65) and EPR states (67). When $n$ is even, the singlet ground state $|T\rangle$ in (25) shares the same symmetries as the Hamiltonian $H_n$ in (23). When $n$ is odd, the degenerate doublet ground states $|T^\pm\rangle$ in (28) with coefficients specified by (29) enjoy the symmetry $U(1)_z$. The enlargement of the Hilbert space introduced in (65), which includes the effective spin for doublet states, allows $|T_n\rangle$ to recover the symmetries of $H_n$. Thus, we may summarize the symmetries of all the tensor states in (65) as

$$|T_n\rangle = \mathcal{G}^{(n)} |T_n\rangle, \quad \mathcal{G}^{(n)} \equiv \begin{cases} G^{\otimes n}, & \text{even } n \\ G^* \otimes G^{\otimes n}, & \text{odd } n \end{cases}, \quad G \in \begin{cases} SU(2), & \Delta_0 = 1 \\ U(1)_z, & \Delta_0 \neq 1 \end{cases}, \quad (87)$$

where $G^*$ is the conjugation of $G$ in the $|\pm\rangle$ basis. Since the tensor states $|T_n\rangle$ are localized at different points of the tensor network, the symmetries in (87) are local symmetries in the bulk and exactly match the global symmetries of the aperiodic Hamiltonian on the boundary. Moreover, we may check that the EPR state in (67) enjoys the symmetry

$$G^* \otimes G \left| bb' \right\rangle = \left| bb' \right\rangle, \quad G \in SU(2). \quad (88)$$

Finally, the top-level tensor at the center of the tiling. Let us denote it as the tensor state $\left| T_p^{\text{top}} \right\rangle$ corresponding to the ground state of $H_p$ with periodic boundary conditions. If $p$ is even, $\left| T_p^{\text{top}} \right\rangle$ is a singlet which shares the same symmetries as $H_p$. If $p$ is odd, $\left| T_p^{\text{top}} \right\rangle$ is a general linear combination of doublet states, similarly to the discussion above. In this case, the symmetries of the state depend on the precise choice of the combination. Thus, for the study of the global symmetries of the TN state $|\Psi\rangle$, we see that the choice of top-level tensor $\left| T_p^{\text{top}} \right\rangle$ plays an important role. For this reason, we make explicit the dependence of the TN state on this tensor by writing $\left| \Psi \left( \left| T_p^{\text{top}} \right\rangle \right) \right\rangle$. Exploiting the symmetries of all

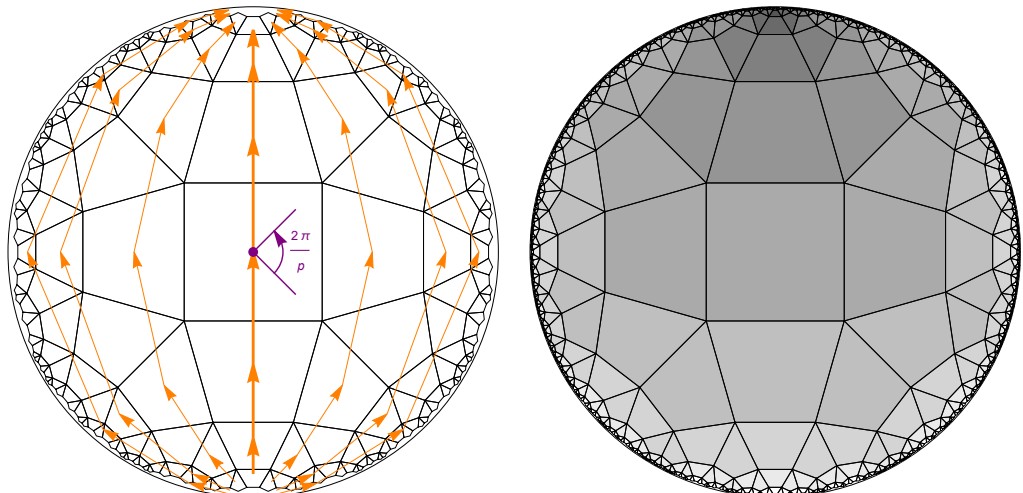

Figure 13: Left: Visualization of the action of two elements of the Fuchsian group describing the isometries of a $\{p, q\}$ tiling. The rotation by an angle $2\pi/p$ around the center of the Poincaré disk is denoted in purple. The effect of a boost of the central tile and how the rest of the tiles are transformed by this isometry is shown by the orange arrows. Right: The Poincaré foliation. The different leafs are represented by different shades of gray. This indicates an inflation direction different from the radial foliation shown in Fig. 2.

the building blocks mentioned above, we find the following relation for the action of the symmetry group of the boundary Hamiltonian on the TN state,

$$
G^{\otimes N} \left| \Psi \left( \left| T_p^{\mathrm{top}} \right\rangle \right) \right\rangle = \left( \bigotimes_{\langle bb' \rangle} \langle bb' | \, G_b^t \otimes G_{b'}^\dagger \right) \left( \bigotimes_b \mathcal{G}_b^{(n)} \left| T_n \right\rangle_b \right) = \left| \Psi \left( G^{\otimes p} \left| T_p^{\mathrm{top}} \right\rangle \right) \right\rangle , \quad (89)
$$

where $N$ is the total number of spins of the chain on the boundary, the subscripts $b$ in $\mathcal{G}_b^{(n)}$ and $G_b$ denote that they are acting on the legs of the tensor state at position $b$. Eq. (89) shows that, by exploiting the symmetries (87) and (88), the transformation $G^{\otimes N}$ acting on the boundary can propagate into the bulk until it reaches $\left| T_p^{\mathrm{top}} \right\rangle$. Thus, the symmetry of $|\Psi\rangle$ depends only on $\left| T_p^{\mathrm{top}} \right\rangle$. In particular, when $p$ is even, the tensor network state $|\Psi\rangle$ exhibits the same symmetries of the aperiodic Hamiltonian on the boundary.

**Spatial symmetries**

Motivated by the fact that the structure of the TN embedded onto the Poincaré disk is different from that of the hyperbolic tiling, we now study the spatial symmetries of the TNs derived in previous subsections that reproduce the ground state of the aperiodic XXX chain.

First, let us recall how the original symmetries of our starting point, $\mathrm{AdS}_{2+1}/\mathrm{CFT}_2$ (cf. Sec. 2.1), have been broken down through the steps in Sec. 2. The group of orientation-preserving isometries of $\mathrm{AdS}_{2+1}$ is $PSL(2, \mathbb{C})$. Since we consider a constant time slice of this space, this restricts the group of isometries to $PSL(2, \mathbb{R})$. The discretization in terms of $\{p, q\}$ hyperbolic tilings further reduces the isometries to a discrete subgroup belonging to the class known as *Fuchsian groups* [98] (for a recent discussion of Fuchsian groups in the context of hyperbolic tilings, we refer to [22, 40, 55, 99]). This subgroup contains infinitely many elements, but for our discussion we want to highlight only two. Their respective

actions on the tiling are visualized in the left panel of Fig. 13 and consist of a rotation about the center of the tiling by $\frac{2\pi}{p}$ (purple arrow), and a boost of the central tile to the center of any other tile (orange arrows).

Second, throughout this work we have constructed the tilings by means of inflation rules as explained in Sec. 2.2. We have always taken the starting point to be the tile at the center of the Poincaré disk and constructed the tiling through iterative inflation steps. This defined a so-called *discrete foliation* of the hyperbolic space, in which each inflation layer defines a *leaf* of the foliation. In particular, by starting from the central tile, we have used what we denote as a radial foliation. Thus, it specifies a particular direction in hyperbolic space in which the tiling is constructed. However, one can think of defining other discrete foliations of the Poincaré disk by changing the starting point. For instance, we can act on the tiling with a particular element of the large class of boosts in the Fuchsian group to shift the central tile to the boundary. Construction of the tilings through inflation starting from this point will then define a different discrete foliation, shown in the right panel of Fig. 13 by different shades of gray. We denote this foliation as Poincaré foliation. Indeed, in terms of continuum coordinates, this amounts to a transformation between global (3) and Poincaré coordinates (5). Note that this foliation also specifies a particular inflation direction that is different from that of the radial foliation. Let us emphasize that the result of an infinite number of inflation steps is the same in both foliations, namely the full hyperbolic tiling. However, we have shown in Sec. 5.1 that the SDRG flow is related to the construction of the tilings through inflation and will thus depend on the specific foliation we choose. Since the TN we construct in the previous subsections is an implementation of this SDRG procedure, we expect it to be sensitive to the choice of foliation as well.

Finally, in the literature, for instance in [34, 35, 37–40, 42, 51, 52], TNs are constructed and connected such that they coincide with the hyperbolic tilings and thus share their symmetries. In our case, we have an explicit Hamiltonian on the boundary and we construct the TN such that it reproduces its ground state, without assuming the tiling's symmetries. The geometric structure of the TN does not coincide with that of the tiling, featuring instead a different set of symmetries which depends on the foliation. Therefore, we are prompted to analyze the symmetries of the resulting TN and how they are affected by different choices of foliation in more detail.

Our analysis relies on the distinction between *global* and *local* symmetries of the TN. A global symmetry is defined as a transformation that preserves the entirety of the TN. This type of symmetries depend explicitly on the choice of foliation. A local symmetry is defined as a transformation that maps a subregion of the TN to another subregion while preserving its tensor structure. By their local nature, these symmetries do not depend on the chosen foliation. With these definitions, the symmetries of the TN representing the ground state of an aperiodic XXX chain are as follows:

**Global symmetries in radial foliation: rotational $\mathbb{Z}_p$ symmetry** In the radial foliation, the central $p$-gon carries the seed word $a^p$ (or $\circledast^3$ for $p = 3$). The corresponding TN thus has a central tensor of rank $p$. Therefore, the TN only inherits the global $\mathbb{Z}_p$ rotational symmetry around the central rank-$p$ tensor, as can be seen in Fig. 11.

**Global symmetries in Poincaré foliation: $\mathbb{Z}$ scaling symmetry** Under the isometry taking the central tile to the boundary, the tiling itself is left invariant, but the central rank-$p$ tensor is shifted. With this choice of a starting point for the inflation procedure, the SDRG and the corresponding TN develop in the direction of the foliation visualized in right panel of Fig. 13. By means of the boost transformation, all but one leg of the central $p$-tensor get shifted to infinity. This removes the $\mathbb{Z}_p$ symmetry present in the radial foliation. However, along the remaining leg, which runs

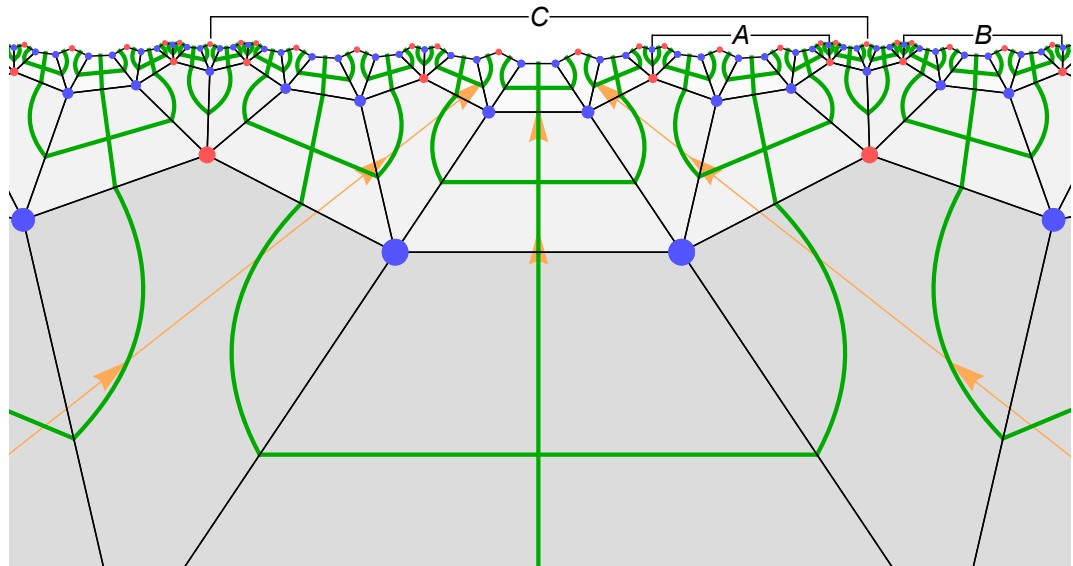

Figure 14: Tensor network reproducing the ground state of the $\{4,5\}$ aperiodic XXX chain, embedded onto the corresponding regular tiling. We zoom in on a part of the TN close to the boundary of the tiling, allowing us to leave unspecified the foliation used to define the inflation. The orange arrows are the same as in Fig. 13. The portions of tensor networks ending on the boundary subregions $A$ and $B$ are related by the local translational symmetry. Moreover, they are both related to the portion of the network ending on the boundary subregion $C$ by a local scaling transformation.

diametrically through the Poincaré disk, the TN enjoys a global $\mathbb{Z}$ scaling symmetry.

**Local symmetries** The fractal structure of the aperiodic sequences defining the modulation of the chain is inherited by the TN. Thus, any arbitrarily large substructure of the TN can be found again at infinitely many other points of the TN. On the one hand, if a given substructure can be found again in the same leaf of the foliation, this can be regarded as a local translational symmetry. A similar notion has been denoted as "approximate" or "quasi" translational symmetry in [40]. On the other hand, if a tensor substructure can be found in a different leaf, this represents a local scaling symmetry. Both these local symmetries, examples of which are shown in Fig. 14, are a consequence of the self-similarity properties of aperiodic sequences.

In this section we have constructed a TN which reproduces the ground state of an aperiodic XXX chain with modulation generated by $\sigma_{\{p,q\}}$. This has been done by exploiting the SDRG approach reviewed in Sec. 3.2. In particular, we have associated a tensor to the decimation of each spin-block. This has led to the building blocks of the tensor network (65), sketched in Fig. 10 for the decimation of 2-spin and 3-spin blocks. Placing and implementing these fundamental tensors along the whole UV chain is equivalent to realizing a single sequence preserving SDRG transformation: by iterating the process, we have constructed the entire tensor network which has the ground state of the original aperiodic XXX chain as its UV state. Such a tensor network, which is defined independently of the discrete bulk geometry determining the inflation rule, can be nevertheless embedded onto the corresponding $\{p,q\}$ tiling. This provides a discrete structure in the bulk, which is different from the one of the regular $\{p,q\}$ tiling of the Poincaré disk (see Fig. 11 for a comparison between two of these tensor networks and their corresponding $\{p,q\}$ tilings). As an application, in Sec. 5.2 we have exploited these tensor networks to determine an

efficient algorithm for computing the SDRG flow of the couplings (33) in aperiodic XXZ chains. This approach has been applied in a thorough analysis reported in Appendix B for the modulations generated by any pair $\{p, q\}$. We would like to highlight that the TN we construct, when embedded onto the discrete AdS space given by the $\{p, q\}$ tilings, describes an RG flow of the couplings from the UV (boundary) to the IR (center of the tiling). This is consistent with the holographic RG flows known from AdS/CFT [73, 100, 101], as opposed to the setup in [41]. Moreover, adapting the computation developed in [52] to our tensor networks, we have provided the general expression (79) for an upper bound on the effective central charge defined in Sec. 3.2 for the boundary aperiodic XXX chain. We have applied this formula to those modulations whose SDRG flows belong to the class I (see the definition in (69)), obtaining the explicit expression (86) for the maximal effective central charge. Finally, we have discussed the spatial symmetries of the TN graph constructed in this section, observing that they depend on the discrete foliation we choose for defining the inflation procedure on the discretized Poincaré disk. Furthermore we have compared these spatial symmetries with the ones of the corresponding regular $\{p, q\}$ tiling, finding that the former are a subgroup of the latter. This can be explained by noticing that the foliations explicitly break the Fuchsian group symmetry of the whole tiling. In spite of the aforementioned mismatch between the spatial symmetries, in (87) and (88) we find that the global symmetries of the boundary Hamiltonian are indeed captured by the TN states (65) localized in the bulk.

# 6 Comparison of results for different effective central charges

The results provided in the previous sections allow us to draw comparisons in order to better understand the underlying setups and the consequences of the discretization procedures. In particular, we will compare the coefficients of the logarithmic growth of the entanglement, i.e. the effective central charges, computed in Sec. 2.3, Sec. 4 and Sec. 5.3 of this work with those of [52]. Our comparison includes some key points to connect and differentiate our analysis from the one in [52].

The main result of Sec. 2.3 provides the effective central charge (21) purely as a function of the geometric Schläfli parameters $p$ and $q$ tied to the hyperbolic tilings. For a given continuous geodesic $\gamma_A$, $c_{\text{eff,bulk}}$ in (21) quantifies the length of the corresponding discrete path $\Gamma_A$ in units of polygon edge lengths $s(p, q)$ (8). In contrast, the setup introduced in [52] implements the logarithm of the bond dimension, $\ln \chi$, as a "length" unit. This is well motivated from the point of view of the TN construction used in [52], though it is not a default quantity defined on any $\{p, q\}$ tiling. Particularly useful to help us understand the difference in these two constructions is the Brown-Henneaux formula $c = \frac{3R}{2G_N}$ describing the central charge of the underlying CFT. On the one hand, in our construction, the formula is assumed to remain true and unaffected after the discretization. In other words, we assume that if the discretization has any effect on the underlying CFT, this will manifest itself as a tiling-dependent multiplicative factor $c_{\text{eff,bulk}} = c \cdot f(p, q)$ instead of a direct dependence of the central charge on the Schläfli parameters. On the other hand, the TN used in [52] assumes a dependence of Newton's constant on the tiling parameters, something that underlies the fact that the bond dimension is fixed to be $\ln \chi = \frac{s(p,q)}{4G_N(p,q)}$. If we assume this identification in order to make contact with our expression of $c_{\text{eff,bulk}}$ in Sec. 2.3, both approaches are equivalent up to a constant numerical prefactor. Even though the effective central charges reported in [52] are maximal with respect to the type of tensors used, the prefactors of our computations are sometimes larger and sometimes smaller than those in [52]. We can trace back this behavior to an averaging procedure introduced in [52]. Our

construction does not consider such averaging, since the orientation of the geodesic is fixed *a priori*. Fluctuations of the numerical prefactor around an average value are thus to be expected. Nevertheless, we emphasize that our construction is in some sense closer to the continuous setup, since it only requires a discretization procedure, without the immediate need to associate a TN to the tiling. Such a construction indeed proves useful *a posteriori* as explained in Sec. 5, but leads to a different bulk TN.

Let us now extend this comparison to include the results obtained in Sec.4 and Sec.5 for the aperiodic XXX chain in its ground state. In Sec. 5.3, exploiting the structure of the TN constructed in Sec. 5.1, we have obtained an upper bound for the effective central charge of some aperiodic XXX chains. In particular, we have focused on chains with modulations generated by those $\sigma_{\{p,q\}}$ whose corresponding SDRG flows belong to the class I, according to the definition (69). The upper bound we derive in (86) does depend on $p$ and $q$ and is saturated in the case of $\{6, q\}$ modulations along XXX chains, where the result (61) of Sec. 4.2 is recovered. The bound (86) has been obtained by employing (79) for the TN considered in this manuscript. The formula in (79) was introduced in [52], where it has been applied to the TN built by identifying each $p$-gon of a $\{p, q\}$ tiling with a tensor. When employed for different networks, (79) provides different bounds and therefore (86) is different from the maximal central charge reported in [52]. This fact can be also seen in the following perspective: since the TNs considered in this manuscript and in [52] are different, the corresponding boundary states are not the same, nor should we expect their entanglement structure to be equal. In spite of this, one could try to draw qualitative comparisons between the effective central charges to gain a birds-eye view of how the Schläfli parameters might influence this object in general. Notice that, for a consistent comparison, the bond dimension $\chi$ in the results of [52] has to be set equal to 2, given that our TN is characterized by a two-dimensional local Hilbert space. Restricting the analyses to those $\sigma_{\{p,q\}}$ with a SDRG flow belonging to class I ($p = 4$, $q \geqslant 5$ and even $p \geqslant 6$, $q \geqslant 4$), we observe that the maximal central charge of [52] is always greater than the one reported in (86). Moreover, once we fix an even value of $p$, the two upper bounds are decreasing as functions of the allowed $q$ (except for the unique case with $p = 6$ and $q = 4$) and are both vanishing in the limit $q \to \infty$. Let us stress again that, despite these qualitative similarities for the behavior of the maximal central charges as function of $q$, the fact that their explicit values are different is totally justified given that the boundary states of the two corresponding tensor networks are not the same.

Finally, we find it interesting to compare the upper bounds reported in (86) with the (proper) central charge of the CFT underlying the homogeneous XXX chain, namely $c = 1$. Notice that the upper bound in (86) can be either smaller or greater than one, depending on the values of $p$ and $q$. Moreover, in the particular regime where $q \to \infty$, the upper bound approaches to zero and and we find that $c = 1$ is larger than $c_{\text{eff}}^{\text{Class I}}$. This is not a contradiction given that $c = 1$ characterizes the critical behavior of the homogeneous model, while $c_{\text{eff}}^{\text{Class I}}$ holds as upper bound in the regime of validity of the SDRG, namely for strong aperiodicities. Given that these two regimes do not overlap, we should not expect any direct relation between $c$ and $c_{\text{eff}}^{\text{Class I}}$.

# 7 Discussion and outlook

In this paper, we present a first step towards a discrete holographic duality involving a dynamical theory on the boundary. Our argument is visualized by the diagram in Fig. 15. As an exemplary quantity benchmarking the different scenarios represented in Fig. 15, we study bipartite entanglement entropy. The starting point is the discretization scheme we

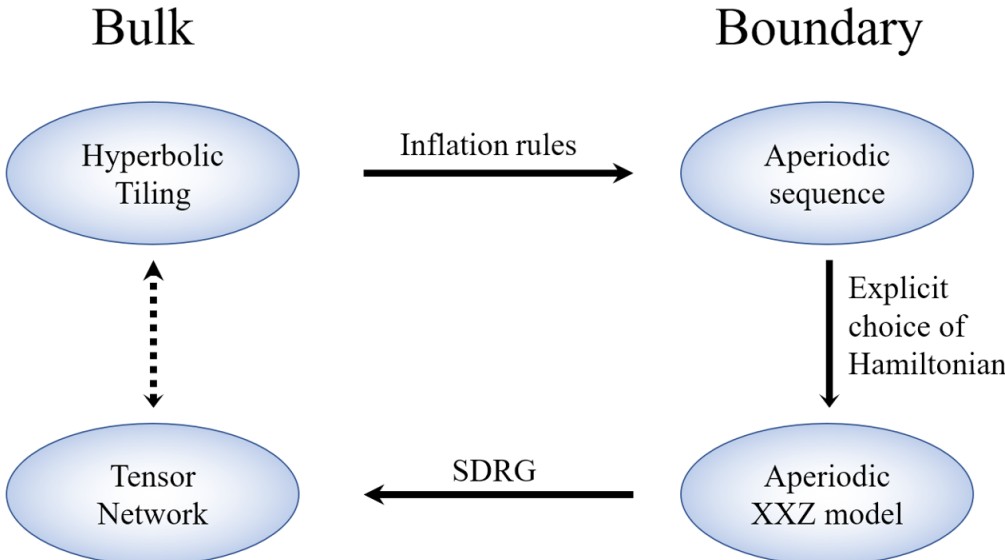

Figure 15: Diagrammatic summary of the relations between hyperbolic tilings, aperiodic spin chains and tensor networks found in this work. A detailed explanation of the individual ingredients of this diagram is provided in the main text.

choose for a constant time slice of $AdS_{2+1}$, which we identify with the Poincaré disk $\mathbb{D}^2$. We implement regular hyperbolic tilings, uniquely characterized by their Schläfli symbol $\{p, q\}$, to canonically discretize $\mathbb{D}^2$. From simple geometric arguments in the bulk of the tiling, we derive a discrete version of the RT formula (17). This allows us to compute the entanglement entropy (20) of a boundary region $A$, which we find to scale logarithmically in the subsystem size $L$. Importantly, we find the coefficient of this logarithmic growth, which we denote as the effective central charge $c_{\text{eff,bulk}}$, to depend in a non-trivial way (21) on $p$ and $q$.

A $\{p, q\}$ tiling may be constructed through an associated inflation rule $\sigma_{\{p,q\}}$. From this inflation procedure, an aperiodic structure arises on the boundary of the tiling after a large number of inflation steps, as visualized in the top right corner of Fig. 15. This structure is characterized by an infinitely long aperiodic letter sequence. Motivated by the aperiodic structure of the asymptotic letter sequence on the tiling's boundary, we expect a potential holographic dual to the discretized bulk to also exhibit some notion of aperiodicity. Moreover, the discrete nature of our setup motivates the choice of a quantum spin chain as a promising candidate for the boundary theory. We thus explicitly define an XXZ quantum spin chain with aperiodically modulated couplings (22) as a boundary theory, shown on the right hand side of Fig. 15. The aperiodicity of this model is induced by the asymptotic letter sequence generated by the inflation rule $\sigma_{\{p,q\}}$. The gapless, interactive nature of this theory for anisotropies $0 \leqslant \Delta_0 \leqslant 1$ makes it a more interesting choice above other quantum spin chains.

We employ SDRG techniques to study the critical properties of this model. In the special case of the aperiodic XXX chain, we find aperiodic $\{p, q\}$ modulations to be relevant for all $p$ and $q$. This means that the aperiodic model flows to a new fixed point with respect to the homogeneous case, induced by strong disorder. Furthermore, we compute the entanglement entropy in aperiodic XXX chains for several choices of $\{p, q\}$ modulations, for which the ground state of the model is found to be in an aperiodic singlet phase. This includes a generalization of previous SDRG techniques [74] to 2-cycle bond-distribution attractors under SDRG. We find the piecewiese linear growth (57) of the entanglement entropy. The

envelope of this functional dependence is a logarithm with a $\{p, q\}$-dependent coefficient, interpreted as the effective central charge $c_{\text{eff}}$. This indicates the effect of the discretization on the entanglement structure of the boundary theory.

Finally, due to their computational advantages and their proven relation to holography, we construct the tensor networks represented in the left bottom part of Fig. 15. These are based on the SDRG and reproduce the ground state of aperiodic XXX chains. The tensor network allows us to not only cross-check the results derived in this work for the aperiodic XXX chain, but also to extend them and fully characterize the ground state of this aperiodic model. Moreover, the natural holographic structure of tensor networks provides a canonical way of extending into the bulk when they are embedded onto $\{p, q\}$ tilings. In Sec. 5.2, we provide an efficient algorithm to compute the flows of the couplings of the chain along the SDRG. Moreover, we find an upper bound for the effective central charges as a function of $p$ and $q$, also in cases where the SDRG involves doublet states. The embedding of our TNs onto the Poincaré disk reveals that their *spatial* symmetries are however different from those of the corresponding $\{p, q\}$ tilings, cf. Sec. 5.4. This is depicted by the dashed arrow in Fig. 15. This difference can be traced back to the choice of a discrete foliation for the tilings. On the other hand, we explain how the *internal* symmetries of the boundary Hamiltonian can also be found in the tensor states that constitute the TN, as explained in Sec. 5.4. In particular, these can be regarded as local bulk symmetries within the discrete geometry provided by the TN. This is reminiscent of the known relation between global boundary symmetries and local bulk symmetries from standard continuum holography.

For eventually obtaining a complete holographic duality, it appears necessary to further investigate the mismatch of the spatial symmetries between tilings and TN. A holographic duality would require the boundary fields to transform covariantly under the Fuchsian group given by the hyperbolic tiling. This, to the best of our knowledge, has not yet been achieved and constitutes a line for future studies. Based on our analysis in Sec. 5.4, we believe that the matching of global to local symmetries in the TN picture is a promising indicator that such a correspondence can be potentially realized.

Let us conclude by commenting upon several avenues for future developments of the results reported in this manuscript.

In this work, we have proposed an aperiodic chain with $SU(2)$ spins at each site as a boundary theory. The nature of the spin variables can be generalized, for instance considering $SU(N)$ spins. This is interesting in view of spin chains with $SU(N)$ global symmetry, whose continuum limit is provided by $SU(N)$ Wess-Zumino-Witten CFTs [102–106]. The central charge of this theory diverges linearly in $N$ when $N \to \infty$. Note that the $SU(N)$ symmetry in this case is global rather than local, meaning that its large $N$ limit is more similar to the vector large $N$ limit than to the matrix large $N$ limit usually considered in holography [107]. It is more akin to large $N$ limit of $O(N)$ models of quantum field theories, where fields transform in the fundamental representation of the symmetry group, and the number of degrees of freedom $N$ is taken to infinity. Interestingly, holographic duals for such models have been proposed in the context of higher-spin gravity theories [108–111] such as Vasiliev gravity [112–114]. It would thus be interesting to study aperiodic $SU(N)$ spin chains in this large $N$ regime. In particular, we aim to understand whether the effective central charge defined as in Sec. 4.1 also grows with $N$ for fixed pairs $\{p, q\}$.

It is known that the conformal spectrum of the homogeneous XXZ chain in the continuum limit contains the spectra of various models with central charge less than one, including *minimal models* [115, 116]. Understanding whether fingerprints of this feature remain in presence of aperiodicity would be an interesting future development. In view of

holography, we note that gravity duals for minimal models in Euclidean spacetime have actually been proposed in [117].

Furthermore, in standard continuum holography, the fields of the boundary CFT transform covariantly under representations of the conformal group, which is isomorphic to the isometry group of the bulk spacetime. Thus, in order to establish a field-operator map between discrete theories, a challenging but promising goal for the future is to translate this argument to our discrete setup, where the isometries of the bulk are given by Fuchsian groups.

As discussed in detail throughout this work, introducing an aperiodic modulation on a homogeneous critical XXZ spin chain can be seen as an RG flow triggered by a relevant deformation of the couplings. This flow interpolates between the homogeneous model in the UV and the aperiodic XXZ chain in the IR. In this picture, the continuum limit of the model at the UV fixed point enjoys conformal symmetry, being a CFT with a central charge $c = 1$. On the other hand, the IR fixed point is characterized by a critical theory, whose aperiodicity explicitly breaks the conformal symmetry. This is similar to the symmetry breaking along RG flows in continuous holographic models such as $\text{AdS}_{d+1}/\text{Lifshitz}_{d+1}$ domain wall [118] or the $\text{AdS}_{d+1}/\text{AdS}_2 \times R_{d-1}$ domain wall (extremal RN black hole) [119]. Understanding the full extent of this analogy an how it might provide insight into a discrete holographic duality presents an interesting goal which we leave for future work.

A further step towards a complete discrete holographic duality requires the construction of a dual bulk theory. For this purpose, the interplay between SDRG and tensor networks discussed in Sec. 5 provides a promising starting point. More precisely, the graph of the TN embedded onto the Poincaré disk can be viewed as a discrete space in the bulk. Based on this, we can define a bulk theory by promoting the couplings $J_i$ defined on the vertices $x_i$ of the TN to background bulk fields $J(x_i)$. The RG flows (27) and (31) can be interpreted as the global radial evolution of the background fields in this discrete space. Given the local nature of the SDRG, we might generalize the global radial evolution to local equations of motion by considering a perturbation $J(x_i) + \delta J(x_i)$ and studying its flow under the SDRG. We leave an investigation into the dynamics of this potential bulk theory for future work.

A further interesting perspective is the investigation of how modifications of the TN influence the corresponding boundary theory. First, we can introduce a TN version of a conical defect into the discrete TN graph mentioned above. This can be done by choosing a different "top-level tensor" of rank $p' \neq p$ described by a new seed word $a^{p'}$. In the UV, the resulting TN state will have the same fine-grained structure as the ground state of the $\{p, q\}$ XXX chain (see Sec. 5.1). It would be interesting to understand how this conical defect can deform the minimal cuts and thus affect the entanglement properties of the TN [38, 120]. Second, it would be insightful to consider excited states, thermal states or more general mixed states of our proposed boundary Hamiltonian (22). For instance, a low-lying excited state of the boundary aperiodic chain could be constructed through the SDRG method by replacing the ground state of one of the local Hamiltonians $H_n$ in (23) with a higher energy eigenstate. The case of a thermal state is more challenging, since the SDRG procedure is no longer valid due to the thermal correlations present in the local ground state, which prevent us from individually decimating strongly-coupled spin blocks. Establishing a more general framework allowing for the description of aperiodic spin chains at finite temperature is a compelling goal which we will pursue in the future.

We would also like to comment upon the connection between hyperbolic tilings and the discretization scheme employed in the **p**-adic AdS/CFT [23–25]. This approach to discrete holography considers the so-called *Bruhat-Tits tree* to be the discrete bulk dual to a theory defined on the field $\mathbb{Q}_\mathbf{p}$, with **p** prime, of **p**-adic numbers (we use bold font to distinguish

between the parameter $\mathbf{p}$ in the $\mathbf{p}$-adic setting and the Schläfli parameter $p$ from hyperbolic tilings). The Bruhat-Tits tree is a $(\mathbf{p}+1)$-valent tree equivalent to a Bethe lattice and can be formally realized as the $p \to \infty$ limit of $\{p, q\}$ tilings with $q = \mathbf{p} + 1$. Currently, our construction exhibits no clear sensitivity to prime values of the Schläfli parameters, other than a complicated structure for the corresponding TN. It would be interesting to access the $p \to \infty$ limit explicitly and possibly obtain results that could be compared to those found in the context of $\mathbf{p}$-adic AdS/CFT. This would serve as a non-trivial consistency check between different approaches to discrete holography.

Finally, in order to include the dynamics of AdS gravity into our approach, it is necessary to extend the hyperbolic tessellation of the time slice into the time direction and make the resulting triangulation of AdS$_3$ dynamical. This can in principle be achieved within the framework of causal dynamical triangulations [121, 122], which simulates precisely the Lorentzian gravitational path integral for such dynamical triangulations. Within this approach, particular time-dependent problems such as the time domain of holographic correlation functions, the dynamics of local and global quenches, and the formation of black holes, can be solved in the context of discrete gravity on hyperbolic tilings. We leave the investigation of discrete dynamical AdS gravity on hyperbolic tessellations using causal dynamical triangulations for future work.

### Acknowledgments

We are grateful to Rathindra Nath Das, Emmanuel Floratos, Haye Hinrichsen and Ronny Thomale for useful discussions. We acknowledge support by the Deutsche Forschungsgemeinschaft (DFG, German Research Foundation) under Germany's Excellence Strategy through the Würzburg-Dresden Cluster of Excellence on Complexity and Topology in Quantum Matter - ct.qmat (EXC 2147, project-id 390858490). The work of J.E., R.M. and Z-Y.X. is also supported by the Collaborative Research Centre SFB 1170 'ToCoTronics', project-id 258499086. Z-Y.X. also acknowledges the support from the National Natural Science Foundation of China under Grants No. 11875053 and No. 12075298.

## A    Length rescaling towards the boundary of hyperbolic tilings

Given a homogeneous non-aperiodic chain, the number of sites $N$ is given by $N = \mathcal{L}/a$, with $\mathcal{L}$ the total length of the system and $a$ the lattice spacing. In the setup of Sec. 2.3, this lattice model was assumed to underlie a CFT defined on a circle of the same length $\mathcal{L}$. However, the relation $N = \mathcal{L}/a$ does not hold in our boundary construction due the fractal structure of the aperiodic lattices. The correct relation is described by the *fractal dimension $d$* of $\{p, q\}$ tilings [123], which we derive in the following from hyperbolic geometry arguments.

The fractal dimension $d$ relates the effective number of sites $N$ on the whole boundary to the total circumference of the circle $\mathcal{L}$ via

$$N \sim \left(\frac{\mathcal{L}}{a}\right)^d. \tag{90}$$

On the one hand, after a large number $n$ of inflation steps, we may equate the circumference $C$ of a hyperbolic circle of radius $\tilde{R}$ centered at the origin of $\mathbb{D}^2$ with the circumference of the CFT circle $\mathcal{L}$

$$\frac{\mathcal{L}}{a} = C = \pi \sinh\left(\tilde{R}\right) \approx \pi e^{\tilde{R}}. \tag{91}$$

where we introduce the UV cutoff $a$ of the CFT to capture the divergence of the circumference for large radii. All lengths in this derivation are implicitly given in units of the AdS radius. For a hyperbolic tiling, we know that the radius will be given in terms of the number of patterns and their tiling-dependent geodesic length, i.e. $\tilde{R} = n\,t(p,q)$. Thus, plugging this into (91) and solving for $n$ we find

$$n = \frac{\ln \frac{\mathcal{L}}{a}}{t(p,q)}\,. \tag{92}$$

On the other hand, the number of vertices $N$ on the tiling's boundary is asymptotically well approximated by $N \approx \lambda_+(p,q)^n$, with $\lambda_+(p,q)$ the largest eigenvalue of the associated substitution matrix $M_{\{p,q\}}$, which implies

$$n = \frac{\ln N}{\ln \lambda_+(p,q)}\,. \tag{93}$$

Equating (93) with (92), we find the correct relation between the number of sites on the boundary and its length

$$\ln(N) = \frac{\ln \lambda_+(p,q)}{t(p,q)} \ln \frac{\mathcal{L}}{a}\,. \tag{94}$$

Naturally, this also holds for a sub-region of the circle

$$\ln(L) = \frac{\ln \lambda_+(p,q)}{t(p,q)} \ln \frac{\ell}{a}\,, \tag{95}$$

where $L$ denotes the number lattice sites on the region $\ell$. From this derivation, comparing (90) with (95), we find that the fractal dimension of the boundary of $\{p,q\}$ tilings is $d = \frac{\ln \lambda_+(p,q)}{t(p,q)}$. The fractal dimension allows us to properly relate lengths in the bulk to lengths on the boundary, which we exploit in our derivations in Sec. 2. Note that the cutoff scale in section 2.3 was denoted as $\epsilon$ and was associated to a radial cutoff in the bulk. In this Appendix, we have kept the underlying cutoff scale $a$ of the boundary theory general. The two cutoff need not be equal, but are related only by a rescaling, which would simply contribute a sub-leading constant term to the universal logarithmic behavior in (95). For this reason, we can insert (95) into (19) to derive the final result in (20).

# B    SDRG flows in generic aperiodic XXZ chains

In this appendix we provide a detailed list of results on the application of SDRG techniques to the aperiodic XXZ chains. The analysis has been performed by considering the modulations generated by $\sigma_{\{p,q\}}$ for any pair $\{p,q\}$. These findings include the classification of all the $\{p,q\}$ modulations according to the criterion discussed around (69), the explicit expressions for all the sequence-preserving transformations associated to their SDRG flows, and the flows of the coupling ratio under SDRG in the case of aperiodic XX and XXX chains.

## B.1    The classifications of SDRG flows

As discussed in Sec. 3.2, the sequence-preserving transformations $\Xi^{\{p,q\}}$ associated to the asymptotic aperiodic sequences generated by $\sigma_{\{p,q\}}$ can be classified according to the number of individual RG steps they are made up of. In particular, when

$$\Xi^{\{p,q\}} = \mathrm{RG}_k \mathrm{RG}_{k-1} \cdots \mathrm{RG}_1\,, \tag{96}$$

we say that the corresponding bond distribution attractor is a $k$-cycle under SDRG [91].

This classification does not involve explicitly the inflation rule $\sigma_{\{p,q\}}$. Moreover, it is not unique since consecutive RG steps in (96) can be combined to give more complicated transformations or single RG steps can be decomposed into more elementary transformations, resulting, in both cases, in a change of the values of the index $k$ labeling the cycle. For these reasons, in Sec. 5.1 the classification of $\Xi^{\{p,q\}}$ *via* the exponent $m$ defined in (69) has been introduced. This exponent, which is unique, quantifies the number of inverse inflation steps $\sigma_{\{p,q\}}^{-1}$ (deflations) that are required for implementing $\Xi^{\{p,q\}}$. Exploiting the explicit expressions of the sequence-preserving SDRG transformations reported in Appendices B.2-B.4 for all the pairs $\{p,q\}$, one can straightforwardly verify that the SDRG flows associated to the $\{p,q\}$ modulations are labeled by one of three integer values $m = 1, 2, 3$. This fact has the remarkable implication that each tensor network constructed in Sec. 5.1 can be placed on the corresponding tiling in a commensurable way. In the context of this classification, the SDRG flows associated to $\sigma_{\{p,3\}}$, for $p \geqslant 7$ require an ad hoc analysis. As explained in Sec. 2.2, the inflation rules $\sigma_{\{p,3\}}$ are peculiar since are usually written in terms of three letters $a, b, c$ [52]. Since the techniques we have exploited in this manuscript are valid for binary inflation rules, we would like to recast $\sigma_{\{p,3\}}$ in a form involving two letters $a, b$ only. This can be done at the price of introducing letters with negative exponents in the inflation rule, *i.e.* $a^{-1}$. Examples of these rules, which are not considered in the standard literature on aperiodic spin chains [47–50,58,91], are given in (102) and (126). Our goal is to understand what happens to the SDRG procedure in XXZ chains in presence of this kind of aperiodicities. The key observations are that the sequence-preserving transformation $\Xi^{\{p,3\}}$ does not contain letters with negative exponents and it is related to the inverse inflation rule $\sigma_{\{p,3\}}^{-1}$ as

$$\Xi^{\{p,3\}}\mathrm{RG}_0 \sim \mathrm{RG}_0\sigma_{\{p,3\}}^{-m}, \tag{97}$$

where $\mathrm{RG}_0$ is an auxiliary RG step and $m$ is an integer number. We generalize the classification by the index $m$ in (69) to (97), by saying that $\Xi^{\{p,3\}}$ belongs to class I, II or III, if $m = 1, 2$ or $3$ respectively. The relation (97) implies that the SDRG associated to $\{p, 3\}$ modulations is not affected by $\mathrm{RG}_0$ and it can thus be treated as the other cases of $\{p, q\}$.

One of the main results reported in this appendix is the flow of the coupling ratio $r$ under SDRG when a generic aperiodic $\{p, q\}$ modulation is applied either to an XX ($\Delta_0 = 0$) or to an XXX ($\Delta_0 = 1$) chain. We can restrict ourselves to these two cases without loss of generality because, as explained in Sec. 3.2, for any $0 < \Delta_0 < 1$ the aperiodic XXZ chain flows to the XX chain under SDRG. After one sequence-preserving transformation, the coupling ratio $r$ becomes $r'$ such that

$$r'/r = \Lambda, \tag{98}$$

with $\Lambda$ independent of the hopping parameters. Crucially, the SDRG analysis for aperiodic XX and XXX chains (see Sec. 3.2) reveals that the anisotropy parameters do not flow in either of these cases, since $\Delta_0 = \Delta_0^* = 0$ and $\Delta_0 = \Delta_0^* = 1$, respectively. As a consequence, $\Lambda$ does not change along the SDRG flow either. Thus, if $\Lambda = 1$, $r$ is constant along the SDRG flow and the corresponding modulation is marginal, while, if $\Lambda < 1$, $r \to r^* = 0$ along the SDRG and the corresponding modulation is relevant. In the following subsections we find the expression for $\Lambda$, for any $\{p, q\}$, as products of various coefficients $\gamma_n$ introduced in (27) and (31). Remarkably, in these products, the indices $n$ of the various $\gamma$ factors are always even. From a numerical analysis, whose results are shown in Fig. 16 for some values of $n$, we observe that $\gamma_n(\Delta_0 = 1) < 1$ for any $n$ and therefore all the $\{p, q\}$ modulations are relevant when applied to the XXX chain. On the other hand, when $n$

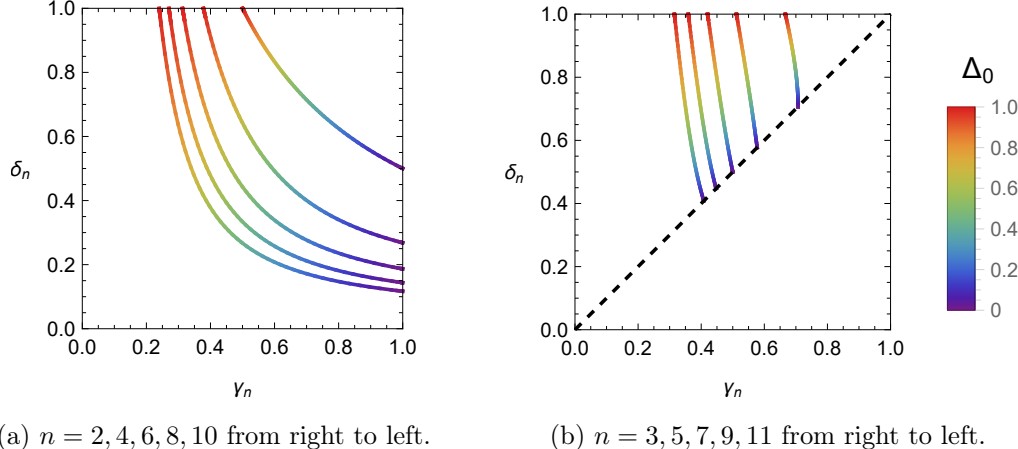

(a) $n = 2, 4, 6, 8, 10$ from right to left.      (b) $n = 3, 5, 7, 9, 11$ from right to left.

Figure 16: Coefficients $\gamma_n$ and $\delta_n$ defined in (27) and (31) shown as functions of $\Delta_0$, for various values of $n$. The black dashed line in the right panel corresponds to $\gamma_n = \delta_n$. The numerical data show that these coefficients are always smaller than 1 in the whole range of $\Delta_0$ considered here.

is even, $\gamma_n(\Delta_0 = 0) = 1$ and therefore all the $\{p, q\}$ modulations are marginal when applied to the XX chain. This latter statement is consistent with the results obtained by applying the exact methods developed in [49] on the $\{p, q\}$ modulations.

In the following subsections we consider the three classes of SDRG flows separately and we collect the results which allowed us to draw the aforementioned conclusions about the nature of the $\{p, q\}$ modulations in XX and XXX chains.

## B.2   Class I: $\{p, q\}$ XXZ chain for even $p$

The modulations in class I are associated to sequence-preserving transformations $\Xi^{\{p,q\}}$, fulfilling (69) (or (97)) with $m = 1$.

**$\{p, q\}$ tiling for even $p$ and $q \geqslant 4$**

The sequence-preserving SDRG transformation can be written in terms of the following RG steps

$$\mathrm{RG}_1 = \{aba \mapsto a, \ a \mapsto b\}, \tag{99}$$

$$\mathrm{RG}_2 = \left\{ b^{(p-6)/2}(ab^{p-5})^{q-3}ab^{(p-6)/2} \mapsto a, \ b^{(p-6)/2}(ab^{p-5})^{q-4}ab^{(p-6)/2} \mapsto b \right\}. \tag{100}$$

It might seem odd that some powers of letters in (99) and (100) can be negative for some choices of $p$, as for instance for $p = 4$. This is however not problematic, since even if this occurs for the individual RG steps, this issue disappears once we combine them for constructing the sequence-preserving transformation $\Xi^{\{p,q\}} = \mathrm{RG}_2\mathrm{RG}_1$. Taking the case $\{4, q\}$ for $q \geqslant 5$ as an example, we have $\mathrm{RG}_2 = \{b^{-1}(ab)^{q-3}ab^{-1} \mapsto a, \ b^{-1}(ab)^{q-4}ab^{-1} \mapsto b\}$, but the sequence-preserving transformation $\mathrm{RG}_2\mathrm{RG}_1 = \{(ba)^{q-3}b \mapsto a, \ (ba)^{q-4}b \mapsto b\}$ does not contain negative exponents for the letters. The same remark is valid for all the other pairs $\{p, q\}$ considered in this appendix. It is straightforward to check that $\Xi^{\{p,q\}} = \mathrm{RG}_2\mathrm{RG}_1$ is equivalent to a single deflation step $\sigma_{\{p,q\}}^{-1}$.

Exploiting the algorithm reported in Sec. 5.2, we can write down how the coupling ratio $r$ is modified after one sequence-preserving transformation, namely

$$r'/r = \gamma_2\gamma_{p-4}. \tag{101}$$

Notice that, since $p$ is even, $p - 4$ is also even. Thus, (101) guarantees the marginality and the relevance of the modulations belonging to this class when applied to XX and XXX chains respectively. Similar conclusions can be drawn at the end of all the following subsections, but we will not write them down explicitly for conciseness.

### $\{p, 3\}$ tiling for even $p \geqslant 8$

This case belong to the special class of modulations discussed in Appendix B.1. By considering also negative exponents for the letters, the inflation rule $\sigma_{\{p,3\}}$ can be written as

$$\sigma_{\{p,3\}} = \left\{ a \mapsto aa^{(p-5)/2}ba^{(p-5)/2}, \ b \mapsto a^{-1} \right\}. \tag{102}$$

The RG step

$$\mathrm{RG}_1 = \left\{ b^{(p-6)/2}ab^{p-7}ab^{(p-6)/2} \mapsto a, \ b^{(p-6)/2}ab^{(p-6)/2} \mapsto b \right\}, \tag{103}$$

combined with the auxiliary transformation

$$\mathrm{RG}_0 = \{aba \mapsto a, \ a \mapsto b\}, \tag{104}$$

leads to the relation

$$\mathrm{RG}_1\mathrm{RG}_0 \sim \mathrm{RG}_0\sigma_{\{p,3\}}^{-1}. \tag{105}$$

Comparing (105) with (97) evaluated for $m = 1$, we can identify $\Xi^{\{p,3\}} = \mathrm{RG}_1$.

The SDRG flow of the coupling ratio can be obtained in this case by $\mathrm{RG}_1$ only and reads

$$r'/r = \gamma_{p-6}, \tag{106}$$

where we notice that $p - 6$ is even for any even $p$.

## B.3   Class II: $\{p, q\}$ XXZ chain for odd $p$ and even $q$

The modulations belonging to this class are associated to sequence-preserving transformations $\Xi^{\{p,q\}}$ satisfying (69) with $m = 2$.

### $\{3, q\}$ tiling for even $q \geqslant 8$

The RG steps are

$$\mathrm{RG}_1 = \{aba \mapsto a, \ a \mapsto b\}, \tag{107}$$

$$\mathrm{RG}_2 = \left\{ b^{(q-6)/2}(ab^{q-7})^{q-5}ab^{(q-6)/2} \mapsto a, \ b^{(q-6)/2}(ab^{q-7})^{q-6}ab^{(q-6)/2} \mapsto b \right\}, \tag{108}$$

and contribute to determine the sequence-preserving transformation $\Xi^{\{3,q\}} = \mathrm{RG}_2\mathrm{RG}_1$. The corresponding flow of the coupling ratio is

$$r'/r = \gamma_2\gamma_{q-6}, \tag{109}$$

where again we remark that $q - 6$ is even when $q$ is even.

**$\{p, q\}$ tiling for odd $p \geqslant 5$ and even $q \geqslant 4$**

Consider the following RG steps

$$\mathrm{RG}_1 = \{aba \mapsto a, \ a \mapsto b\}, \tag{110}$$

$$\mathrm{RG}_2 = \left\{b^{(p-5)/2}ab^{p-6}ab^{(p-5)/2} \mapsto a, \ b^{(p-5)/2}ab^{(p-5)/2} \mapsto b\right\}, \tag{111}$$

$$\mathrm{RG}_3 = \left\{b^{(q-4)/2}ab^{q-5}ab^{(q-4)/2} \mapsto a, \ b^{(q-4)/2}ab^{(q-4)/2} \mapsto b\right\}, \tag{112}$$

$$\mathrm{RG}_4 = \left\{b^{(p-5)/2}(ab^{p-4})^{q-3}ab^{(p-5)/2} \mapsto a, \ b^{(p-5)/2}(ab^{p-4})^{q-4}ab^{(p-5)/2} \mapsto b\right\}. \tag{113}$$

These four RG steps combine to provide the sequence-preserving transformation $\Xi^{\{p,q\}} = \mathrm{RG}_4\mathrm{RG}_3\mathrm{RG}_2\mathrm{RG}_1$. After one application of $\Xi^{\{p,q\}}$ on the original aperiodic (XX or XXX) chain, the ratio $r$ between the hopping parameters transforms into $r'$ in such a way that

$$r'/r = \gamma_2\gamma_{p-5}\gamma_{p-3}\gamma_{q-4}. \tag{114}$$

Rather remarkably, $p - 5$, $p - 3$ and $q - 4$ are even precisely when $p$ is odd and $q$ is even.

## B.4 Class III: $\{p, q\}$ XXZ chain for odd $p$ and odd $q$

Finally, we consider here those $\{p, q\}$ modulations such that the corresponding $\Xi^{\{p,q\}}$ satisfy the condition (69) or (97) with $m = 3$.

**$\{3, q\}$ tiling for odd $q \geqslant 7$**

The three RG steps that contribute to the sequence-preserving transformation $\Xi^{\{3,q\}} = \mathrm{RG}_3\mathrm{RG}_2\mathrm{RG}_1$ read

$$\mathrm{RG}_1 = \{aba \mapsto a, \ a \mapsto b\}, \tag{115}$$

$$\mathrm{RG}_2 = \left\{b^{(q-7)/2}ab^{q-8}ab^{(q-7)/2} \mapsto a, \ b^{(q-7)/2}ab^{(q-7)/2} \mapsto b\right\}, \tag{116}$$

$$\mathrm{RG}_3 = \left\{b^{(q-7)/2}(ab^{q-6})^{q-5}ab^{(q-7)/2} \mapsto a, \ b^{(q-7)/2}(ab^{q-6})^{q-6}ab^{(q-7)/2} \mapsto b\right\}. \tag{117}$$

The SDRG flow of coupling ratio is

$$r'/r = \gamma_2\gamma_{q-7}\gamma_{q-5}. \tag{118}$$

Also in this case, all the integer indices of the $\gamma$ factors are even.

**$\{p, q\}$ tiling for odd $p \geqslant 5$ and odd $q \geqslant 5$**

For this class of $\{p, q\}$ modulations we need the following six RG steps

$$\mathrm{RG}_1 = \{aba \mapsto a, \ a \mapsto b\}, \tag{119}$$

$$\mathrm{RG}_2 = \left\{b^{(p-5)/2}ab^{p-6}ab^{(p-5)/2} \mapsto a, \ b^{(p-5)/2}ab^{(p-5)/2} \mapsto b\right\}, \tag{120}$$

$$\mathrm{RG}_3 = \left\{b^{(q-5)/2}(ab^{q-4})^{p-3}ab^{(q-5)/2} \mapsto a, \ b^{(q-5)/2}(ab^{q-4})^{p-4}ab^{(q-5)/2} \mapsto b\right\}, \tag{121}$$

$$\mathrm{RG}_4 = \{aba \mapsto a, \ a \mapsto b\}, \tag{122}$$

$$\mathrm{RG}_5 = \left\{b^{(q-5)/2}ab^{q-4}ab^{(q-5)/2} \mapsto a, \ b^{(q-5)/2}ab^{(q-5)/2} \mapsto b\right\}, \tag{123}$$

$$\mathrm{RG}_6 = \left\{b^{(p-5)/2}(ab^{p-4})^{q-3}ab^{(p-5)/2} \mapsto a, \ b^{(p-5)/2}(ab^{p-4})^{q-4}ab^{(p-5)/2} \mapsto b\right\}, \tag{124}$$

for constructing the sequence-preserving transformation $\Xi^{\{p,q\}} = \mathrm{RG}_6\mathrm{RG}_5\mathrm{RG}_4\mathrm{RG}_3\mathrm{RG}_2\mathrm{RG}_1$. After one application of $\Xi^{\{p,q\}}$, the coupling ratio $r$ transforms into $r'$ such that

$$r'/r = \gamma_2^2 \gamma_{p-5} \gamma_{p-3} \gamma_{q-3}^2. \tag{125}$$

Notice that, since both $p$ and $q$ are odd, $p-5$, $p-3$ and $q-3$ are even.

**$\{p,3\}$ tiling for odd $p \geqslant 7$**

The last set of modulations we are left with belongs to the exceptional class discussed in Appendix B.1. The inflation rule containing negative powers of letters can be written as

$$\sigma_{\{p,3\}} = \left\{ a \mapsto aa^{(p-5)/2}ba^{(p-5)/2}, \ b \mapsto a^{-1} \right\}. \tag{126}$$

By defining the RG steps

$$\mathrm{RG}_1 = \left\{ b^{(p-7)/2}(ab^{p-6})^{p-5}ab^{(p-7)/2} \mapsto a, \ b^{(p-7)/2}(ab^{p-6})^{p-6}ab^{(p-7)/2} \mapsto b \right\}, \tag{127}$$

$$\mathrm{RG}_2 = \left\{ a^{(p-5)/2}ba^{p-6}ba^{(p-5)/2} \mapsto a, \ a^{(p-5)/2}ba^{(p-5)/2} \mapsto b \right\}, \tag{128}$$

and the auxiliary transformation

$$\mathrm{RG}_0 = \{aba \mapsto a, \ a \mapsto b\}, \tag{129}$$

we can write

$$\mathrm{RG}_2\mathrm{RG}_1\mathrm{RG}_0 \sim \mathrm{RG}_0\sigma_{\{p,3\}}^{-3}. \tag{130}$$

Comparing this last equation with (97) for $m = 3$, we identify $\Xi^{\{p,3\}} = \mathrm{RG}_2\mathrm{RG}_1$, which allows to determine the SDRG flow of the coupling ratio

$$r'/r = \gamma_2\gamma_{p-7}\gamma_{p-5}, \tag{131}$$

where, again, all the integer indices of the $\gamma$ factors are even.

# C  Entanglement structure and improved bound for effective central charge

The entanglement structure of a tensor network state depends on the network structure as well as on the detail of its building blocks, *i.e.* the tensor states. In this appendix, we will study the entanglement structure of the tensor states defined in (65), which are the building blocks of the TN constructed in Sec. 5.1. We combine our results with the network structure to improve the bound (79) for the effective central charge of the $\{p,q\}$ aperiodic XXX spin chain.

## C.1  Entanglement in tensor states

The tensor states $|T_n\rangle$ in (65) are obtained by diagonalizing the local Hamiltonian $H_n$ of a block of $n$ consecutive spins given by (23). In this appendix we calculate the entanglement entropy and the Rényi entropies of $k$ spins in the tensor state $|T_n\rangle$, with $k < n$.

For $n = 1, 2$, both tensor states are EPR states, namely $|T_1\rangle = (|+\rangle |+\rangle + |-\rangle |-\rangle)/\sqrt{2}$ and $|T_2\rangle = (|+\rangle |-\rangle - |-\rangle |+\rangle)/\sqrt{2}$. The reduce density matrix for each spin is $I_2/2$, where $I_2$ is the 2-by-2 identity matrix. The entanglement spectrum is flat and both the entanglement entropy and all the Rényi entropies are $\ln 2$. Since the EPR states are maximally entangled, the tensors associated to the states $|T_1\rangle$ and $|T_2\rangle$ are perfect. Thus, if these two tensors are the only ones appearing in a given TN, the resulting entanglement entropy saturates the bound (74).

For $n \geqslant 3$, we have computed the entanglement entropies numerically. Notice that for these values of $n$ the states $|T_n\rangle$ exhibit a non trivial dependence on the anisotropy $\Delta_0$. Some of the results of our analysis are shown in Fig. 17, where we present the curves for two distinct values of $n$ (left and right panels respectively) as functions of the anisotropy parameter $\Delta_0$ (top panels) and of the Rényi index $\alpha$ (bottom panels). In this appendix we denote by $S^{(\alpha)}_{\{j_1,...,j_k\}}(|T_n\rangle)$ the $\alpha$-th Rényi entropy of $k$ spins, where the (distinct) integer numbers $j_1,...,j_k$ are respectively associated to the vectors $|m_{j_1}\rangle,...,|m_{j_k}\rangle$ in the definition (65) of $|T_n\rangle$. We recall that the limit of $S^{(\alpha)}_{\{j_1,...,j_k\}}(|T_n\rangle)$ for $\alpha \to 1$ provides the entanglement entropy $S_{\{j_1,...,j_k\}}(|T_n\rangle)$. In the following we will refer to the entanglement entropy and the Rényi entropies with $\alpha \neq 1$ collectively as entanglement entropies.

For odd $n \geqslant 3$, the tensor state $|T_n\rangle$ is constituted by $n$-spin doublets. The reduced density matrix of any single spin ($k = 1$) is $I_2/2$, and therefore the corresponding entanglement entropies are $S^{(\alpha)}_{\{j_1\}}(|T_n\rangle) = S_{\{j_1\}}(|T_n\rangle) = \ln 2$, for any $\alpha$ and $j_1 = 0, 1, ..., n$. On the other hand, the reduced density matrix of ($k \geqslant 2$) spins depends on $\Delta_0$ and it is not proportional to the identity matrix. We find that the entanglement entropies of these bipartitions are such that $\ln 2 < S_{\{j_1,...,j_k\}}(|T_n\rangle) < k \ln 2$ and $\ln 2 < S^{(\alpha)}_{\{j_1,...,j_k\}}(|T_n\rangle) < k \ln 2$, for any $\alpha$. This means that the tensors associated to $|T_n\rangle$ with odd $n \geqslant 3$ are not perfect tensors. Moreover, we have found that the Rényi entropies as functions of the Rényi index $\alpha$ do not follow the behavior occurring in homogeneous critical lattice models and CFTs, which, at leading order in the subsystem size, reads [62,63]

$$S^{(\alpha)} \propto 1 + \frac{1}{\alpha}. \tag{132}$$

The features discussed above can be observed, for the particular case of the tensor state $|T_3\rangle$, in the left panels of Fig. 17.

For even $n \geqslant 4$, the tensor state $|T_n\rangle$ is an $n$-spin singlet state. Our analysis provides findings very similar to the ones obtained for odd $n \geqslant 3$. In particular, the reduced density matrix of any single spin ($k = 1$) is $I_2/2$, while the reduced density matrix of $k \geqslant 2$ spins is not proportional at the identity and depends on $\Delta_0$. As for the entanglement entropies, we find that, for any $\alpha$, $S^{(\alpha)}_{\{j_1\}}(|T_n\rangle) = S_{\{j_1\}}(|T_n\rangle) = \ln 2$ (for any $j_1 = 1, ..., n$) and both $S^{(\alpha)}_{\{j_1,...,j_k\}}(|T_n\rangle)$ and $S^{(\alpha)}_{\{j_1,...,j_k\}}(|T_n\rangle)$ are strictly included between $\ln 2$ and $k \ln 2$. Thus, also the tensors associated to $|T_n\rangle$ with even $n \geqslant 4$ are not perfect tensors. Furthermore, the behavior of Rényi entropies as functions of $\alpha$ does not satisfy (132). These properties are shown in the right panels of Fig. 17, where the results obtained for the exemplary case $n = 4$ are reported.

To summarize, our analysis shows that the tensors associated to the tensor states (65) are not perfect when $n \geqslant 3$. Thus, the entanglement entropy obtained from a TN where states $|T_n\rangle$ with $n \geqslant 3$ do appear does not saturate (74). On the other hand, when only $|T_1\rangle$ and $|T_2\rangle$ appear in a TN, the corresponding entropy saturates (74). We have also investigated the behavior of the Rényi entropies as a function of Rényi index $\alpha$, finding that they behave differently from (132) (see also the discussion at the end of Sec. 4.1). This is a further hint of the fact that aperiodic XXZ chains cannot be described by underlying

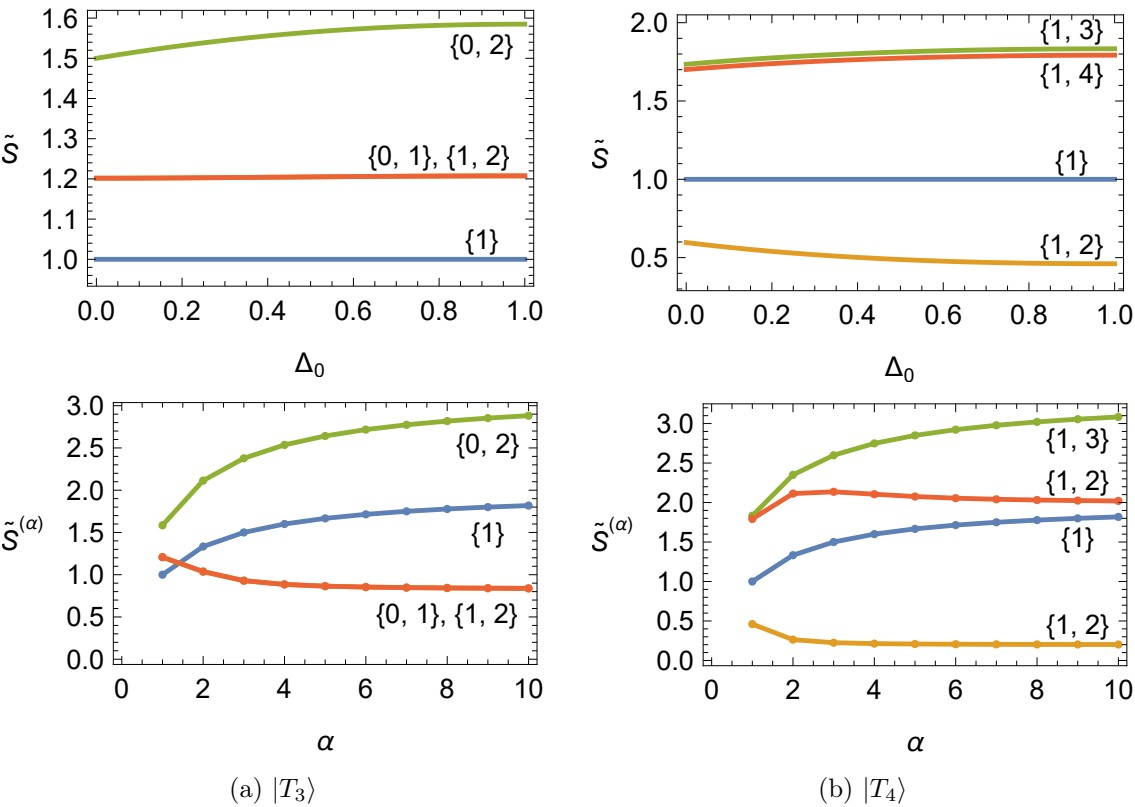

Figure 17: The reduced entanglement entropies $\tilde{S} \equiv S_{\{j_1,\ldots,j_k\}}(|T_n\rangle)/\ln 2$ as functions of $\Delta_0$ (top panels) and the reduced Rényi entropies $\tilde{S}^{(\alpha)} \equiv S^{(\alpha)}_{\{j_1,\ldots,j_k\}}(|T_n\rangle) \cdot 2/[(1+1/\alpha)\ln 2]|_{\Delta_0=1}$ as functions of the Rényi index $\alpha$ (bottom panels). We consider two values of $n$, namely $n = 3$ (left panels) and $n = 4$ (right panels). In each panel we present the entropies of different choices of the set of spins, labeled by $\{j_1\}$ or $\{j_1, j_2\}$. Notice that for the left panels, the entropies of the spins $\{0, 1\}$ and $\{1, 2\}$ are equal.

conformal field theories.

## C.2 An improved bound for effective central charges

The upper bound for the effective central charge, reported in the right hand side of the inequality (79), is obtained by assuming that all the tensors in the TN are perfect. Thus, this expression depends only on the geometry of the TN graph, without any information about the actual properties of the tensors. As explained in Sec. 5.1, each tensor is associated to one of the states reported in (65). Thus, studying the entanglement properties of these states, one can understand the actual contribution that each tensor gives to the entanglement of the whole TN. Exploiting the findings reported in Appendix C.1 above, it is possible to improve the upper bound (79) for the effective central charge associated to the ground state of any $\{p, q\}$ aperiodic XXX chain. In the following we describe how to derive these results generalizing the construction discussed in Sec. 5.3, and we provide the expression of the new bound as function of $q$ for two exemplary choices of $p$.

In order to compute the improved bound for the effective central charge, the fine-grained inflation rule $\tilde{\sigma}_{\{p,q\}}$ defined in Sec. 5.3 needs to be further refined. Each of the letters $a_i$ ($b_i$) is endowed with an additional index $n$ in such a way that $a_{in}$ ($b_{in}$) is the $i$-th letter of type $a$ ($b$) in the $n$-spin singlet/doublet along the TN layer which $a_{in}$ ($b_{in}$) belongs to. We denoted the further fine-grained inflation rule by $\tilde{\tilde{\sigma}}_{\{p,q\}}$. It can be obtained from the

original one $\sigma_{\{p,q\}}$ by considering the words $w_a(a,b)$, $w_b(a,b)$ and substituting each $a$ and $b$ with $a_{in}$ and $b_{in}$, according to the new classification introduced above. The number of different kinds of vertex in $\tilde{\tilde{\sigma}}_{\{p,q\}}$, *i.e.* the number of pairs of indices $(i,n)$ for the letters $a$ and $b$, depends in a highly non trivial way on the choice of $p$ and $q$. Once the new inflation rule $\tilde{\tilde{\sigma}}_{\{p,q\}}$ has been specified, the corresponding substitution matrix $\widetilde{\widetilde{M}}$ can be constructed in the same manner we obtained $\widetilde{M}$ from $\tilde{\sigma}_{\{p,q\}}$ in Sec. 5.3. Also the eigenvectors of $\widetilde{\widetilde{M}}$, $\tilde{\tilde{u}}_+$ and $\tilde{\tilde{v}}_+$, are defined following the construction of $\tilde{u}_+$ and $\tilde{v}_+$ from $\widetilde{M}$. Notice that, for the same reason explained in Sec. 5.3, the largest eigenvalue of $\widetilde{\widetilde{M}}$ is equal to the one of $\widetilde{M}$, which in turn is equal to $\lambda_+$ given in (14) as function of $p$ and $q$. Next, a new entanglement matrix $\widetilde{\widetilde{E}}$ has to be defined. Its entries no longer contain the number of legs cut by curves connecting two given vertices, but rather the actual entanglement entropy (modulo $\ln 2$) associated to each of the aforementioned legs. Since this entropy is not accessible *via* analytical computations, the entries of $\widetilde{\widetilde{E}}$ have to be obtained numerically, for instance from the results shown in Fig. 17. Notice that these components of $\widetilde{\widetilde{E}}$ are smaller than the ones of $\widetilde{E}$ because the latter ones contain the entanglement contributions to the network assuming that all the tensors are perfect, *i.e.* maximally entangled.

Combining all these ingredients, we can improve the bound (79) for SDRG flows of class I as follows

$$c_{\text{eff}} \leqslant \frac{6 \sum_{ij} \widetilde{\widetilde{E}}_{ij} \widetilde{\widetilde{M}}_{ij} \tilde{\tilde{u}}_i \tilde{\tilde{v}}_j}{\lambda_+ \ln \lambda_+ \sum_i \tilde{\tilde{u}}_i \tilde{\tilde{v}}_i} \ln 2 \leqslant \frac{6 \sum_{ij} \widetilde{E}_{ij} \widetilde{M}_{ij} \tilde{u}_i \tilde{v}_j}{\lambda_+ \ln \lambda_+ \sum_i \tilde{u}_i \tilde{v}_i} \ln 2 \,. \qquad (133)$$

Similarly to the bound computed in Sec. 5.3, this new bound can also be generalized to SDRG flows of class II and III, by replacing $\left( \widetilde{\widetilde{M}}, \lambda_+ \right)$ with $\left( \widetilde{\widetilde{M}}^2, \lambda_+ \right)$ and $\left( \widetilde{\widetilde{M}}^3, \lambda_+ \right)$ respectively. Although this approach provides an improved bound for the effective central charge through a more careful analysis for the entanglement entropy of the tensor states (65), we cannot achieve the exact entanglement entropy in this way. The reason is the following: when computing the variation of the entropy along one inflation step of the tiling, (133) overestimates the entanglement associated to those doublet states, *i.e.* $T_{m_1 \ldots m_n}^{m_0}$ in (65), where legs associated to an upper and a lower index are simultaneously cut. Moreover, we stress that (133) holds as bound on the effective central charge of $\{p,q\}$ aperiodically modulated XXX chain only. Indeed, in this regime, all the $\{p,q\}$ modulations are relevant and at the strong-disorder fixed point the SDRG, on which our TN is fully based, becomes exact.

We now consider two explicit examples where we apply the above mentioned analysis. In the first one, we focus on the $\{4,q\}$ aperiodic XXX chain (with $q \geqslant 5$), whose corresponding SDRG flow belongs to class I (see Appendix B). For this modulation, the inflation rule $\tilde{\tilde{\sigma}}_{\{4,q\}}$ reads

$$\tilde{\tilde{\sigma}}_{\{4,q\}} = \left\{ a_{00} \mapsto b_{13} a_{00} (b_{12} a_{00})^{q-4} b_{13}, \ b_{1n} \mapsto b_{13} a_{00} (b_{12} a_{00})^{q-5} b_{13} \right\}, \quad n = 2, 3. \qquad (134)$$

By ordering the vertices appearing in this rule as $(a_{00}, b_{12}, b_{13})$, the substitution matrix is given by

$$\widetilde{\widetilde{M}}_{\{4,q\}} = \begin{pmatrix} q-3 & q-4 & q-4 \\ q-4 & q-5 & q-5 \\ 2 & 2 & 2 \end{pmatrix} . \qquad (135)$$

In order to determine the entanglement matrix, we notice that the presence of both $b_{12}$ and $b_{13}$ requires us to know the entanglement of a single spin in a block made up of two

and three spins, respectively. As discussed above, the entanglement of a single spin in a 2-spin block is $\ln 2$. Moreover, by looking at the blue curve in the top right panel of Fig. 17, we notice that $\ln 2$ is also the entanglement of a single spin into a 3-spin block. Thus, the entanglement matrix reads

$$
\widetilde{\widetilde{E}}_{\{4,q\}} = \begin{pmatrix} 0 & 0 & 0 \\ 1 & 1 & 1 \\ 1 & 1 & 1 \end{pmatrix} . \tag{136}
$$

Plugging (135) (with its eigenvectors) and (136) into (79), we obtain

$$
c_{\text{eff}}(4, q) \leqslant \frac{3 \ln 2}{\ln \left( \sqrt{(q-2)(q-4)} + q - 3 \right)} . \tag{137}
$$

Notice that the right hand side of (137) is equal to the right hand side of (86) when $p = 4$. This means that in the bound (133) is the same as (79) when $p = 4$. This is particular to the $\{4, q\}$ modulation and does not hold in general. We can convince ourselves of this by considering as a second example the SDRG flow induced by the $\{8, q\}$ modulation, with $q \geqslant 4$. This flow belongs to the class I as well and its fine-grained inflation rule reads

$$
\tilde{\tilde{\sigma}}_{\{8,q\}} = \{ a_{in} \mapsto a_{13} a_{00} b_{12} (a_{00} a_{14} a_{24} a_{14} a_{00} b_{12})^{q-3} a_{00} a_{13},
$$
$$
b_{12} \mapsto a_{13} a_{00} b_{12} (a_{00} a_{14} a_{24} a_{14} a_{00} b_{12})^{q-4} a_{00} a_{13}, \quad n = 3, 4 \} . \tag{138}
$$

The corresponding substitution matrix, in the basis $(a_{00}, a_{13}, a_{14}, a_{24}, b_{12})$, is

$$
\widetilde{\widetilde{M}}_{\{8,q\}} = \begin{pmatrix} 2q-4 & 2q-4 & 2q-4 & 2q-4 & 2q-6 \\ 2 & 2 & 2 & 2 & 2 \\ 2q-6 & 2q-6 & 2q-6 & 2q-6 & 2q-8 \\ q-3 & q-3 & q-3 & q-3 & q-4 \\ q-2 & q-2 & q-2 & q-2 & q-3 \end{pmatrix} . \tag{139}
$$

For constructing the entanglement matrix we now need not only the entanglement entropy of a single spin in a 2- or 3-spin block (see discussion above), but also the entanglement entropy of two spins in a 4-spin block. The latter can be obtained numerically and its result is reported in the top right panel of Fig. 17 (cf. yellow curve). By considering $\Delta_0 = 1$, the entropy modulo $\ln 2$ reads $s \simeq 0.46$ and therefore the entanglement matrix is given by

$$
\widetilde{\widetilde{E}}_{\{8,q\}} = \begin{pmatrix} 0 & 0 & 0 & 0 & 0 \\ 1 & 1 & 1 & 1 & 1 \\ 1 & 1 & 1 & 1 & 1 \\ s & s & s & s & s \\ 1 & 1 & 1 & 1 & 1 \end{pmatrix} . \tag{140}
$$

Exploiting (139) with its eigenvalues and (140) in (133), we find

$$
c_{\text{eff}}(8, q) \leqslant \frac{9q - 24 - s \left( 9q - 2 \left( 12 + \sqrt{3(q-2)(3q-8)} \right) \right)}{(3q-8) \ln \left( 3q - 7 + \sqrt{3(q-2)(3q-8)} \right)} \ln 2. \tag{141}
$$

Notice that the right hand side of (141) is strictly smaller than the right hand side of (86) with $p = 8$, for any value of $q \geqslant 4$ and therefore (141) provides an actual improvement of the bound on the effective central charge.

Summarizing the results of this appendix, we have found that even though we can compute the precise entanglement structure of each tensor state, the exact entanglement entropy of a generic block of consecutive spins in the TN state is still analytically intractable. This obstacle is also present for the exact effective central charge of the aperiodic XXX spin chain. The reason is that the entanglement is not local in the these TNs, namely the entanglement entropy depends on the global structure of the RG trajectories, which is not localized throughout separated steps of deflation. However, as discussed in Sec. 4 and Sec. 5.3, there are exceptions for which the exact entanglement entropy can be obtained, namely those tensor networks where only tensor states $|T_1\rangle$ and $|T_2\rangle$ appear.

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
