# Peer review of "Towards Explicit Discrete Holography: Aperiodic Spin Chains from Hyperbolic Tilings"

_SciPost Physics_

## Round 1 · Referee Report · Anonymous (Referee 1) · 2022-7-20

Strengths
2 - The paper adopts differing perspectives and methodologies, thereby cross-checking and firmly solidifying some of its results.
3 - The paper is clearly written with each step of its derivations carefully explained, allowing any qualified reader to learn the techniques it develops.
Weaknesses
2 - Overall, more conciseness would be desirable.
Report
The main findings of the paper include a derivation of a broad Ryu-Takayanagi type formula that exhibits the dependency of entanglement entropy on system size and choice of regular tiling, a similar formula for specific boundary spin chains (generalizing formulas from 2007), and a charting of the RG flow and IR phase diagram of these spin chains in a perturbatively accessible regime.
The various findings of the paper raise several questions and further challenges. It would be fascinating if the authors were able to provide more exact quantitative matches between bulk and boundary theories, e.g. not matching only the logarithmic growth of entanglement entropy but also matching coefficients. Relatedly, it would be interesting to develop a physical understanding of the dependency the authors find on the choice of discretization. After all, more broadly when a physicist studies discretization (say 2d quantum gravity and matrix models), one generally hopes to understand some universal physics that does not depend on details of discretization and agree with the continuum theory. It is also difficult to imagine that there does not exist some elaboration of the formalism of the paper that would allow central charge to admit of an interpretation in terms of degrees of freedom.
In any event, the paper represents a definite step forward in the study of discrete holography. More precisely it contains a nice assemblage of intriguing smaller steps that weave together different approaches to discrete holography and open up new pathways to the furtherance of the field.
The paper is well-written in a clear and flawless prose. To make the paper self-contained the authors have chosen to include reviews of necessary background information (such as on tilings and strong disorder renormalization group flow). This choice, together with the careful pedagogical style, does lead to a rather lengthy exposition.
The selection of equations I have checked have all withstood scrutiny and I have no doubt as to the validity of the paper's findings.
In light of the above comments, it is my opinion that the paper merits publication, though I do have a couple of suggestions for possible improvements to the authors.
Requested changes
1- It is not entirely clear why the authors choose to work with two types of vertices, a and b. Generally there are more than two kinds of non-isomorphic vertices (you could distinguish between red vertices depending on whether they are incident to blue vertices or by how far removed they are from blue vertices in their own layer). It may be worth motivating this choice. Is it a balance between exhibiting a wider range of physics compared to just one type of vertex whilst also enjoying greater computational tractability than is the case when operating with three or more kinds of vertices?
2– The definition of a and b type vertices given on page 10 (determined by whether or not a vertex is incident to a vertex in the previous layer) does not apply to the case p = 3 and it appears a separate definition is needed (I believe the definition in this case is that red vertices are incident to one vertex in the previous layer, blue to two).
3– The discussion on page 21 took me a little longer to decipher than needed, and I believe some small changes could result in greater ease of reading. For example the authors could add primes to the letters on the RHS in (36), and add a short comment that the reason for the new names a' and b' is that the a bond is stronger than the renormalized aba bond (the old weak bond becomes the strong bond). Similar comments apply to page 22. Also, below (39), when saying that the resulting sequence is now the original one, maybe it's worth referring back to equation (35) since in a sense the RG steps described precisely undo this inflation rule.
4– On pages 24-25 the authors use superscripts (o) and (e). Looking at equations (42) to (45) would it not make more sense to swap these superscripts since the (o) superscript is associated to even indices for \Lambda and \rho, while (e) is associated to odd indices?
5– In equation (55) it seems to me that the upper limit of the second sum should be n(L)/2 rather than n(L)/2-1. Since n(L) is even, there should be the same number of odd and even terms.
6– I would suggest that the authors consider adding a brief comparison to some related works which offer complementary explorations of discrete holography: 1812.04057 (the current draft does already include a discussion of p-adic AdS/CFT but since this paper might be the p-adic paper most closely related to the current draft, perhaps it's worth highlighting) and 1807.05942, 1906.02305.
Author: Giuseppe Di Giulio on 2022-08-29 [id 2768]
(in reply to Report 1 on 2022-07-20)
We are grateful to the referee for the useful and insightful comments which arose from a thorough and meticulous analysis of our manuscript. We have revised our manuscript and replied to these observations. A detailed account of the changes is given below, where we follow the enumeration in the referee's original report.
-
We agree with the referee that we have made a choice based on computational tractability. To further motivate the choice of binary inflation rules in our setup, we have added two comments, one in the Introduction on page 4 and the other at the end of Sec.2.2 on page 12. In detail, the choice of classifying the vertices of the tiling into two types leads to binary inflation rules generating the aperiodic boundary. The relation between binary aperiodic sequences and spin chains is well-known in the literature and various techniques, including the strong-disorder renormalization group (SDRG), were available for our analyses. A more general treatment, involving more types of vertices, would require to improve the machinery of the SDRG for the boundary theory. From a technical point of view such a generalization seems feasible but computationally challenging. For instance, an SDRG procedure on n types of letters would have to keep track of n-1 ratios of the couplings. This makes finding the maximal coupling analytically and controlling the hierarchy of these couplings very difficult. Such a generalization constitutes an interesting avenue for future research in and of itself. We would like to point out that a generalization involving more types of vertices has been considered in Sec.5.3 and appendix C of the manuscript. In this case the different vertices are classified by using the tensor network introduced in Sec.5.1. Given that the tensor network is an a posteriori construction in our setup, introduced after having defined the boundary spin chain, such a classification could not have been used in the beginning of the manuscript. In other words, this classification is bound to the tensor network structure rather than to pure inflation rules as introduced for the generation of the tiling.
-
To address the comment of the referee, on page 11, we have added a clarification about the difference between the classification of the types of vertices in {p,q} tilings with p\neq 3 and the one in {3,q} tilings.
-
We agree with the referee that it is worth adding some clarifications on these points and therefore we have included two comments below Eq.(37) and Eq.(39). However, we prefer not to modify the notations for the following reasons. The types of vertices, i.e. the letters a and b do not get renormalized by the RG step. This only renormalizes the values of the couplings, which become quantitatively different. This difference is what we denote with the primes. For this reason, we believe that adding primes to the letters as well would be misleading, since the types of vertices are always the same. In other words, the letter sequence after the RG step still contains solely the letters a and b, but their associated couplings now have a different value J'_a and J'_b, respectively. Moreover, we would like to maintain the notation used in the standard literature (see for instance [60,94] in the manuscript).
-
We agree with the referee that this notation might be misleading at first, so have added a comment clarifying it between Eq.(43) and Eq.(44), along the following lines. The notation we have used reflects the fact that the typical lengths of a given generation of singlets are dictated by the properties of the previous generation. Thus, the generations labeled by odd numbers directly affect the ones labeled by even numbers and viceversa. For this reason, we would like to keep the notation as it is in the first version of the manuscript.
-
We confirm that the issue pointed out by the referee was a typo and we have fixed it in Eq.(55).
-
We have added to the manuscript the three references suggested by the referee. A brief description of them has been inserted in the overview of the related literature on page 3.
Anonymous on 2022-09-06 [id 2792]
(in reply to Giuseppe Di Giulio on 2022-08-29 [id 2768])The authors have carefully considered all my suggestions and made edits where appropriate. The paper merits publication.
Author: Giuseppe Di Giulio on 2022-08-29 [id 2769]
(in reply to Report 2 on 2022-08-19)We are grateful to the referee for the interesting comments and the deep questions which arose from a careful reading of the manuscript. We have revised our manuscript and addressed these points. The account of the changes is given below, where we follow the enumeration in the referee's original report.
Indeed, as the referee is pointing out, writing down a discrete version of the GKPW dictionary is the ultimate goal of the discrete holography program and many of our future perspectives go in this direction. To address this comment, at the end of Sec.5.4, we have added a schematic computation for correlation functions of bulk fields from boundary data exploiting our TN construction. This has been accompanied by a few lines on page 51 to highlight this application of our setup. Our work focuses on only one entry of such a dictionary, namely the correspondence between ground state of boundary theories and bulk without matter fields. Qualitatively, our results suggest that the entry in this discrete holographic dictionary consists of a tiling without degrees of freedom, which is dual to the ground state of the boundary Hamiltonian. Quantitatively, the main obstacle to provide a rigorous derivation of a GKPW-like formula is finding the bulk action corresponding to our proposed boundary Hamiltonian. Insights on this aspect can be gained in the tensor network (TN) approach considered in Sec.5 of the manuscript. Indeed, one can compute matrix elements of local bulk fields on the TN based on boundary data. In principle, these can be used to reconstruct the bulk action corresponding to such fields.
If we have understood correctly the comments of the referee, the question is whether our construction exists in higher dimensions and if there is a notion of fractal dimension in the bulk. We have added a comment on these topics on page 51, summarizing the following discussion. In our work, we have started from the simplest yet non-trivial spacetime, i.e. Euclidean AdS_2, based on our motivation of discretizing a constant time-slice of AdS_{2+1}. There indeed exist tessellations of hyperbolic space in dimensions higher than two. These naturally require regular polyhedra rather than polygons. Some key properties of the boundaries of regular tessellations of hyperbolic d-dimensional spaces, as for instance the fractal dimension of the limit set, drastically change when we go from d=2 to d>2. Thus, we believe this generalization to be a promising line of future research, as we now also have mentioned in the text. Furthermore, in this work we consider fixed tilings in the bulk, i.e. discrete geometries without dynamical degrees of freedom, implying that the only dimension we can assign to the bulk is that of the underlying continuum space. However, we cannot exclude that a notion of fractal dimension arises by considering other bulk states (corresponding to excited states of the boundary theory), where the bulk dynamics of play a more prominent role.
In the text just below Eq.(12), we have added the example of a specific tiling, which also appears in Fig.2, to facilitate the comparison between the information in the text and the figure.

---

## Round 1 · Referee Report · Anonymous (Referee 2) · 2022-8-19

Report
The paper is well written and appears correct. Its scope is broad and covers many aspects of discrete holography with interesting and valuable developments. I recommend publication with optional suggestions and questions detailed below.
Requested changes
1- Given the explicit spin-chain models invoked in this work, assuming they have such a discrete dual, do they provide any insight into how to formulate a partition function for discrete holography, along the lines of a discrete GKPW dictionary? 2- The tiling is explicitly 2d by construction (it is grown according to rules that ensure that it is manifold-like). But more generally is there an expectation that the dimension of spacetime should be so well defined for discrete quantum gravity? Relatedly, is there a notion of fractal or diffusion dimension on the {p,q} tilings that is different from d=2? 3- When the notion of an inflation rule is introduced, I think some simple explicit examples would help the presentation significantly.

---

## Editorial Decision

unknown